# Opposite effects of aerosols and meteorological parameters on warm clouds in two contrasting regions over eastern China

Yuqin Liu[1,4,7], Tao Lin[1,4,7], Jiahua Zhang[2], Fu Wang[3], Yiyi Huang[1], Xian Wu[1,7], Hong Ye[1], Guoqin Zhang[1], Xin Cao[1,4,7], Gerrit de Leeuw[5,6]

1 Key Lab of Urban Environment and Health, Institute of Urban Environment, Chinese Academy of Sciences, Xiamen 361021, China

2 Key Laboratory of Digital Earth Sciences, The Aerospace Information Research Institute, Chinese Academy of Sciences, Beijing 100094, China

3 CMA Earth System Modeling and Prediction Centre (CEMC), Beijing 100081, China

4 Fujian Key Laboratory of Digital Technology for Territorial Space Analysis and Simulation, Fuzhou 350108, China

5 Royal Netherlands Meteorological Institute (KNMI), R&D Satellite Observations, 3730AE De Bilt, The Netherlands

6 Aerospace Information Research Institute, Chinese Academy of Sciences (AirCAS), No.9 Dengzhuang South Road, Haidian District, Beijing 100094, China

7 Xiamen Key Laboratory of Smart Management on the Urban Environment, Xiamen 361021, China

*Correspondence to: Gerrit de Leeuw( gerrit.de.leeuw@knmi.nl), Tao Lin (tlin@iue.ac.cn)*

**Abstract.** The sensitivity (S) of cloud parameters to the influence of different aerosol and meteorological parameters has in most previous aerosol-cloud interaction (aci) studies been addressed using traditional statistical methods. In the current study, relationships between cloud droplet effective radius (CER) and aerosol optical depth (AOD, used as a proxy for cloud condensation nuclei, CCN), i.e. the sensitivity (S) of CER to AOD, is investigated with different constraints of AOD and cloud liquid water path (LWP). In addition to traditional statistical methods, the geographical detector method (GDM) is applied in this study to quantify the relative importance of the effects of aerosol and meteorological parameters, and their interaction, on S. Moderate Resolution Imaging Spectroradiometer (MODIS) C6 L3 data and European Centre for Medium-Range Weather Forecasts (ECMWF) ERA-5 reanalysis data, for the period from 2008 to 2022, were used to investigate aci over eastern China. Two contrasting areas were selected: the heavily polluted Yangtze River Delta (YRD) and a relatively clean area over the East China Sea (ECS). Linear regression analysis shows that CER decreases with the increase of AOD (negative S) in the moderately polluted atmosphere (0.1<AOD<0.3) over the ECS, whereas, in contrast, CER increases with increasing AOD (positive S) in the polluted atmosphere (AOD>0.3) over the YRD. Evaluation of S as function of the LWP shows that in the moderately polluted atmosphere over the ECS, S is negative in the LWP interval [40 g m$^{-2}$, 200 g m$^{-2}$], and the sensitivity of CER to AOD is substantially stronger as LWP is larger. In contrast, in the polluted atmosphere over the YRD, S is positive in the LWP interval [0

g m$^{-2}$, 120 g m$^{-2}$] and does not change notably as function of LWP in this interval. The study further shows that over the ECS the CER is larger for higher LTS and RH but lower for higher PVV. Over the YRD, there is no significant influence of LTS on the relationship between CER and AOD. The GDM has been used as an independent method to analyse the sensitivity of cloud parameters to AOD and meteorological parameters (relative humidity, RH; lower tropospheric stability, LTS; and pressure vertical velocity, PVV). The GDM has also been used to analyse the effects of interactions between two parameters and thus obtain information on confounding meteorological effects on the aci. Over the ECS, cloud parameters are sensitive to almost all parameters considered except for cloud top pressure (CTP), and the sensitivity to AOD is larger than that to any of the meteorological factors. Among the meteorological factors, the cloud parameters are most sensitive to PVV and least sensitive to RH. Over the YRD, the explanatory power of the sensitivity of cloud parameters to AOD and meteorological parameters is much smaller than over the ECS, except for RH which has a statistically significant influence on CTP and can explain 74% of the variation of CTP. The results from the GDM analysis show that cloud parameters are more sensitive to the combination of aerosol and a meteorological parameter than to each parameter alone but confounding effects due to co-variation of both parameters cannot be excluded.

**Key words:** AOD, Cloud parameters, LWP, Geographical detector method, Confounding effects, MODIS, East China

## 1 Introduction

The atmosphere is primarily composed of gases, i.e. nitrogen, oxygen and several noble gases, as well as a wide variety of trace gases that occur in relatively small and highly variable amounts. In addition, liquid and solid particles are suspended in the atmosphere. The suspension of solid and liquid particles in the gaseous medium is technically defined as an aerosol, but usually the term aerosol refers to the particulate component only (Seinfeld and Pandis, 1998). The aerosol particles originate from a large variation of both direct and indirect sources. The concentrations and chemical and physical properties of aerosol particles change under the influence of a variety of atmospheric processes and thus are variable in space and time. The residence time of tropospheric aerosol particles varies from hours to weeks (Bellouin et al., 2020), depending on particle size and atmospheric conditions. Directly emitted aerosol types include,

e.g., sea spray, dust, smoke, volcanic ash, pollen etc. Secondary formation of aerosol particles occurs through nucleation and subsequent growth by physical and chemical processes such as condensation, coagulation and multiphase chemical reactions on the particle surface, involving precursor gases such as sulphur dioxide ($SO_2$), nitrogen dioxide ($NO_2$), ammonia ($NH_3$), volatile organic compounds (VOCs),

etc.

Aerosol particles are important for climate, air quality and heterogenous chemical processes. Aerosol particles affect climate by their interaction with radiation (aerosol radiation interaction, ari) which exerts a radiative forcing on the Earth energy budget, which results in rapid adjustments of global mean atmospheric quantities such as temperature. The sign and strength of radiative forcing (RF) due to ari

($RF_{ari}$) vary with environmental parameters (Bellouin et al., 2020). In particular, aerosol particles scatter incoming solar radiation back into space, but the effect of $RF_{ari}$ depends on the brightness of the aerosol with respect to that of the underlying surface. The scattering of (bright) aerosol over a darker surface results in cooling and reduction of the warming effect of greenhouse gases (GHG). In contrast, the interaction of absorbing aerosol particles with solar radiation may result in local heating and thus

reinforce the GHG effect and influence meteorological processes.

Aerosol particles can act as cloud condensation nuclei (CCN, in liquid clouds) or ice nucleating particles (INP, in ice clouds), depending on their chemical composition and size. When CCN are activated they can modify cloud microphysical properties and precipitation and thus indirectly influence the Earth's radiative budget (aerosol-cloud interactions, aci) (Tao et al., 2012; Fan et al., 2016; Rosenfeld et al., 2019;

Rao and Dey, 2020; Bellouin et al., 2020). An increase in CCN concentrations leads to an increase in the number of cloud droplets ($N_d$) and, if the cloud liquid water path (LWP) remains unchanged, the decrease of the cloud droplet effective radius (CER). The smaller CER in turn results in the enhanced reflection of solar radiation and thus cloud albedo and enhanced RF due to aci ($RF_{aci}$). This effect of the increase of the number of aerosol particles on cloud properties at constant LWP is often referred to as the

"Twomey" effect (Twomey, 1977; Feingold, et al., 2001; Matheson et al., 2005; Koren et al., 2005; Meskhidze and Nenes, 2010; Costantino et al., 2010; 2013). Another component of RFaci are rapid adjustments which may also lead to the modification of other cloud properties in response to the increase of $N_d$ and decrease in CER, such as a decrease in precipitation efficiency, resulting in the increase of the LWP and the amount of clouds, thus enhancing the reflection of solar radiation (Albrecht, 1989). These

two effects of aci are sometimes referred to as the cloud albedo and cloud lifetime effects (Quaas et al., 2008).

The CER is an important factor affecting cloud physical processes and optical properties. Slingo (1990) pointed out that a reduction in the average CER by 15% - 20% can balance the radiative forcing at the top of the atmosphere caused by a doubling of carbon dioxide. Therefore, small changes in cloud

microphysical properties may lead to important climate impacts (Zhao et al., 2018). Further study on the sensitivity of CER to aerosols ($S_{CER-A}$, further referred to as S), together with meteorological parameters influencing aci, can improve our understanding of these processes and the effects of aci on RF, leading to improved aerosol-cloud parameterizations in regional climate models. The variation in $N_d$ with CCN is referred to as the susceptibility β (β= $d \ln N_d / d \ln A$; e.g., Gryspeerdt et al. (2023)) and the variation

of CER with CCN is referred to as the sensitivity S (eq. 1 in Section 3.1). Much of the variation of aerosol-cloud effective radiative forcing in ensembles of climate models is due to the variation in β, while β is also central to the strength of cloud adjustments (Gryspeerdt et al., 2023).

The sensitivity of microphysical properties of clouds to aerosol have been studied based on data from a large number of monitoring campaigns, using satellite, aircraft and ground based observations, and by

using model simulations. Because of the large spatial coverage, satellite instruments have been widely used to study aerosol-cloud interaction in different conditions, confirming the high sensitivity of cloud properties to aerosol (e.g., Yuan et al., 2008; Rosenfeld et al., 2014; Saponaro et al., 2017; Liu et al., 2018; Pandey et al., 2020; Christensen et al., 2020; Liu et al., 2021). In studies on S utilizing satellite data, which is the subject of the current study, the aerosol optical depth (AOD) is often used as a proxy

for the aerosol concentration, which is justified by the correlation of AOD and CCN published by Andreae (2009). However, AOD is determined by all aerosol particles in the atmospheric column, including particles that do not act as CCN, depends on the relative humidity (RH) throughout the atmospheric column, does not provide information on chemical composition and may be influenced by aerosol in disconnected layers. The use of the Aerosol Index (AI), the product of AOD and the Ångström

Exponent (AE; describing the spectral variation of AOD), is suggested as a better indicator of CCN because AE includes information on aerosol size (e.g., Nakajima et al., 2001). However, the AE is determined from AOD retrieved at two or more wavelengths and the evaluation of the results versus ground-based reference data shows the large uncertainty in AE. Therefore, in recent MODIS product

Collections, AE is not provided over land (e.g., Levy et al., 2013; Kourtidis et al., 2015). AE is also not well-defined for low AOD for which uncertainty is largest (Bellouin et al., 2020; Gryspeerdt et al., 2023). The issues associated with using AOD or AI as proxy for CCN were discussed by, among others, Rosenfeld et al. (2014) who do not recommend the use of AI while also concluding that no better proxy is available. Therefore, in this study, AOD is used as a proxy for CCN to study S. It is noted that in other studies, e.g., Jia et al., 2022, both AOD and AI have been used and the results show similar behaviour.

Many studies confirmed the Twomey effect (e.g., Chen et al., 2014; Christensen et al., 2016; Jia et al., 2019). However, other studies show that, over some areas and especially over land in situations with high AOD, the CER increases with the increase of AOD, in contrast to the hypothesis of the "Twomey effect" (e.g., Feingold et al., 2001; Yuan et al., 2008; Grandey and Stier, 2010; Tang et al., 2014; Wang et al., 2015; Jia et al., 2019; Liu et al., 2020). It is noted that in these studies, the relationship between CER and aerosol concentration was not constrained by LWP, although this is the premise of the Twomey effect. Meteorological conditions are important factors determining both the occurrence of clouds and cloud properties and therefore, in aci studies, the variation of meteorological conditions needs to be considered together with the variation of AOD (e.g., Myhre et al., 2007; Tang et al., 2014). On the one hand, meteorological parameters influence the Twomey effect. Jones et al. (2009) concluded that vertical motion, aerosol type, and aerosol layer height do make a significant contribution to $RF_{aci}$ and that these factors are often more important than total aerosol concentration alone and that the relative importance of each differs significantly from region to region. Wang et al. (2014) proved that the well-recognized aerosol effect mingles with meteorological conditions (RH and PVV), which likely is the main reason for the positive values of S over land. Tang et al. (2014) observed the Twomey effect over ocean, but a positive CER-AOD relationship over Eastern China which they attributed to changes in relative humidity and wind fields. Tang et al. (2014) concluded that "our results suggest that the effect of meteorology may not be negligible when investigating the aerosol indirect effect on a large scale, especially when the weather conditions are complex and change frequently.". Andersen and Cermak (2015) studied biomass burning aerosol over the Atlantic Ocean (Sep-Dec) in stable and unstable environments (LTS) and observed that the aerosol effect is stronger in unstable environment, especially during biomass burning episodes. These authors concluded that "the observed absolute differences in CER between stable and unstable environments are driven by cloud dynamical effects (CER and LWP are positively associated),

or meteorology". Jia et al. (2020) inferred that S increases remarkably with both cloud-base height and cloud geometric thickness (proxies for vertical velocity at cloud base), suggesting that stronger aci generally occurs under larger updraft velocity conditions. On the other hand, the meteorological parameters also influence the potential adjustments. Koren et al. (2010) reported that observed cloud top height and cloud fraction correlate best with model pressure updraft velocity and relative humidity. Quaas et al. (2010) discussed the relationship between total cloud cover and AOD, often observed in satellite data, based on model simulations to test six hypotheses. These authors concluded that the increase of aerosol optical depth that accompanies the swelling of aerosol particles in humid airmasses is the dominant process contributing to the observed correlation, confirming earlier conclusions by Myhre et al. (2007). Boucher and Quaas (2012) reported that aerosol humidification has a large impact on the relationship between AOD and rain rate and that discriminating the data into classes of pressure vertical velocity and/or relative humidity does not eliminate these meteorological effects. Gryspeerdt et al. (2014) studied the relationship between aerosol and initial cloud cover as a function of RH and vertical convection strength. Liu et al. (2017) showed that the increase in cloud cover is promoted in an environment with high RH. A rising air mass can promote the formation of thicker and higher clouds.

The above are examples of studies addressing the influence of different aerosol and meteorological parameters on the sensitivity of cloud parameters to aerosol and potential confounding effects. Most of them used traditional statistical methods or stratified the data according to confounding meteorological parameters (e.g., Saponaro et al., 2017; Ma et al., 2018). In the current study the geographical detector method (GDM) is applied as a complementary tool to quantify the relative importance of the effects of nine parameters on S. The GDM is explained in detail in Section 3.2. In brief, a set of statistical methods is used to detect the spatial variability of aerosol and cloud properties, which are spatially differentiated, and evaluate the occurrence of correlations in their behaviour and the driving forces behind these correlations (Wang and Hu, 2012; Wang et al., 2016). The basic idea of the GDM is that the spatial distributions of two variables tend to be similar if these two variables are connected (Zhang and Zhao, 2018). The method is used in this study to analyse the relative importance of different factors, and interactions between them, influencing aci.

The focus of the current study is to establish a CER-aerosol parameterization scheme by the application of the GDM to satellite data over two contrasting areas, i.e. the Yangtze River Delta (YRD) in eastern

China, with high aerosol concentrations, and a relatively clean area over the East China Sea (ECS). The satellite data are first used to study the CER sensitivity to aerosol for different AOD regimes and all LWP values, followed by constraining the LWP in different intervals. It is noted that $RF_{aci}$ is formulated in terms of $N_d$, whereas studies on the Twomey effects often use CER alone instead of $N_d$, such that they were not really looking at the Twomey effect in isolation and not really studying the RFaci either (McComiskey and Feingold, 2012). CER is readily available as a satellite retrieval product, although in particular over land the reliability is questioned (Grandey and Stier, 2010), whereas $N_d$ is derived from CER and the cloud optical thickness (COT) (e.g., Grandey and Stier, 2010; Arola et al., 2022). While $N_d$ is affected by biases in the CER retrieval, these are different to the CER biases alone (and in some cases may offset each other; Painemal and Zuidema, 2011). For marine stratocumulus clouds, the $N_d$ retrieval appears to be surprisingly accurate (Gryspeerdt et al, 2022). The comparison of global maps of the sensitivities of CER and $N_d$ to AOD by Grandey and Stier (2010) exhibits very similar patterns. In this study, the CER sensitivity to AOD is stratified by LWP, which however poses problems in the evaluation of $RF_{aci}$. However, the current study focuses on understanding effects of different parameters on CER sensitivity to aerosol rather than the application to determine $RF_{aci}$.

The results from the CER sensitivity study are used to guide the application of GDM to determine the relative effects of different parameters on aci. Relations between CER and AOD, meteorological conditions and several cloud properties are determined, including combined effects of different influencing parameters.

## 2 Approach

### 2.1 Study area

The complex aerosol composition and the high aerosol concentrations render eastern China an interesting area for a variety of studies of processes involving aerosols, including the current study on the use of satellite data for the systematic assessment of aci, i.e., S, adjustments and confounding meteorological factors. The study focuses on two areas, i.e. the Yangtze River Delta (YRD, 26°N-35°N; 113°E-122°E) in eastern China and the East China Sea (ECS, 19°N-28°N; 125°E-134°E). The locations of the YRD and the ECS are shown in the map in Figure 1.

The YRD has a developed economy, with much industrial activity, large harbors (sea and river) and

205 related busy ship traffic, dense population in large urban centers, all with high traffic intensity and high

energy consumption. In addition to the direct emission of black carbon, also aerosol precursor gases such

as $NO_2$, $SO_2$ and VOCs are emitted from the combustion of biomass, coal and petrochemical fuels,

leading to the formation of secondary aerosol particles such as nitrate and sulfate aerosols, while

agricultural activities result in the emission of dust, ammonia and biological VOCs (BVOCs) into the

210 atmosphere. These activities and associated emissions result in the occurrence of high AOD over the

YRD. Over the East China Sea (ECS) the main aerosol types are sea spray aerosol generated by the

interaction between wind and waves and anthropogenic pollutants transported from the Asian continent

over the ocean in the East Asian outflow. During transport over hundreds of km, aerosol particles are

removed by several processes such as dry and wet deposition and hence the aerosol concentrations

decrease and the AOD becomes relatively low and is dominated by sea spray aerosol. In view of the

differences in aerosol composition and concentrations, the polluted YRD area and the relatively clean

ECS area were selected as contrasting regions for the study of the influence of aerosols on cloud

properties over land and over ocean.

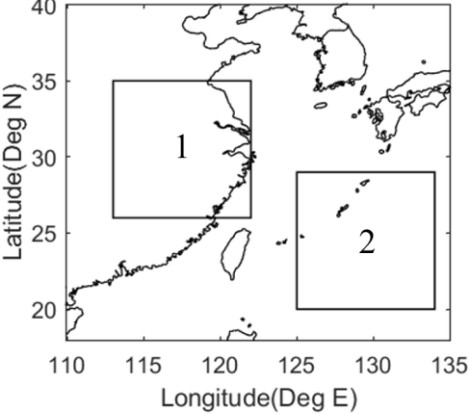

**Figure 1. Map showing the locations of the two study areas selected for aerosol - cloud interaction studies: area 1 is the Yangtze River Delta (YRD; 26°N-35°N, 113°E-122°E), and area 2 indicates the selected Eastern China Sea area (ECS; 20°N-29°N, 125°E-134°E).**

**2.2 Data used**

In this study, aerosol and cloud properties were used which were derived from measurements from the

225 Moderate Resolution Imaging Spectroradiometer (MODIS) on-board the Aqua satellite, for the period

2008-2022 (15 years). This data was selected because the MODIS data are widely used and therefore

they are well-characterized. In addition, the Aqua satellite flies in an afternoon orbit with local overpass

time around 13:30, when the atmospheric boundary layer is well-developed. MODIS L3 Collection 6.1 daily aerosol and cloud parameters were downloaded from the LAADS website (Liu, 2022a) with a spatial resolution of 1°x1°. Aerosol retrieval is only executed in clear sky conditions whereas cloud properties can only be retrieved in cloudy skies. Hence, it is not possible to obtain co-located aerosol and cloud data from satellite. For satellite-based aci studies it is assumed that, following, e.g., Jia et al. (2022), aerosol properties are homogeneous enough to be representative for those in adjacent cloud areas. Consequences of this assumption were discussed by McComiskey and Feingold (2012). The MODIS instrument has 36 spectral bands - aerosol properties are retrieved using the first seven of these (0.47-2.13 μm) (Remer et al., 2005; Levy et al., 2013; Sayer et al., 2014; 2017) while additional wavelengths in other parts of the spectrum are used for the retrieval of cloud properties (Platnick et al., 2003; 2017). Detailed information on algorithms for the retrieval of aerosol and cloud properties is provided at http://modis-atmos.gsfc.nasa.gov (last access: 01 July 2021). In this study we use the AOD at 550 nm (referred to as AOD throughout this manuscript), CER, COT, cloud liquid water path (LWP), cloud top pressure (CTP), cloud fraction (CF) and cloud top temperature (CTT). The MODIS Collection 6.1 AOD product over China has been validated by, e.g., Che et al. (2019) and globally over land and ocean by Wei et al. (2019). MODIS C6.1 cloud products were evaluated by Platnick et al. (2017). The validation of CER and LWP, the primary cloud products used in this paper, was described by Painemal and Zuidema (2011), who compared MODIS C5 with in situ data (aircraft), and likewise the MODIS C6.1 CER product was evaluated by Fu et al. (2022) by comparison with airborne measurements. Fu et al. (2022) concluded that their "validation, along with in situ validation of MODIS CER from other regions (e.g., Painemal and Zuidema, 2011; Ahn et al., 2018), provides additional confidence in the global distribution of bias-adjusted MODIS CER reported in Fu et al. (2019)." It is noted that COT and CER are retrieved whereas LWP is secondarily derived (e.g., Painemal and Zuidema, 2011). AOD is used as a proxy for the amount of CCN in the atmospheric column to investigate aci (Andreae, 2009) which seems to be the best alternative (Rosenfeld et al., 2014). As discussed in the Introduction, the use of an AE-based correction is not recommended over land (e.g., Kourtidis, et al., 2015). Comparisons with surface-based sun photometer data shows that Collection 6 improves upon Collection 5, and overall, 69.4% of MODIS Collection 6 AOD fall within the expected uncertainty of $\pm (0.05 + 15\%)$ (Levy et al., 2013; Tan et al., 2017). To reduce a possible overestimation of the AOD (e.g., due to cloud contamination), cases with

AOD greater than 1.5 were excluded from further analysis. The choice of this threshold is based on reports by Christensen et al. (2017) and Varnái and Marshak, (2009), rather than 0.6 used by Brendan et al. (2005), who used MOD06 Collection 04 products. Christensen et al. (2017) used MOD06 C6 data (1km x1km) and reported that "large aerosol optical depths remain in the MODIS-observed pixels near cloud edges, due primarily to 3-D effects (Varnái and Marshak, 2009) and the swelling of aerosols by higher relative humidity." Varnái and Marshak (2009) noted that beyond 15 km contamination effects were minimized in MODIS data (1km x1km). Furthermore, we discarded scenes (1° by 1°) in which the aerosol distribution is heterogeneous, i.e. with a standard deviation higher than the mean value (Saponaro et al., 2017; Jia et al., 2022). As most aerosol particles are located in the lower troposphere (Michibata et al., 2014), to avoid deep convective clouds, the focus in this study is on warm clouds with CTT larger than 273K and CTP larger than 700 hPa, while LWP larger than 200 g m$^{-2}$ is excluded (Wang et al., 2014). . Transparent-cloudy pixels (COT<5) were discarded to limit uncertainties (Grosvenor et al., 2018). The solar zenith angle was restricted to SZA < 65° and the viewing zenith angle to VZA <55° to avoid the large biases in COT and CER retrievals at larger angles (Grosvenor et al., 2018). To ensure that the data used only included single layer liquid clouds and nonprecipitating cases, the filtering criteria described by Saponaro et al. (2017) were applied.

Confounding meteorological effects on aci were explored using the daily temperature at the 700 and 1000 hPa levels, RH at the 750 hPa level and PVV at the 750 hPa level. Low tropospheric stability (LTS), which is defined as the difference in potential temperature between the free troposphere (700 hPa) and the surface (1000 hPa), is used as a measure of the strength of the inversion that caps the planetary boundary layer (Klein and Hartmann, 1993; Wood and Bretherton, 2006). These meteorological data were retrieved from the ECMWF ERA-5 reanalysis data which provide global meteorological conditions at 0.25°x0.25° resolution for 37 pressure levels in the vertical (1000-1 hPa), for every 1 h (UTC). The meteorological parameters were resampled to the MODIS/Aqua overpass time at 13:30 (local time) by taking a weighted average at the two closest times (05:00 UTC and 06:00 UTC) provided by the ECMWF ERA-5 reanalysis data.

**Table 1. Parameters used in the present study, together with the sources, time periods and spatial resolutions.**

| Source | Time period | Resolution | Parameters |
|--------|-------------|------------|------------|
| MYD08 | Jan 2008-Dec 2022 | Daily, 1°x1° | AOD at 550 nm |
| | | | COT at 2.1 um |

| | | | CER at 3.7 um and 2.1 um |
| --- | --- | --- | --- |
| | | | Cloud-top temperature |
| | | | Cloud-top pressure |
| | | | LWP at 2.1 um |
| | | | Cloud Fraction |
| | | | Solar zenith angle |
| | | | Sensor zenith angle |
| | | | Cloud multi-layer flag |
| | | | Cloud phase flag |
| ERA5 | Jan 2008-Dec 2022 | hourly, 0.25°x0.25° | Temperatures at 700 and 1000 hPa |
| | | | Relative humidity at 750 hPa |
| | | | Vertical velocity at 750 hPa |

## 3 Methods

### 3.1 Sensitivity of cloud parameters to changes in aerosol concentrations

Changes in aerosol loading lead to an adjustment of cloud optical or microphysical parameters (COT, CER, etc.). Aerosol particles can become CCN or INP, depending on their chemical composition and ambient temperature. When these nuclei are activated, they become cloud droplets due to condensation of water vapor. When the concentration of aerosol particles increases, often also the number of CCN or INP may increase and thus the number of cloud droplets may increase. However, if the liquid water content in the cloud does not change (as indicated by a constant LWP), the condensable water will be distributed over more cloud droplets which thus remain smaller, i.e. the CER decreases and the cloud albedo increases when the aerosol concentration increases. On the basis of findings of Kaufman and Fraser (1997), Feingold et al. (2001) pointed out that the sensitivity of cloud microphysical properties (e.g. CER) to changes in aerosol (e.g., AOD) can be described by the following formula:

$$S=S_{CER-A}= \frac{d \ln r_e}{d \ln \alpha}|_{LWP} \quad 0 < S < -0.33 \quad (1)$$

Where $r_e$ represents the CER and $\alpha$ represents the AOD. Following Andreae (2009), AOD and CCN are correlated and AOD varies with CCN following a power law relationship. Eq. (1) describes the relative change of CER with the relative change of the AOD for constant LWP. It is noted that this formulation differs from that used in recent studies (e.g., Bellouin et al., 2020) where S is expressed in $N_d$ with no restriction in LWP. The sensitivity S of CER to AOD can be determined as the slope of a linear fit to a log-log plot of CER versus AOD. It is noted that S is a function of CER and effects on CER

directly influence S. In this study effects on S and CER are used interchangeably. Relations between CER and AOD are determined through Eq. 1 and correlation coefficients R. The significance of these relations is determined by using the student's t test, i.e. the results are statistically significant when the p value is smaller than 0.01, where p is defined as the probability of obtaining a result equal to or "more extreme" than what was actually observed.

## 3.2 Geographical detector method

The geographical detector method (GDM) is introduced to analyze which factors influence the aci and identify possible correlations between different factors. The GDM is based on the assumption that if an independent variable has an important influence on its dependent counterpart, their spatial distributions should also have evident similarities (Wang and Hu, 2012; Wang et al., 2016). The GDM not only accounts for the rank order of the variables as determined by the Spearman's Rank method but also spatial information. The geographical detector provides four modules, including factor detector, interaction detector, risk detector and ecological detector. In this study, the first two modules are used to detect interactions between different parameters, based on their spatial variations, and thus reveal the driving factors for aerosol-cloud interaction over the target regions. The influencing factors ($x$) considered in this study are aerosol and meteorological parameters and the dependent factors ($y$) are S and cloud parameters. In the GDM, for example, the CER data are recorded in a raster grid as illustrated in Figure 2. The data in the raster grid is transformed into 2D point vector files, each point containing a value for the CER and for one of the influencing parameters $x$. The dependent (CER) and influencing ($x$) parameters are separated into 2 layers with the same grid. In the $x$ layer, the Jenks natural breaks classification method (Brewer and Pickle, 2002), aiming to minimize the variance within the group and maximize the variance between groups, was applied to categorize the whole region into $i$ sub-regions (3 in Figure 2), according to pre-defined ranges of influencing factors (e.g., AOD). In each sub-region, the influencing factor ($x$) varies within certain limits, with variance $\sigma_i$. The power of determination $q$ of $x$ to $y$ (also referred to as power of the influencing factor) determines the extent to which a factor ($x$) influences the dependent factor ($y$) over the whole study area and is calculated using Eq. (2):

$$q = 1 - \frac{\sum_i^L N_i \sigma_i^2}{N \sigma^2} \qquad (2)$$

where i (1, …, L) is the number of subregions of factor $x$; $N$ represents the total number of spatial units

over the entire study area; $N_i$ denotes the number of samples in sub-region i; and $\sigma_i^2$ and $\sigma^2$ denote the variance of the samples in the subregion i and the total variance in the entire study area, respectively. The value of $q$ varies between 0 and 1, i.e. $q[0,1]$, where 0 indicates that factor $x$ has no influence on y and the closer $q$ is to 1, the greater the influence of $x$. For instance, if $q = 0.5$, $x$ can explain 50% of the variation of $y$. In this study, multi-years of mean values of influencing factors ($x$) and dependent factors ($y$) were calculated for each raster grid. Then, we classified the influencing factors (e.g. AOD and meteorological parameters) into 5 sub-regions by the Jenks natural breaks classification method (Brewer and Pickle, 2002). For example, AOD needs to be classified into 5 levels using the Jenks natural breaks classification method, and the AOD source data needs to be reclassified into 1-5 natural numbers from small to large, and then counted into the grid. Therefore, the input of the independent variable AOD is a type variable. However, it should be noted that the GDM also has unstable characteristics. On the one hand, it is due to the MAUP (Modified Area Unit Problem) variable area unit problem, which can be understood as the influence of "scale effect". Due to the limitation of data resolution used in this study, the spatial statistical unit is 1°x1°. On the other hand, the methods used for data discretization can also have an impact. This study attempts to determine the optimal number of classifications by examining the impact of number of classification levels (3-8) on the GDM output results. The results show that the number of classification levels does not affect the relative importance of cloud factors on the cloud. Here we classify the values of each cloud factor into 5 levels during the period of 2008-2022.

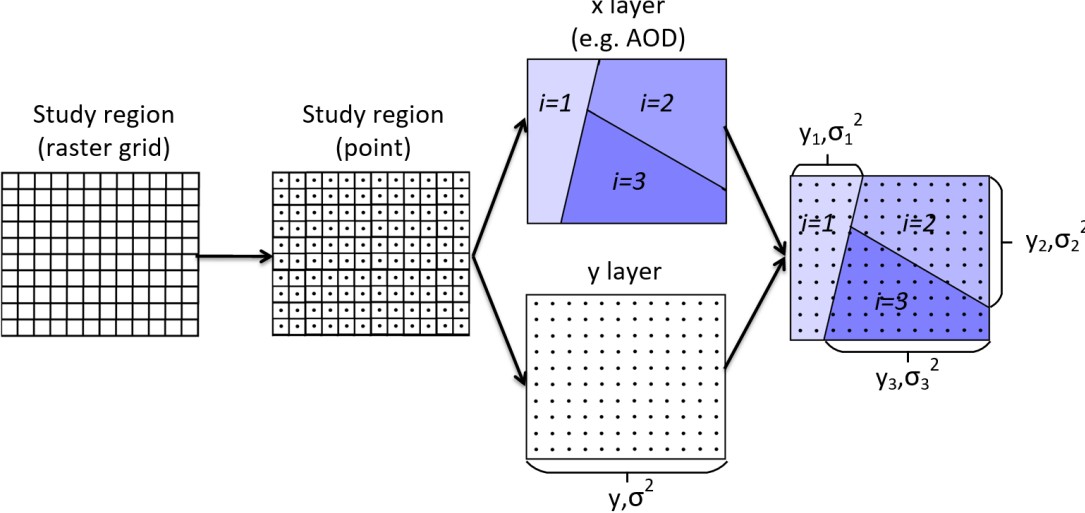

**Figure 2. The principle of the geographical detector method. See text for explanation.**

The interaction detector can be used to test for the influence of interaction between different influencing factors, e.g., $x1$ and $x2$, on the dependent factor ($y$) and whether this interaction weakens or enhances the

influence of each of $x1$ or $x2$ on the dependent variable, $y$, or whether they are independent in influencing $y$. For example, Figure 3(a) shows the spatial distribution of the dependent variable, $y$. The factors $x1$ and $x2$ both vary across the study region, but in different ways, and for each factor different sub-regions can be distinguished by application of the Jenks classification method described above to each factor separately. This is illustrated in Figures 3(b) and 3(c) where, as an example, three different sub-regions are considered for each factor. Usually, the dependent variable $y$ is influenced by several different factors $xi$ (Figure 3) and the combined effect of two or more factors may have a weaker or stronger influence on y than each of the individual factors. The $q$ values for the influences of factors $x1$ and $x2$ on $y$, obtained from the application of the factor detector method (Eq. 2), may be represented as $q(x1)$ and $q(x2)$. Hence, a new spatial unit and subregions may be generated by overlaying the factor strata $x1$ and $x2$, written as $x1{\cap}x2$, where $\cap$ denotes the interaction between factor strata $x1$ and $x2$ as illustrated in Figure 3(d). Thus, the $q$ value of the interaction of $x1{\cap}x2$ may be obtained, represented as $q(x1{\cap}x2)$. Comparing the $q$ value of the interaction of the pair of factors and the $q$ value of each of the two individual factors, five categories of the interaction factor relationship can be considered which are summarized in Table 2. If $q(x1{\cap}x2) > q(x1) + q(x2)$, this is referred to as a nonlinear enhancement of two variables. And if $q(x1{\cap}x2) > $ Max$[q(x1), q(x2)]$, this is referred to as a bilinear enhancement of two variables. The occurrence of nonlinear enhancement and bilinear enhancement are indicated with the $q$ values in Table 2 and in the caption of Figure 7.

It is noted that the $q$-values of multiple influencing factors are considered separately they may sum up to larger than 100%. However, when the variables are correlated they must be considered together and the interaction $q$-value must be evaluated.

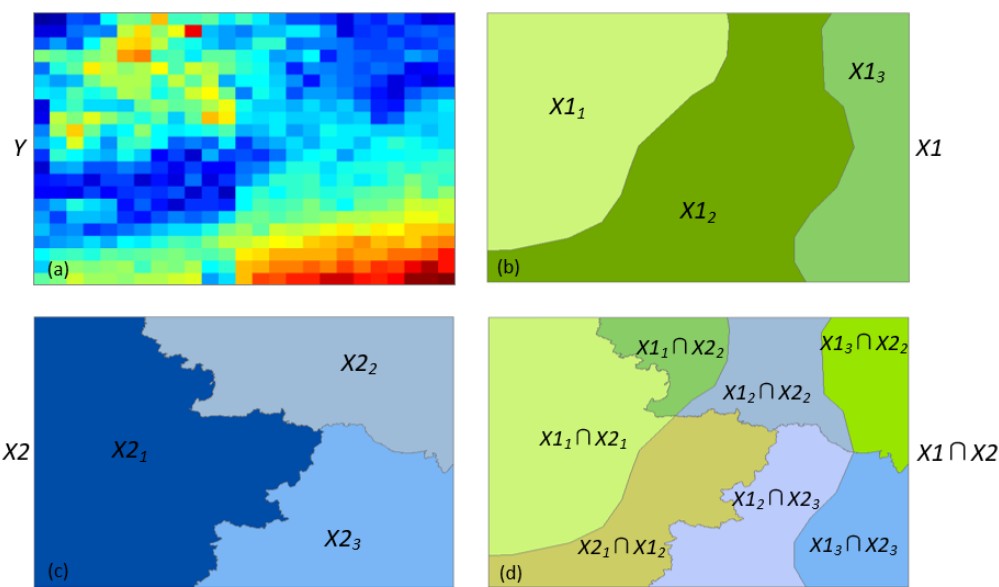

**Figure 3. Detection of interaction (see text for explanation).**

**Table 2. Interaction categories of two factors and the interaction relationship.**

| Illustration | Description | Interaction |
|---|---|---|
| | $q(x1 \cap x2) < \mathrm{Min}[q(x1), q(x2)]$ | Weakened, nonlinear |
| | $\mathrm{Min}[q(x1), q(x2)] < q(x1 \cap x2) < \mathrm{Max}[q(x1), q(x2)]$ | Weakened, unique |
| | $q(x1 \cap x2) > \mathrm{Max}[q(x1), q(x2)]$ | Enhanced, bilinear |
| | $q(x1 \cap x2) = q(x1) + q(x2)$ | Independent |
| | $q(x1 \cap x2) > q(x1) + q(x2)$ | Enhanced, nonlinear |

The geographical detector method has been used to detect influencing factors for several different

purposes (e.g., Wang et al., 2018; Zhang and Zhao, 2018; Zhou et al., 2018). For example, the GDM was

used to detect the influence of annual and seasonal factors on the spatial-temporal characteristics of

surface water quality (Wang et al., 2018). Other examples are the application of the GDM to examine

factors influencing regional energy-related carbon emissions (Zhang and Zhao, 2018) and to examine

effects of socioeconomic development on fine particulate matter (PM2.5) in China (Zhou et al., 2018).

In the current study, the GDM was used to detect the impact of nine variables and their interactions on S

and cloud parameters over land and ocean. The advantages of using the GDM in this approach are (1)

stratified independent variables enhance the representation of a sample unit, so it has higher statistical

accuracy than other models with the same sample size; (2) the use of a q-statistic value can afford a

higher level of explanatory power, but does not require the existence of a linear relationship between

independent and dependent variables; (3) the GDM can determine the true interaction between two variables and is not limited to pre-established multiplicative interactions (Wang et al., 2010); (4) the use of the GDM does not need to consider the collinearity of multiple independent variables (Wang et al., 2010).

**4 Results**

**4.1 Spatial distribution and correlation analysis of AOD and cloud parameters**

The spatial variations of the AOD and the cloud properties (CER, COT, CF, CTP and LWP) over the study area, averaged over the years 2008-2022, are presented in Fig. 4. Figure 4(a) shows a large difference between the AOD over land and ocean, with the highest values over the northern part of the YRD (averaged AOD larger than 0.5), and the lowest values over the southeastern part of the ECS (<0.1); the AOD decreases gradually from land to ocean. The spatial distributions of the CER, COT, CF, CTP and LWP over the YRD and ECS in Figs. 4(b)-(f) shows that for each of them there is a distinct difference between those over land and over ocean both as regards the values and the spatial variation. Over the ECS, the CER is largest in the south and decreases toward the north of this area and the values are overall substantially larger than over the YRD, where the CER varies somewhat and decreases from north to south. The variation of the CER with AOD over the YRD is opposite to what would be expected, which will be discussed in Sect. 4.2. The COT also varies somewhat over the YRD, but contrary to the CER, COT increases from north to south. Over the ECS, the COT is generally lower than over the YRD, with the highest values in the northwest which gradually decrease toward the southeast. Clearly, the CER is higher and the COT is lower over the ECS than over the YRD.

The spatial distributions of CF, CTP and LWP are clearly different. Over the ECS, CF increases from the southeast to the northwest, opposite to the variations of the CTP and the LWP which are lower in the north of the ECS than in the south. Over ocean the clouds are generally lower (higher CTP) than over land, and CTP varies over the study area with the highest values over land, in the north. Over the YRD, the spatial patterns of the CF and CTP are opposite, with CF increasing from south to north and CTP decreasing. Over the YRD, the spatial distributions of COT and LWP are similar with higher values toward the south. Over the ECS, the LWP varies with the lowest values in the northwest and the highest values in the south. The high values of the CER over the ECS could be due to the dominance of sea spray

aerosol, the high hygroscopicity of which makes these particles very efficient CCN, which in this

environment over ocean with high water vapor concentrations, results in larger CER. The influence of

different factors on the sensitivity of cloud parameters to aerosol and the adjustments are discussed in

the following sections, based on both statistical methods and the application of the GDM.

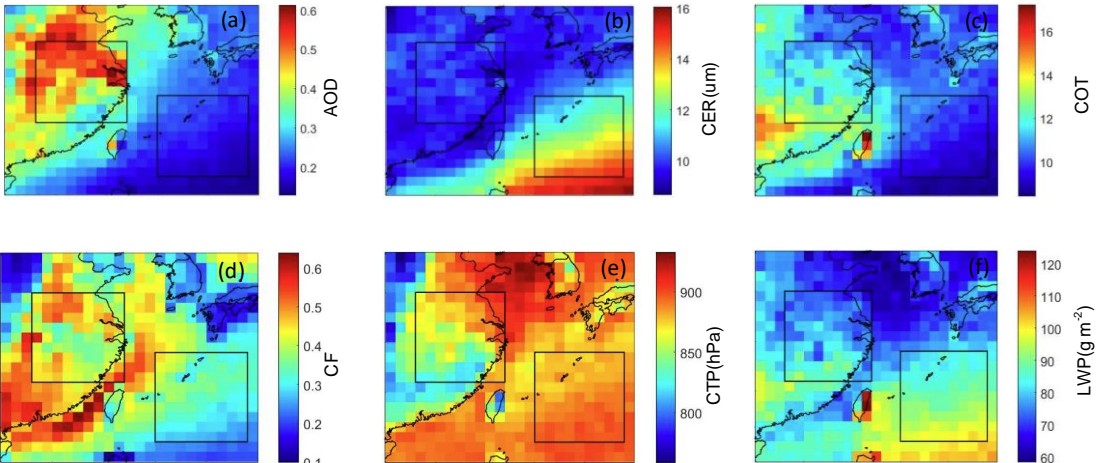

**Figure 4. Spatial distributions of AOD (a), CER (b), COT (c), CF (d), CTP (e) and LWP (f), averaged over**
425 **the years 2008 – 2022, over the study area, with the YRD and ECS marked by the squares.**

## 4.2 Sensitivity of CER to AOD

Eq. (1) shows that the value of the sensitivity S of CER to AOD is determined by the slope of a linear fit

to a log-log plot of CER versus AOD. To investigate S, we used correlated data pairs for 15 years and

the data was binned in AOD intervals with a bin width of 0.02, and the CER data in each AOD bin were

430 averaged. Logarithmic plots of the averaged CER data versus AOD over the YRD and the ECS are

presented in Figure 5. Figures 5(a) and 5(b) show different regimes for the variation of the CER with the

AOD over the YRD and the ECS. The first regime, for AOD ≤ 0.05, shows the increase of CER with

AOD over both regions, followed by a variable CER over the YRD and a gradually stronger decrease

over the ECS for AOD between 0.05 and 0.1. In view of this variability and the uncertainty of AOD of

435 ± (0.05 + 15 %) over land and ± (0.03 + 5 %) over ocean (Levy et al., 2013), S will not be investigated

for AOD < 0.1. For higher AOD, S changes for AOD around 0.3. Thus, the second regime is selected as

the part of the CER vs AOD relationship where AOD varies between 0.1 and 0.3. In this AOD regime,

the CER fluctuates a little with AOD over the YRD (Figure 5(a)) and S is close to 0 (no discernible

Twomey effect). In contrast, over the ECS the CER clearly decreases with AOD for AOD increasing

from 0.1 to 0.3 (Figure 5(b)), in good agreement with expectation based on the Twomey effect, and the

correlation between CER and AOD is high with R=0.99 and statistically significant. Note however, that no selection was made for LWP and the condition of constant LWP was not fulfilled. This will be further discussed in Section 4.3.

In the third regime, where AOD > 0.3, CER increases with increasing AOD over the YRD, with correlation coefficient R= 0.79. In contrast, over the ECS the CER does not significantly change with increasing AOD for AOD>0.3 (very small S). However, the large uncertainty in the bin-averaged CER in this AOD regime, increasing with increasing AOD, indicates a very variable S between high-AOD events which on a statistical basis cannot be further analysed and likely depends on the type of aerosol present during each event and the meteorological conditions. The reason for the increase of CER with increasing AOD (S positive) over the YRD may be similar to that described by Feingold et al. (2001), i.e., in the presence of a large number of aerosol particles (CCN) competing for a limited amount of water vapor, only a subset of aerosol particles is activated. Once activated, these particles continue to grow faster, thus preventing water vapor from condensing onto smaller aerosol particles that are less susceptible to activation. As a result, the amount of available water vapor is distributed over a subset of aerosol particles which thus become cloud droplets with relatively large CER and the CER in turn increases with further increasing AOD (Liu et al., 2017).

The CER sensitivity to AOD is stronger over the ECS (0.1<AOD<0.3) than over the YRD (AOD>0.3). It is anticipated that during the relatively low AOD over the ECS in AOD regime 2 (0.1<AOD<0.3) the aerosol number concentration is dominated by sea spray aerosol particles (de Leeuw et al., 2011) which are hygroscopic and thus provide good CCN, while over open ocean also the RH is generally high. Hence the available water vapor will be readily distributed over all CCN, resulting in the decrease of the CER and a strong correlation with AOD. Further, the AOD over open ocean does not reach high values in the absence of continental influence, even in very high wind speeds the AOD does not exceed 0.2 (Huang et al., 2010; Smirnov et al., 2012). Hence AOD higher than 0.2 over the ECS is influenced by long-range transport of aerosol produced over land with lower hygroscopicity, and thus lower susceptibility to act as CCN, which explains the breakdown of the Twomey effect over the ECS for elevated AOD. In fact, the data in Figure 5(b) show that the CER-AOD relationship starts to flatten for AOD ~0.2 and is flat for AOD larger than ~0.3. Overall, Figure 5 shows that the Twomey effect is clear in the second AOD regime over the ECS and the anti-Twomey effect in the third AOD regime over the YRD. For this reason, the

further analysis focuses on the aci over the ECS for AOD between 0.1 to 0.3, and over the YRD for

AOD > 0.3.

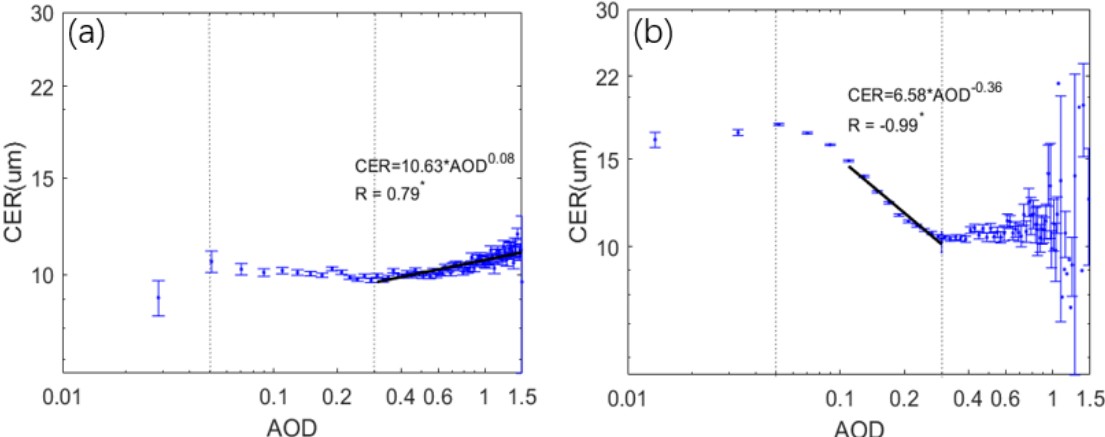

**Figure 5. Variation of CER with AOD over the YRD (a) and the ECS (b). Here all CER data were averaged in AOD bins, from 0.0 to 1.5 with a step of 0.02. Note that the data are plotted on a log-log scale. The lines for**
**the YRD data for AOD>0.3 and for the ECS data for 0.1<AOD<0.3 represent least-square fits to the binned data, and the resulting relations are presented in each figure. The marker ∗ at the top right corner of the R value indicates that the correlation is statistically significant with p < 0.01. The thin vertical lines indicate the AOD regimes as explained in the text.**

To study the spatial variation of S over the study area, S has been calculated in each grid cell by
application of Eq. (1) to all observations over the YRD for which AOD>0.3 and to all observations over

the ECS for which 0.1<AOD<0.3. The results are plotted in Figure 6, which shows maps of S, the

correlation coefficient R between CER and AOD and the statistical P-value for each grid cell over the

study area. Figure 6(b) shows that over the ECS, for the second AOD regime (0.1-0.3), S is negative,

with large negative correlation coefficients (-0.66 to -0.98) which mostly are statistically significant (p <
0.01). These results show the good correlation between CER and AOD, consistent with the cloud albedo

effect. In contrast, over the YRD, for the third AOD regime (>0.3), S is mostly positive and the correlation

between CER and AOD is positive, i.e. high aerosol loading results in larger CER for AOD>0.3, as was

also concluded from Figures 5. The data in Figure 6(a) also show that, over the YRD, S is largest over

the area to the north of Shanghai but R is relatively weak (0.11 to 0.35) and for the majority of the cells
the correlations are not statistically significant (p ~ 0.1 or larger). South of Shanghai the correlations are

small and not statistically significant. The observed anti-Twomey effect of aerosols over the YRD has

also been reported in earlier publications such as Jin and Shepherd (2008), Yuan et al. (2008) and Liu et

al. (2017). Factors influencing the relationship between AOD and cloud parameters have been reported

in the literature, such as hygroscopic effects (e.g., Qiu et al., 2017), atmospheric stability, cloud dynamics,

cloud height (Shao and Liu, 2005) and land cover type (Jin and Shepherd, 2008; Ten Hoeve et al., 2011).

The effects of competing mechanisms and their possible influence on the observed response of CER to

high AOD in the YRD will be further discussed in the following sections.

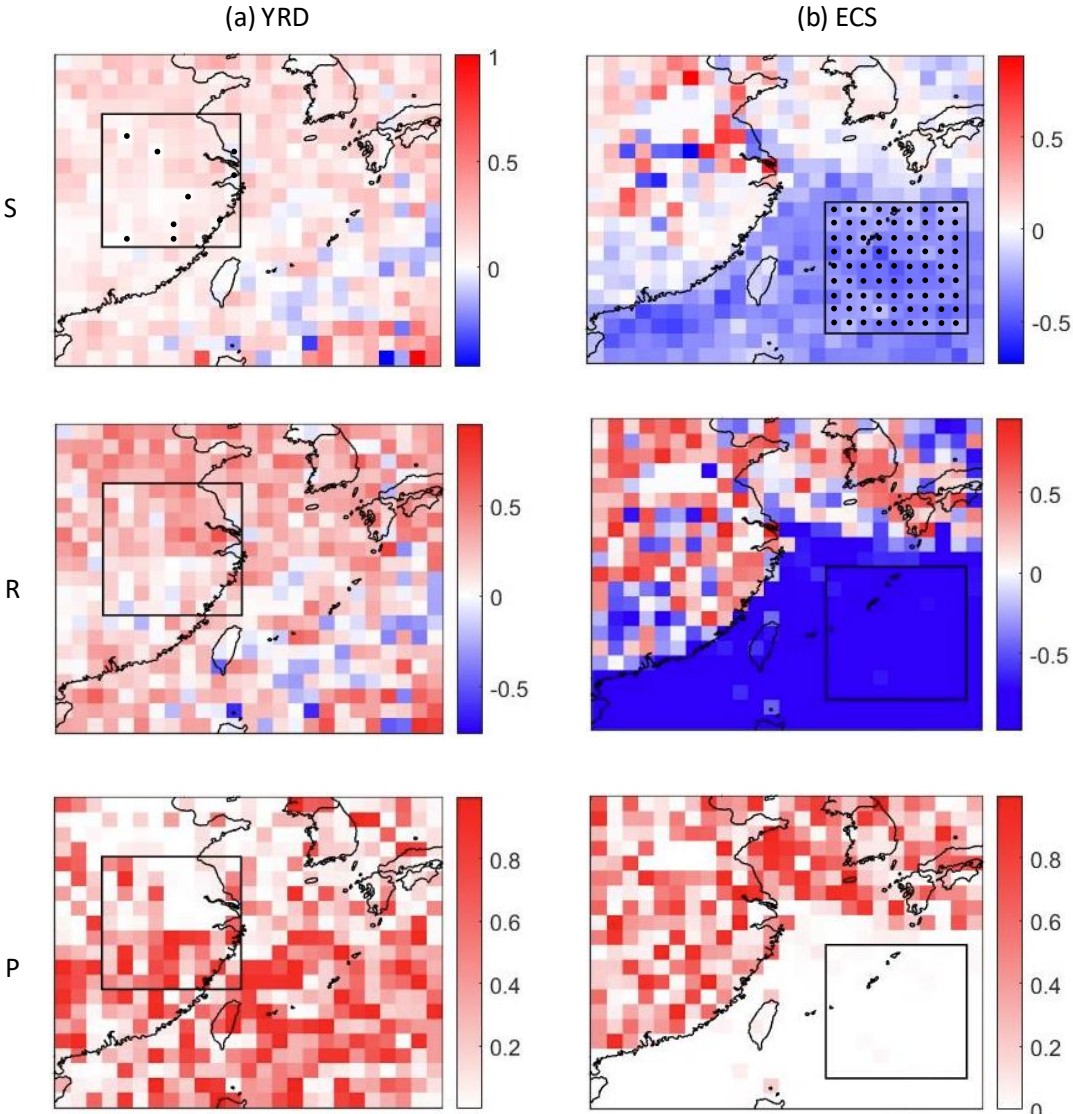

**Figure 6. Using the AOD as a proxy for CCN, estimates of the CER sensitivity to aerosol (S) were calculated**
**for each grid point in both study areas. Maps of the spatial distributions of S, the correlation coefficients and**
**the statistical P-values in each grid point are presented over the YRD (left column, Figure 6(a)) for the AOD**
**regime with AOD>0.3 and over the ECS (right column, Figure 6(b)) for the AOD regime with 0.1<AOD<0.3.**
**S, R and P-values are color coded following the color bars at the right of each figure. The black solid dots in**
**the top figures (S), indicate that the S value is negative in the grid point over the YRD and ECS.**

**4.3 Sensitivity of CER to AOD stratified by LWP**

In the data presentation and discussion of S in Section 4.2, the condition of constant LWP for the

application of Eq. (1) and the occurrence of the cloud albedo effect, was not considered. In this Section

the effect of LWP on S will be further investigated. To this end, the condition of constant LWP is

approached by stratifying LWP into five intervals, each with a width of 40 g m$^{-2}$, for the LWP range of

[0 g m$^{-2}$, 200 g m$^{-2}$]. S was calculated over the YRD and the ECS, for each LWP interval using Eq. (1)

for all observations over the YRD for which AOD>0.3 and for all observations over the ECS for which

0.1<AOD<0.3. The results are presented in Table 3, together with the corresponding correlation

coefficients R between CER and AOD in the relevant AOD regimes. The data in Table 3 show that over

the ECS, S is negative and statistically significant for all four LWP ranges between 40 and 200 g.m$^{-2}$.

The sensitivity becomes stronger as LWP increases, i.e., S changes from -0.19 (LWP 40-80 g.m$^{-2}$) to -

0.46 in the highest LWP range (160-200 g.m$^{-3}$), with corresponding R of -0.98 to -0.99. Thus, the

magnitude of the LWP has a substantial influence on the albedo effect. Over the YRD, S is positive and

statistically significant in the first three LWP regimes, with values varying between 0.06 and 0.10 and a

correlation R between 0.57 and 0.81. These data show that, in contrast to the ECS, over the YRD the

variation of the LWP has little influence on S and thus the magnitude of the LWP has little influence on

the cloud    albedo effect.

In summary, the data show that both over the ECS and the YRD the relationships between the CER and

the AOD are significant, but for different LWP intervals ([0 g m$^{-2}$, 120 g m$^{-2}$] over the YRD and [40 g m$^{-2}$, 200 g m$^{-2}$] over the ECS) and for different AOD regimes (0.1<AOD<0.3 over the ECS and AOD>0.3

over the YRD), and that the CER-AOD relation follows the Twomey effect over the ECS and the anti-

Twomey effect over the YRD.

The variation of S with changes in LWP indicates that the condition of constant LWP is not truly satisfied:

if the data would be stratified according to smaller LWP intervals (quasi-constant LWP, Ma et al., 2018),

S would likely vary more smoothly with LWP. As mentioned in the Introduction, LWP is not directly

retrieved but calculated form CER and COT and thus also the calculation of S is to some extend affected

by LWP. We further note the results by Ma et al. (2018), i.e. the slope of CER versus AI (comparable to

S in this paper) varies little with LWP, with positive values over land and negative values over ocean and

thus behaves similar to the data in Table 3 for YRD and ECS.

In the following study on the effects of the AOD and different cloud and meteorological properties on S

and adjustments, these differences will be taken into account, i.e. over the YRD only data with AOD >

0.3 and LWP in the range from 0 to 120 g m$^{-2}$ will be used and over the ECS only data with AOD in the

interval [0.1, 0.3] and LWP in the range from 40 to 200 g m$^{-2}$ will be used.

**Table 3. Estimates of S, computed using Eq. (1), and correlation coefficients R between CER and AOD, stratified by LWP, over the ECS for 0.1<AOD<0.3 and over the YRD for AOD>0.3. Statistically significant data points are indicated with * (p value < 0.01).**

| LWP (g m$^{-2}$) | ECS (0.1<AOD<0.3) | | YRD (AOD>0.3) | |
|---|---|---|---|---|
| | S | R | S | R |
| 0-40 | 0.10 | 0.94* | 0.08 | 0.63* |
| 40-80 | -0.19 | -0.98* | 0.10 | 0.81* |
| 80-120 | -0.38 | -0.99* | 0.06 | 0.57* |
| 120-160 | -0.41 | -0.99* | -0.03 | -0.11 |
| 160-200 | -0.46 | -0.98* | -0.14 | -0.42* |

## 4.4 Behaviour of CER and other cloud properties with the increase of AOD

Scatterplots of CER versus other cloud properties (COT, CF and CTP), with AOD as third parameter (color-coded), over the ECS and the YRD, are presented in Figure 7. Over the ECS, the CER and CTP decrease (the cloud top height increases) with the increase of AOD, and the COT and CF increase. The increase of AOD indicates an increase of the aerosol concentration and thus potentially the number of CCN, which in turn, upon activation, results in the increase of the number of cloud droplets and thus an increase of the COT. The positive correlation between COT and AOD over the ECS suggests that the thicker clouds contain more water droplets and are formed in a more polluted atmosphere, which, as discussed in Section 4.2, results from the influence of long-range transport of aerosol produced over land on the aerosol burden over ocean. But at the same time, as Figure 7(a) shows, CER decreases with increasing AOD, resulting in the increase in cloud albedo and thus also in the increase of COT. The increase of cloud top height with AOD indicates that both the horizontal and vertical expansion of the clouds are also enhanced. These observations are in agreement with the strong correlation between aerosol loading and cloud vertical development for convective clouds over the North Atlantic reported by Koren et al. (2005). Although there is a strong correlation between AOD, CF and CTP, this does not imply evidence of an aerosol effect (Quaas et al., ACP, 2010; Gryspeerdt et al., ACP, 2014).

In contrast to the situation over the ECS, over the YRD the increase of AOD results in an increase of the CER and CTP (the cloud top height decreases), and a decrease of the COT. These observations are consistent with those proposed by Liu et al. (2017) in the same study region. The decrease of the CF with increasing AOD could be explained as follows. Due to the high concentration of smoke particles over the YRD (Shen et al., 2021), aerosol particles absorb solar radiation which results in local heating of the

aerosol layer and cooling of the surface (Li et al., 2017). This in turn stabilizes the temperature profile and reduces the relative humidity and surface moisture fluxes (evapotranspiration) (Koren et al., 2008) and thus also cloudiness. Reduced cloud cover exposes greater areas of the aerosol layer to direct

irradiation from the Sun and therefore produces more intense heating of the aerosol layer, further reducing cloudiness (Koren et al., 2008). It is noted that this process is different from that proposed by Liu et al. (2017), i.e. that the CF increases with increasing AOD in polluted and heavily polluted conditions (AOD>0.3). In the study of Liu et al. (2017), the LWP range was not constrained, i.e. aerosol-cloud interaction was studied considering the whole LWP range. The data presented in Table 3, shows

that S significantly changes between the three LWP intervals between 0 and 120 g m$^{-2}$ where S is positive (anti-Twomey effect) and for larger LWP it is negative but statistically not significant. Figure 8 shows that CER and CTP substantially increase, whereas COT and CF decrease with increasing AOD in the two LWP intervals between 40-120 g m$^{-2}$. However, in the other three LWP intervals the relationships between these cloud parameters and AOD are not evident. The different explanations offered here and in Liu et

el. (2017) may be related to the different aerosol and cloud data sets used by Liu et al. (2017) and in the current study. On the one hand, the data sets have a different spatial resolution and cover a different time period. The dataset used in the study of Liu et al. (2017) are MYD04 Level 2 Collection 5 and MYD06 Level 2 Collection 5 in the period from 2007 to 2010. During that period the AOD over the YRD was at a maximum and decreased substantially in later years (Liu et al, 2021; de Leeuw et al., 2022; 2023). On

the other hand, in the study of Liu et al. (2017), the MODIS-retrieved AOD was averaged over an area with a radius of 50 km from the CALIOP target and the MODIS-retrieved cloud data were averaged within a radius of 5 km from the CALIOP target. Hence the AOD and cloud parameters were not representative for the same area, in particular in cases with inhomogeneous spatial distributions.

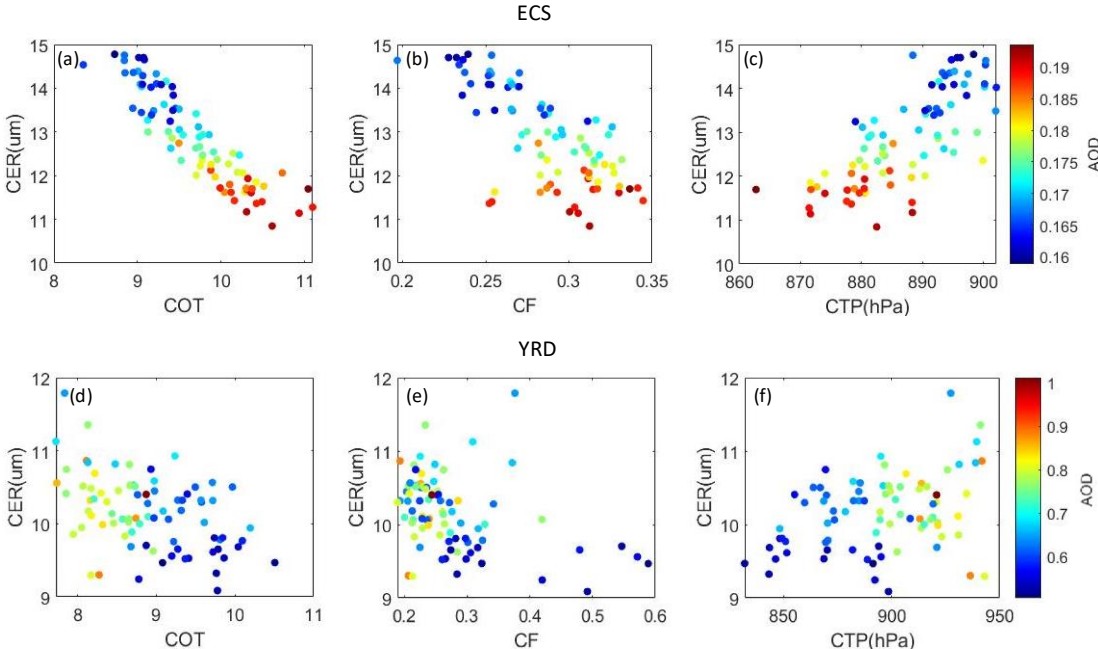

**Figure 7. Scatterplots of CER versus other cloud parameters (COT, CF and CTP; left to right) over the ECS (top row) and the YRD (bottom row), with AOD as third parameter, color coded following the scale at the right.**

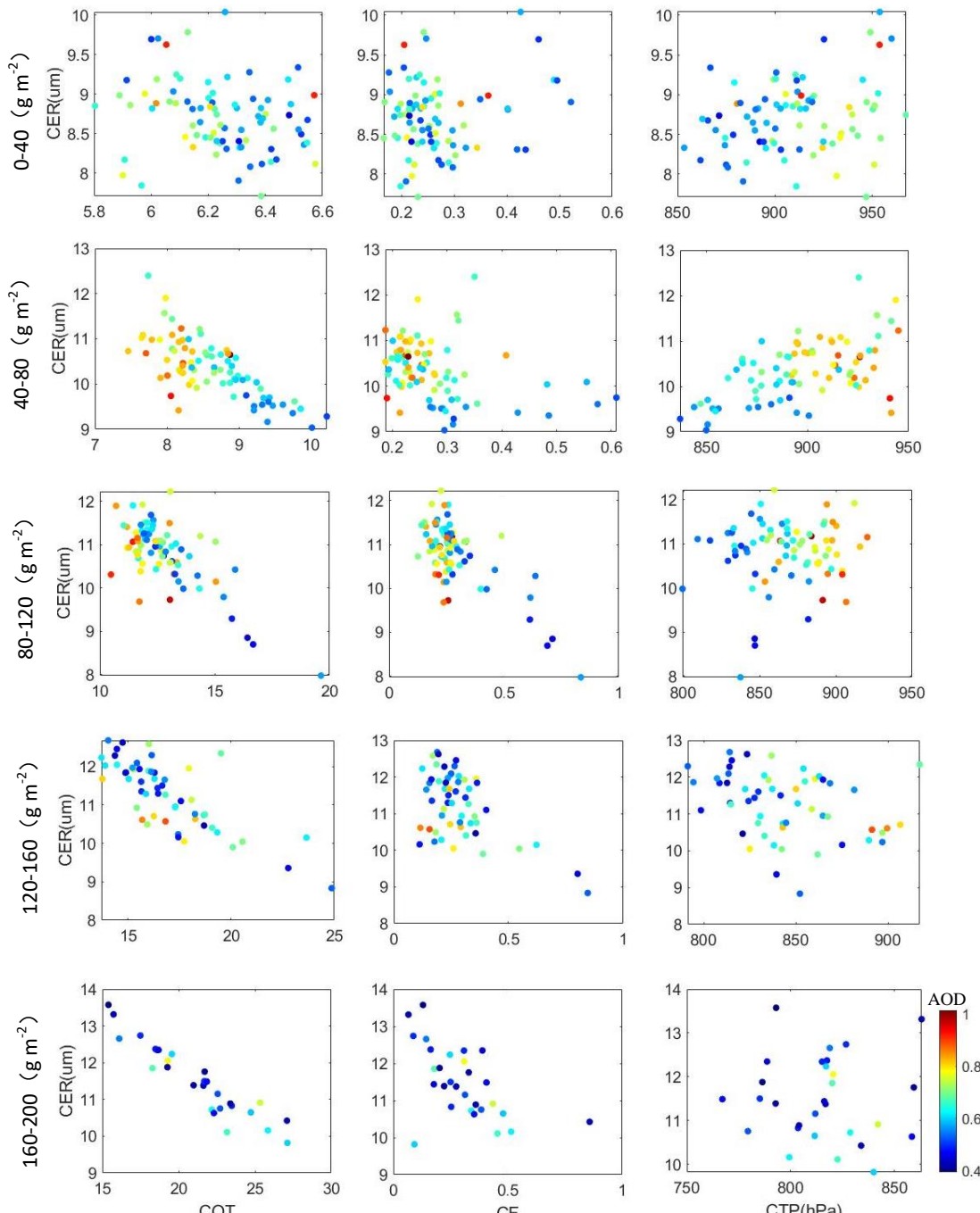

**Figure 8. Scatterplots of CER versus other cloud parameters (COT, CF and CTP; left to right) over the YRD, for five different LWP intervals between 0 and 200 gm$^{-2}$. The AOD for each grid point is color coded following the scale at the right.**

## 4.5 Behaviour of CER and AOD in different meteorological conditions

Scatterplots of the CER versus AOD over the ECS and the YRD, with meteorological factors (LTS, RH, PVV) (color coded) as third parameter, are presented in Figure 9. Over the ECS (Figure 9(a)), the AOD is inversely related to LTS, whereas the CER increases with increasing LTS. This observation is different

from the findings of Saponaro et al. (2017) who reported that there is no significant influence of atmospheric stability (LTS) on the relationship between CER and AOD. Likewise, the AOD is inversely related to RH whereas CER increases with increasing RH. These two observations indicate that RH and LTS have a similar effect on the relationship between AOD and CER. In contrast, with the increase of PVV, the AOD becomes larger but the CER becomes smaller. The CER vs AOD curves show that, overall, the meteorological conditions do not change the functional relationship between AOD and CER, but quantitatively they do have an effect. The change of meteorological conditions plays an important role in the variation of CER.

Different from the situation over the ECS, over the YRD the effect of meteorological conditions on the CER is weak as shown in Figures 9(d)-(f). RH and PVV have an inverse effect on the relationship between AOD and CER. There is no significant influence of atmospheric stability (LTS) on the relationship between CER and AOD as suggested by Saponaro et al. (2017). Overall, aerosol concentration plays a more important role in the effects of different factors on CER over the YRD.

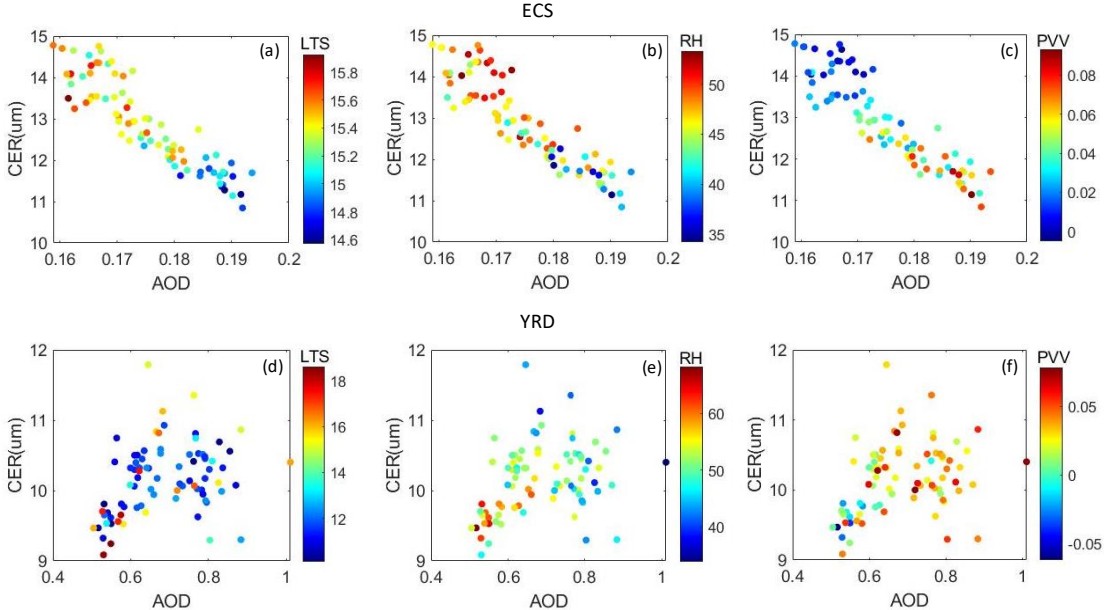

**Figure 9. Scatterplots of CER versus meteorological parameters (LTS, RH and PVV; left to right) over the CS (top row) and the YRD (bottom row). The AOD for each grid point is color coded following the scale at the right.**

## 4.6 Application of the geographical detector method

### 4.6.1 Factor detector analysis

The GDM factor detector module was used to analyze the influence of 9 factors (AOD, cloud and meteorological parameters) on S over the YRD and the ECS, for the conditions summarized at the end of Section 4.3. These factors are summarized in Table 4, together with q, i.e. the explanatory power of that factor to S (Eq. 2), over the ECS and the YRD. The data in Table 4 show that the influences of the 9 proxy variables on S are rather weak and not statistically significant. They can explain only 1%-15% of

the variation of S in both target regions.

**Table 4. q values for factors which may influence S over the ECS and the YRD, evaluated for data collected in the period from 2008-2022.**

| Study Area | Aerosol parameter | Cloud parameters | | | | | Meteorological parameters | | |
|---|---|---|---|---|---|---|---|---|---|
| | AOD | CER | COT | LWP | CF | CTP | RH | LTS | PVV |
| ECS | 0.07 | 0.06 | 0.06 | 0.10 | 0.01 | 0.13 | 0.10 | 0.11 | 0.09 |
| YRD | 0.05 | 0.09 | 0.06 | 0.05 | 0.04 | 0.06 | 0.15 | 0.09 | 0.09 |

Note: *** indicates that the q value is significant at the 0.01 level (p < 0.01).

The GDM factor detector module was also used to analyze the influence of the AOD and meteorological parameters (RH, LTS and PVV) on adjustments of cloud properties. The results in Table 5 show that AOD and PVV influence all cloud parameters over the ECS except CTP, with q-values which are statistically significant at the 1% level. The q-values for AOD show that this factor can explain 46% (for

CF) to 81% (for CER) of the variation in the cloud parameters considered in this study, and PVV can explain 47% (for CF) to 70% (for CER) of the variation in the cloud parameters. For LTS and RH, the q-values for CER are statistically significant but with smaller explanatory power than for AOD and PVV. In contrast, the q-value of LTS for LWP is statistically significant and not much smaller than for PVV.

**Table 5. q values for factors which may influence cloud parameters over the ECS, evaluated for data collected**

**in the period from 2008-2022.**

| Cloud parameters | AOD | RH | LTS | PVV |
|---|---|---|---|---|
| CER | 0.81[***] | 0.33[***] | 0.44[***] | 0.70[***] |
| COT | 0.69[***] | 0.40 | 0.38 | 0.67[***] |
| LWP | 0.68[***] | 0.23 | 0.43[***] | 0.49[***] |
| CF | 0.46[***] | 0.20 | 0.09 | 0.47[***] |
| CTP | 0.47 | 0.53 | 0.18 | 0.58 |

Note: ***indicates that the q value is significant at the 0.01 level (p < 0.01).

The results from a similar analysis of the data over the YRD (Table 6) show that AOD has a statistically significant influence at the 1% level on COT and CF, but with much smaller explanatory power than over the ECS. AOD can explain 31% of the variation of CER but the statistical significance is small ($p<0.1$).

Among the meteorological parameters, RH has a statistically significant influence on CTP and can explain 74% of the variation of the CTP and LTS can explain 55% of the variation of the LWP and 50% of the variation of the CF with $p<0.01$. The explanatory power for the effects of RH (32%) and PVV (18%) on LWP have low statistical significance ($p<0.1$).

**Table 6. q values for factors which may influence cloud parameters over the YRD, evaluated for data collected in the period from 2008-2022.**

| Cloud parameters | AOD | RH | LTS | PVV |
|---|---|---|---|---|
| CER | 0.31 | 0.25 | 0.13 | 0.18 |
| COT | 0.61*** | 0.45 | 0.12 | 0.29 |
| LWP | 0.16 | 0.32 | 0.55*** | 0.18 |
| CF | 0.30*** | 0.02 | 0.50*** | 0.07 |
| CTP | 0.50 | 0.74*** | 0.32 | 0.56 |

Note: ***indicates that the q value is significant at the 0.01 level ($p < 0.01$).

Tables 5 and 6 list q values for individual factors, together with p showing the absence of statistical significance in many cases, especially over the YRD, and often the explanatory power is not high when the significance is low. These data show that cloud parameters are dominated by aerosol effects over the ECS but meteorological influences on cloud parameters predominate over the YRD, as was also concluded from the analysis from "traditional" statistical methods presented in Section 4.5 and these conclusions are consistent with the results published by Andersen and Cermak (2015). Among the meteorological parameters, we also find that PVV (with highest q in the three meteorological parameters) predominately influences cloud parameters over the ECS. Jones et al. (2009) and Jia et al. (2022) reported that stronger aerosol cloud interactions typically occur under higher updraft velocity conditions. In addition, we find that CTP is mainly affected by RH (q = 0.74***) and PVV (q = 0.56) over the YRD, as suggested by Koren et al. (2010). Koren et al. reported that observed cloud top height correlates best with model pressure updraft velocity and relative humidity. To some extent, LTS influences CER (q = 0.44***) and LWP (q = 0.43***) over the ECS, while, in contrast, over the YRD LTS predominately influences CF (q = 0.50***) and LWP (q = 0.55***). Matsui et al. (2004) and Tan et al. (2017) reported that aerosol impact on CER is stronger in more dynamic environments that feature a lower LTS and argue that very high LTS environments dynamically suppress cloud droplet growth and reduce aci intensity.

While strong correlations between AOD and cloud parameters have been previously observed, they are likely due to the swelling of aerosol particles in humid airmasses (Quaas et al, 2010), rather than an aerosol influence, which is in agreement with findings by, e.g., Myhre et al. (2007), Twohy et al. (2009) and Quaas et al. (2010).

### 4.6.2 Interaction detector analysis

The q values of the combined effect of two parameters (AOD, RH, LTS, PVV) influencing the cloud parameters over the YRD and the ECS, derived using the GDM as described in Section 3.2, are presented in the matrix in Figure 10. The data in Figure 10 show that the q-values for the interaction of a pair of factors are larger than the q-values for any of the individual parameters (Table 5). Over the ECS, the combined effects all exhibit a bilinear enhancement over the time period of this analysis. The q values for the combined effects on CER over the ECS show that the explanatory power of AOD together with each of the three meteorological parameters, RH, LTS and PVV is high with 86%, 84% and 92%, respectively. Also for the combination of LTS and PVV the explanatory power is high (90%). Further inspection of the data in Figure 10 shows that the explanatory powers of the combined effects are high for several combinations of parameters, such as the combination of AOD with RH, LTS or PVV which all have high explanatory power for COT. The data in Figure 10 show that the combination of AOD and PVV results in high explanatory power for their influence on 4 cloud parameters (CER, COT, LWP and CF) and the combination of LTS with RH has high explanatory power for their effects on CTP. Among the meteorological parameters, we find that the combined effect of AOD and PVV predominantely influences cloud parameters over the ECS. The result is in accord with the findings of Jones et al. (2009) and Jia et al. (2022) that stronger aerosol cloud interactions typically occur under higher updraft velocity conditions.

Over the YRD, half of the q values for the combined effects on cloud properties exhibit nonlinear enhancement over the time period of this analysis, indicating that the combined effects on cloud properties are much larger than that over the ECS. The data in Figure 10 show that the combination of AOD and RH results in high explanatory power for their influence on CER and COT, and the combination of AOD with LTS has high explanatory power for their effects on LWP and CTP. The combined effects of PVV and LTS on the CF result in the highest explanatory power of 0.84. The data in Fig. 10 also show

that cloud parameters are more sensitive to the combination of AOD and a meteorological parameter than to AOD alone (Table 6). What's more, the data do show that meteorological factors enhance the explanatory power of the AOD on cloud parameters over both regions. For example, the individual q values for the influence of AOD and PVV over the ECS were 0.81 and 0.70 but for the combined influence the q-statistic is as high as 0.92. The results from the GDM interaction detector analysis clearly show the enhancement of the interaction q-values over the q-values for the individual factors. In other words, the explanatory power of the combined effects of aerosol and a meteorological parameter is larger than that of each parameter alone. Thus, the GDM provides an alternative way to obtain information on confounding effects of different parameters. We can conclude that aerosol and meteorological conditions do make a significant contribution to cloud parameters and that confounding effects of different factors are often more important than each parameter alone and that the relative importance of each parameter differs significantly over the ECS and YRD.

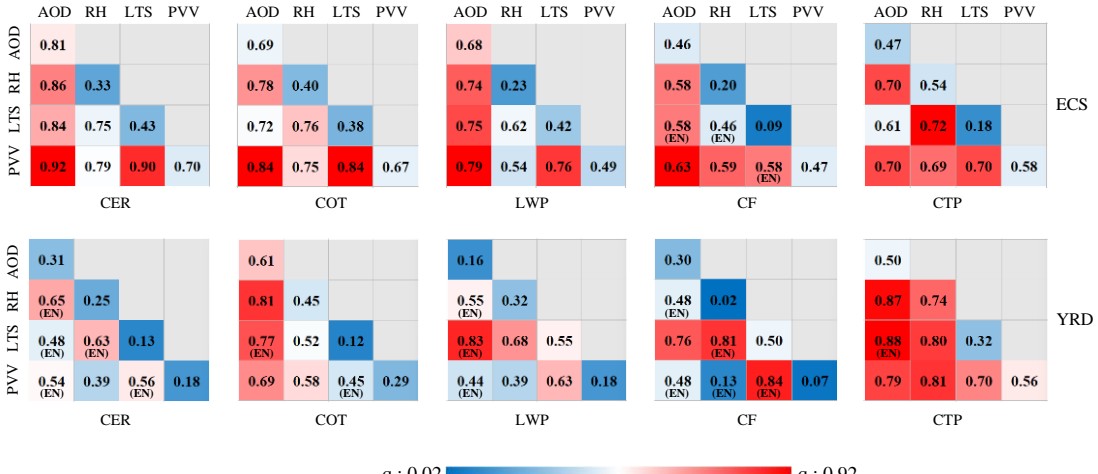

**Figure 10. q values derived using the GDM for the combined effects of AOD, RH, LTS and PVV on cloud parameters over the ECS (top) and the YRD (bottom). In addition to the numbers, the q values are colour coded according to the colour scale (linear from 0.04 to 0.92) at the bottom, for easy identification. (EN) below a q value indicates the nonlinear enhancement of two variables (if q(x1∩x2) > q(x1) + q(x2)), the absence of a label below a q value indicates a bilinear enhancing of two variables (if q(x1∩x2) > Max[q(x1), q(x2)]).**

## 5 Discussion

Warm cloud properties over eastern China have been investigated in relation to aerosol and meteorological conditions using 15 years (2008-2022) of data from passive (MODIS/Aqua) satellite measurements, together with the ECMWF ERA-5 reanalysis meteorological data. The Yangtze River

Delta, a heavily polluted region in eastern China, and the East China Sea with a relatively clean atmosphere, were selected as study areas. Relationships between cloud droplet effective radius and AOD (used as a proxy for CCN), i.e. the sensitivity S of CER to changes in AOD, were constructed for different constraints of AOD and LWP. The effects of AOD on CER were investigated for three AOD regimes. In view of the uncertainty of MODIS-retrieved AOD and the scatter in the CER-AOD relations, data for AOD<0.1 were not considered. In the moderately polluted AOD regime (0.1<AOD<0.3), the CER over the YRD did not change significantly with AOD, whereas over the ECS the CER strongly decreased with AOD and the derived relationship between CER and AOD is statistically significant. In the third AOD regime, with AOD > 0.3, the CER increased with increasing AOD over the YRD. In contrast, over the ECS there was no clear relation between CER and AOD, although CER variability increased with AOD>0.3, especially for higher AOD (> ~0.8). Based on these results, two different AOD regimes were selected for further investigation of aci: 0.1<AOD<0.3 over the ECS and AOD > 0.3 over the YRD. The spatial distribution of S, here defined as the relative change in CER as a function of the relative change in AOD (Eq. 1), averaged over the 15-years study period, shows that it was negative and statistically significant over the ECS and positive over the YRD. These results were obtained using data with no restriction on LWP. Stratification by LWP shows that over the YRD, for AOD > 0.3, S is positive for LWP in the interval [0-120 g m$^{-2}$] with very small differences between three LWP intervals (0-40, 40-80 and 80-120 g m$^{-2}$). In contrast, over the ECS, for AOD in the range from 0.1 to 0.3, S is negative in the LWP interval [40-200 g m$^{-2}$] and the value of S is substantially different between the 4 LWP intervals, with S increasing with LWP, as shown in Table 3.

These results were obtained using data from a period of 15 years. During this period, the aerosol properties changed in response to expanding economy, resulting in the increase of the AOD until 2007, and the implementation of emission reduction policy resulting in the decrease of the AOD from 2014 which flattened from about 2018 (de Leeuw et al., 2021; 2022; 2023). To account for these changes, the sensitivity S was determined for the periods 2008-2014 and 2015-2022, without stratification for LWP (see Figures S1 and S2 in the Supplementary). The results for the ECS show no significant difference between the CER-AOD relations during these two periods. Over the YRD, however, the data for 2008-2014 show a clear decrease of CER with increasing AOD for 0.1<AOD<0.3 and for larger AOD the CER increased, with a statistical significant correlation (R=0.87) and S=0.10 as compared to S=0.08 for the

whole period. In contrast, the data for 2015-2022 show no clear correlation between CER and AOD for both AOD intervals over the YRD. A similar exercise for shorter periods, i.e. for each year between 2008 and 2022, show similar behavior as for the whole period 2008-2022, over both study areas, with interannual variations of the value of S. However, the statistical significance is low (large p) due to the small number of data samples in each year.

It is noticed that in recent papers (e.g., Gryspeerdt et al., 2023; Arola et al., 2022) the usefulness of correlating aerosol and cloud parameters has been seriously challenged because cloud variability and retrieval errors are such that correlations between AOD and cloud properties ($N_d$, CER, LWP) can be spurious. Gryspeerdt et al. (2023) discussed aci in terms of the susceptibility $\beta$ of $N_d$ to aerosol rather than the sensitivity S of CER to aerosol (see the discussion in the Introduction on the use of $N_d$ vs CER), and the problem arises with low aerosol conditions due to larger aerosol retrieval uncertainty due to surface correction (larger surface effect on the radiance at the top of the atmosphere), which applies equally to $\beta$ and S. In the current study we did not consider the lowest aerosol conditions by limiting the data to situations with AOD $\geq$ 0.1, as discussed in Section 4.2. Furthermore, we stratified the analysis for moderate (0.1 $\leq$ AOD < 0.3) and high (0.3 $\geq$ AOD) aerosol regimes, based on the data.

Arola et al. (2022) addressed the susceptibility of $N_d$ to changes in aerosol and the adjustment of LWP (using satellite observations), and confounding factors, in particular co-variability of $N_d$ and LWP induced by meteorological effects. They show how errors in the retrieved CER and COT or spatial heterogeneity in cloud fields influence the $N_d$ - LWP relation. However, both $N_d$ and LWP are not retrieved but derived from CER and COT. Using Eq. 1 and Eq. 2 in Arola et al. (2022), the $N_d$-LWP relationship can be shown to have a highly non-linear dependence on CER and thus it is no surprise that any error in CER strongly affects the relation between $N_d$ and LWP. Their experiments, i.e. using smaller scales (5° x 5°) to reduce spatial meteorological variability, or using snapshots to remove meteorological variability in time, did not lead to a conclusion whether the $N_d$ - LWP variability is due to spatial heterogeneity in the cloud fields or due to retrieval errors. The main message from this part of the study (using satellite data) by Arola et al. (2022) is "the spatial variability of CER introduces a bias which moreover becomes stronger in conditions where the CER values are lower on average". Experiments with simulated measurements show that "the main cause of the negative LWP vs $N_d$ slopes is the error in CER". Arola et al. emphasize that the spatial cloud variability and retrieval errors in CER and COT are

similar sources for negative bias in LWP adjustment and that these sources could not be separately assessed in their simulations. The implication of the findings of Arola et al. (2022) on the adjustment of LWP for the results of the current study on the sensitivity of CER to aerosol (or CCN, using AOD as proxy) is that the assumption of constant LWP may be violated. This would affect the results presented in Section 4.3 where LWP was stratified and S was found to vary with LWP. In view of the LWP adjustment to changes in aerosol, the variation of CER sensitivity with LWP may be somewhat different from that reported in section 4.3.

The above results were obtained by using traditional statistical methods where relationships were derived from scatterplots of CER versus AOD, stratified in two different AOD regimes and five different LWP regimes, as discussed above. The data were also analyzed by using the GDM to determine which factors influence aci and identify how interactions between different parameters influence the results of the aci analysis, i.e. the sensitivity and resulting adjustments. In particular, the GDM provides information on the extent to which the effect of individual factors is influenced by other factors. As shown in Section 4.6.1, the effect of individual factors may be overestimated when confounding effects of other factors are not accounted for. The interaction detector analysis (Section 4.6.2) shows a more realistic estimate of the effects on aci when different factors are analyzed together. The factor detector analysis (Section 4.6.1) shows that over the ECS, cloud parameters are most sensitive to AOD, as indicated by the large and statistically significant q values. Among the meteorological factors, PVV has more influence on the variations of the cloud parameters than RH and LTS. Over the YRD, AOD has the largest influence on COT, with large and significant q values. Among the meteorological factors, the effect of LTS on CF is greater than that of RH and PVV. However, the q-values may sum up to over 100% when the variables are not independent, i.e. the explanatory power of such variables is too high. The evaluation of the effects of interaction between different factors on aci corrects these clearly unrealistic situations. The analysis in section 4.6.2 shows that the interactive q-statistic values derived in this study are larger than any of the values for single variables, i.e. the explanatory power of a combination of factors is higher than that of individual factors, but less than 100%. However, although the GDM provides evidence of the effects of aerosol and meteorological factors and their interactions on cloud properties and quantifies the relative contributions to aci, it cannot quantify the absolute contributions with confidence. Moreover, it should be noted that although the results show correlations, they do not provide evidence that the aerosol

variation indeed causes some change in cloud properties. As regards large regions: Grandey and Stier (2010) recommend 4° x 4° as the largest size and "if data exist at higher gridded resolution the possibility of analyzing data at this higher resolution should be seriously considered." In this study the resolution of MYD 08 data used is 1° x 1°, the GDM doesn't detect significant relationships for regions smaller than 9° x 9° due to insufficient samples. In the future, higher resolution data can be used for GDM by

controlling the size of the study area to be less than 4° x 4°.

**6 Conclusions**

The response of different cloud parameters to variations in AOD and in meteorological conditions has been analyzed using traditional statistical methods to determine the sensitivity S of CER to aerosol for different aerosol regimes and stratified according to LWP. The results show the contrasting behavior over

a polluted region over land (YRD) and a relatively clean region over ocean (ECS). In the intermediate aerosol regime (0.1<AOD<0.3), CER does not significantly change with AOD over the YRD (S≈0), but over the ECS S is negative and increases with increasing LWP. In the high aerosol regime (AOD>0.3), S is positive over the YRD but varies little with LWP, whereas over the ECS the CER does not change with AOD. These results may be influenced by confounding effects of meteorological parameters. The study

further shows that over the ECS the CER is larger for higher LTS and RH but lower for higher PVV. Over the YRD, there is no significant influence of LTS on the relationship between CER and AOD.

The GDM has been applied to determine which factors influence S and cloud parameters and the interaction detector analysis has been used to determine the combined effect of different parameters on cloud parameters. The results from the GDM interaction detector analysis clearly show the enhancement

of the interaction q-values over the q-values for the individual factors. In other words, the explanatory power of the combined effects of aerosol and a meteorological parameter is larger than that of each parameter alone. Thus, the GDM provides an alternative way to obtain information on confounding effects of different parameters. We conclude that aerosol and meteorological conditions significantly influence cloud parameters and that combined effects of different factors are often more important than

the effect of each individual factor. The relative importance of each factor differs significantly over the ECS and YRD.

The results of this study contribute to improve the understanding of the indirect effects of aerosols and

the role of various driving factors on the cloud microphysical properties. By comparison with aerosol and cloud observations, the regional climate model's ability to simulate changes in cloud parameters can

be evaluated. A more accurate description of the relative contribution of meteorological factors can improve the parameterization scheme of the model over eastern China.

### *Data availability*

All data used in this study are publicly available. The satellite data from the MODIS instrument used in this study were obtained from https://ladsweb.nascom.nasa.gov/search/ (last access: 12 July 2022, Liu,

2022a). The the ECMWF ERA-5 reanalysis data were collected from the ECMWF ERA-5 reanalysis data server https://cds.climate.copernicus.eu/cdsapp#!/dataset/reanalysis-era5-pressure-levels-monthly-means?tab=form (last access: 12 July 2022, Liu, 2022b).

### *Author contributions*

YL and GL designed the research. YL led the analyses. YL and LT wrote the manuscript with major

input from JH, GL and further input from all other authors. All authors contributed to interpreting the results and to the finalization and revision of the manuscript.

### *Competing interests*

The authors declare that they have no conflict of interest.

### *Acknowledgements*

This work was supported by the National Natural Science Foundation of China (Grant No. 42001290), the Natural Science Foundation of China (Grant No. 41871253) and the National Key R&D Program of China (Project No. 2022YFC3800701). We are grateful for the easy access to MODIS data products provided by NASA. We also thank ECMWF for providing daily ERA-5 reanalysis data. The study contributes to the ESA / MOST cooperation project DRAGON5, Topic 3 Atmosphere, sub-topic 3.2 Air-

Quality.

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
