# Peer review of "Opposite effects of aerosols and meteorological parameters on warm clouds in two contrasting regions over eastern China"

_EGUsphere, 2023_

## Author Comment (AC1)

**Response to Referee #1**

The authors look at the primarily at the controls on the relationship between AOD and cloud effective radius (CER), both meteorological parameters and other cloud properties. Concentrating on two regions (over land and ocean near China), this study also introduces the geographical detector method (GDM) to the study of aerosol-cloud interactions. They also look at the impact of meteorological properties on the relationship between AOD and cloud properties more generally.

The introduction of the GDM to this study is novel and of interest to the readers of ACP. With some extra explanation, I think this paper would make a really useful introduction to the method for others in atmospheric science. However, I have a number of concerns about the paper as a whole that would have to be addressed before I would recommend publication.

The authors are grateful to Referee #1 for the valuable time spent on thorough reading our manuscript and providing expert views to guide us for improving the manuscript with the main and specific points and the references. We have taken notice of all comments, listed below in black, and made many changes to the manuscript to address these, together with the comments from the other referees. We address each of your comments below and refer to our responses in the revised manuscript and provide line numbers and copy text in "quotes".

To ensure that the data used only included single layer liquid clouds and nonprecipitating cases, the filtering criteria described by Saponaro et al. (2017) were applied. It is noted that all the figures have been updated throughout the revised manuscript.

**Main points**

1. The introduction of the GDM method is a really nice aspect of this study. However, I feel it could be explained and examined in more detail, as there are a number of factors that are unclear to someone meeting this method for the first time. For example, the impact of the explanatory variables in Tables 4 and 5 sum to over 100%. This is not what I would have expected. Similarly, I am not familiar with the 'interaction detector' or the 'interactive q-values'. What do these mean? How should they be interpreted? Likewise, the term 'nonlinear enhancement of the influence of the independent parameters' on L456 is not straightforward to someone new to this method.

**Answer**: In statistics, the q-value is a measure used to evaluate the explanatory power of variables on the dependent variable. When multiple independent variables are considered separately, it is indeed possible for the sum of the q-values of multiple X variables to exceed 100%. When they are considered together, this is referred to as 'interaction q-value'. This situation is quite common and similar to the issue in multiple linear regression. The main reason for this is the presence of correlation among the X variables, indicating that these variables are not independent. Consequently, multiple independent variables may contribute to the dependent variable in a similar manner, leading to a sum of q-values over 100%.

To better explain this and clarify "interaction detector" and "interaction q-values", we have replaced the text below figure 2 (lines 354-373) with "The interaction detector can be used to test for the influence of interaction between different influencing factors, e.g., x1 and x2, on the dependent factor (y) and whether this interaction weakens or enhances the influence of each

of x1 or x2 on the dependent variable, y, or whether they are independent in influencing y. For example, Figure 3(a) shows the spatial distribution of the dependent variable, y. The factors x1 and x2 both vary across the study region, but in different ways, and for each factor different sub-regions can be distinguished by application of the Jenks classification method described above to each factor separately. This is illustrated in Figures 3(b) and 3(c) where, as an example, three different sub-regions are considered for each factor. Usually, the dependent variable y is influenced by several different factors xi (Figure 3) and the combined effect of two or more factors may have a weaker or stronger influence on y than each of the individual factors. The q values for the influences of factors x1 and x2 on y, obtained from the application of the factor detector method (Eq. 2), may be represented as q (x1) and q (x2). Hence, a new spatial unit and subregions may be generated by overlaying the factor strata x1 and x2, written as x1∩x2, where ∩ denotes the interaction between factor strata x1 and x2 as illustrated in Figure 3(d). Thus, the q value of the interaction of x1∩x2 may be obtained, represented as q (x1∩x2). Comparing the q value of the interaction of the pair of factors and the q value of each of the two individual factors, five categories of the interaction factor relationship can be considered which are summarized in Table 2. If q(x1∩x2) > q(x1) + q(x2), this is referred to as a nonlinear enhancement of two variables. And if q(x1∩x2) > Max[q(x1), q(x2)], this is referred to as a bilinear enhancement of two variables. The occurrence of nonlinear enhancement and bilinear enhancement are indicated with the q values in Table 2 and in the caption of Figure 7.".

The occurrence of nonlinear enhancement is indicated with the q values in Table 1 (Table 2 in the revised manuscript) and in the caption of Figure 7 in the revised manuscript.

**Table 1. Interaction categories of two factors and the interaction relationship**

| Illustration | Description | Interaction |
|---|---|---|
| | $q(x1 \cap x2) < Min[q(x1), q(x2)]$ | Weakened, nonlinear |
| | $Min[q(x1), q(x2)] < q(x1 \cap x2) < Max[q(x1), q(x2)]$ | Weakened, unique |
| | $q(x1 \cap x2) > Max[q(x1), q(x2)]$ | Enhanced, bilinear |
| | $q(x1 \cap x2) = q(x1) + q(x2)$ | Independent |
| | $q(x1 \cap x2) > q(x1) + q(x2)$ | Enhanced, nonlinear |

2. The AOD-CER relationship is difficult to interpret as the Twomey effect, especially where LWP is not controlled for (McComiskey and Feingold, 2012). Several previous studies have also investigated the potential controls on the AOD-CER relationship (e.g. Tan et al, 2017; Yuan et al, 2008; Myhre et al, 2007; Tang et al, 2014; Andersen and Cermak, 2015). There should be a clearer distinction around what is added by this work (which can be the GDM).

**Answer**: We have taken notice of the references provided and added them to the revised manuscript where appropriate. McComiskey and Feingold (2012) discuss effects of unconstrained LWP, which in the revised manuscript is mentioned in Section 4.2 (lines 442-444): "Note however, that no selection was made for LWP and the condition of constant LWP was not fulfilled. This will be further discussed in Section 4.3." In section 4.3 and Discussions (lines 757-777) we discuss the effect of LWP on S in detail. McComiskey and Feingold (2012) also discuss the spatial separation of satellite observations of cloud and aerosol is addressed in

Section 2.2 (lines 225-229) and a reference to McComiskey and Feingold (2012) has been added: "Aerosol retrieval is only executed in clear sky conditions whereas cloud properties can only be retrieved in cloudy skies. Hence, it is not possible to obtain co-located aerosol and cloud data from satellite. For satellite-based aci studies it is assumed that, following, e.g., Jia et al. (2022), aerosol properties are homogeneous enough to be representative for those in adjacent cloud areas. Consequences of this assumption were discussed by McComiskey and Feingold (2012).".

Potential controls on the CER-AOD relationship and confounding meteorological effects are discussed in the Introduction, supported with many references (lines 129-160): "Meteorological conditions are important factors determining …. promote the formation of thicker and higher clouds" and in Discussions (lines 778-798): "The above results were obtained by using traditional statistical methods where relationships were derived from scatterplots of CER versus AOD, stratified in two different AOD regimes and five different LWP regimes, as discussed above. The data were also analyzed by using the GDM to determine which factors influence aci and identify how interactions between different parameters influence the results of the aci analysis, i.e. the sensitivity and resulting adjustments. In particular, the GDM provides information on the extent to which the effect of individual factors is influenced by other factors. As shown in Section 4.6.1, the effect of individual factors may be overestimated when confounding effects of other factors are not accounted for. The interaction detector analysis (Section 4.6.2) shows a more realistic estimate of the effects on aci when different factors are analyzed together. The factor detector analysis (Section 4.6.1) shows that over the ECS, AOD has the largest influence on cloud parameters, as indicated by the large and statistically significant q values. Among the meteorological factors, PVV has more influence on the variations of the cloud parameters than RH and LTS. Over the YRD, AOD has the largest influence on COT, with large and significant q values. Among the meteorological factors, the effect of LTS on CF is greater than that of RH and PVV. However, the q-values may sum up to over 100% when the variables are not independent, i.e. the explanatory power of such variables is too high. The evaluation of the effects of interaction between different factors on aci corrects these clearly unrealistic situations. The analysis in section 4.6.2 shows that the interactive q-statistic values derived in this study are larger than any of the values for single variables, i.e. the explanatory power of a combination of factors is higher than that of individual factors, but less than 100%. However, although the GDM provides evidence of the effects of aerosol and meteorological factors and their interactions on cloud properties and quantify the relative contributions to aci, it cannot quantify the absolute contributions with confidence.".

We have added the references provided by Referee#1 with a brief summary of the findings reported in these references. In Section 6 Conclusions, we have added (lines 807-809): "These results may be influenced by confounding effects of meteorological parameters. The study further shows that over the ECS the CER is larger for higher LTS and RH but lower for higher PVV. Over the YRD, there is no significant influence of LTS on the relationship between CER and AOD.

However, the main comment of Referee #1 is about the question "what is added by this work" and the answer is indeed that it is the GDM as was already mentioned in the Introduction at lines 164-172 in the revised manuscript: "In the current study the geographical detector method (GDM) is applied as a complementary tool to quantify the relative importance of the effects of nine parameters on S. The GDM is explained in detail in Section 3.2. In brief, a set of statistical methods is used to detect the spatial variability of aerosol and cloud properties, which are

spatially differentiated, and evaluate the occurrence of correlations in their behaviour and the driving forces behind these correlations (Wang and Hu, 2012; Wang et al., 2016). The basic idea of the GDM is that the spatial distributions of two variables tend to be similar if these two variables are connected (Zhang and Zhao, 2018). The method is used in this study to analyse the relative importance of different factors, and interactions between them, influencing aci."

Furthermore, we have changed the first sentence of Section 3.2 (lines 312-313) to "The geographical detector method (GDM) is introduced to analyze which factors influence the aci and identify possible correlations between different factors" and the text below Figure 2 has been changed (see also response to comment 1).

And in the conclusions we have added the text cited above (lines 810-819): "The GDM has been applied to determine which factors influence S and cloud parameters and the interaction detector analysis has been used to determine the combined effect of different parameters on cloud parameters. The results from the GDM interaction detector analysis clearly show the enhancement of the interaction q-values over the q-values for the individual factors. In other words, the explanatory power of the combined effects of aerosol and a meteorological parameter is larger than that of each parameter alone. Thus, the GDM provides an alternative way to obtain information on confounding effects of different parameters. We conclude that aerosol and meteorological conditions significantly influence cloud parameters and that combined effects of different factors are often more important than the effect of each individual factor. The relative importance of each factor differs significantly over the ECS and YRD."

3. The majority of more recent studies have used Nd for calculation of the ACI, rather than CER (Quaas et al, 2008; Gryspeerdt et al, 2017; McCoy et al, 2018; Hasekamp et al, 2019). There are also useful studies that investigate the susceptibility (AOD-Nd relationship) and the impacts on this value (Jia et al, 2022). This avoids the LWP-CER issue prevalent in previous work and presents a cleaner separation of the Twomey and adjustments. It could be worth including a section on why CER is used and might be something to consider for future work in this area.

**Answer**: We address the use of CER rather than $N_d$ with the following text, added to the Introduction (lines 177-186): "It is noted that $RF_{aci}$ is formulated in terms of $N_d$, whereas studies on the Twomey effects often use CER instead of $N_d$. CER is readily available as a satellite retrieval product, although in particular over land the reliability is questioned (Grandey and Stier, 2010), whereas $N_d$ is derived from CER and the cloud optical thickness (COT) (e.g., Grandey and Stier, 2010; Arola et al., 2022). This implies that $N_d$ is subject to the same retrieval errors as CER, including a possible relation between CER and LWP. The comparison of global maps of the sensitivities of CER and $N_d$ to AOD by Grandey and Stier (2010) exhibits very similar patterns. In this study, the CER sensitivity to AOD is stratified by LWP, which however poses problems in the evaluation of $RF_{aci}$. However, the current study focuses on understanding effects of different parameters on CER sensitivity to aerosol rather than the application to determine $RF_{aci}$.".

4. Previous studies have shown that it is difficult to interpret correlations over large regions as an aerosol effect due to the impact of meteorological confounders (Grandey and Stier, 2010). Correlations between AOD and cloud properties are also fraught with potential confounding effects (Quaas et al, 2010; Boucher and Quaas, 2012; Gryspeerdt et al, 2014). Does the GDM method address these issues? If so how? If not, this study should be much clearer about the claims of causality it puts forward.

**Answer**: Thank you for this comment. We have added the references provided in the introduction with a brief summary of the findings of each of them. This comment addresses two issues: large regions and confounding effects. As regards large regions: Grandey and Stier (2010) recommend 4°x4° as the largest size and "if data exist at higher gridded resolution the possibility of analyzing data at this higher resolution should be seriously considered." In this study we have followed this recommendation: we have considered two large study regions (YRD and ECS) for the initial evaluation of CER-AOD relationships. Based on these results, we have selected data ranges for which a clear ACI relationship occurs and for these conditions we have refined this study to smaller scales using 1°x1° grid cells and the results are presented in Figure 6 and discussed in Sections 4.2 and 4.3 in the revised manuscript.

We have also considered the size of the study area, or grid size, when using the GDM. In the GDM, the y data are recorded in a raster grid, over a total study area of 9°x9°, as illustrated in Figure 1 (Figure 2 in the manuscript). The data in the raster grid is transformed into dot files, each dot containing a value for y and for one of the influencing parameters x. The dependent (y) and influencing (x) parameters are separated into 2 layers with the same grid. As the resolution of MYD 08 data used in this study is 1°x1°, the data transformed into dot files is based on raster grid 1°x1°. Here, 15-year averaged distributions of clouds (y, 5 layers) and aerosols/meteorological conditions (x, 4 layers) are used as input in the GDM. Tables 2 and 3 show the q values for factors which may influence cloud parameters over the ECS and YRD in different regions sizes, evaluated for data collected in the period from 2008-2022. The data in Tables 2 and 3 show that for regions smaller than 9°x9°, the GDM result is not significant and that the results become more significant when the area is getting larger (for example 9°x9°, 10°x10°, 11°x11°, 12°x12°). In future research, higher resolution data can be used for GDM by controlling the size of the study area to be less than 4°x4°.

The problem of large regions and the effect of possible meteorological confounders was also addressed by Arola et al. (2022) and in response to a comment by Referee#2 we have added the following text to Section 5 (Discussion, lines 757-777): "Arola et al. (2022) addressed the susceptibility of $N_d$ to changes in aerosol and the adjustment of LWP (using satellite observations), and confounding factors, in particular co-variability of $N_d$ and LWP induced by meteorological effects. They show how errors in the retrieved CER and COT or spatial heterogeneity in cloud fields influence the $N_d$ - LWP relation. However, both $N_d$ and LWP are not retrieved but derived from CER and COT. Using Eq. 1 and Eq. 2 in Arola et al. (2022), the $N_d$ -LWP relationship can be shown to have a highly non-linear dependence on CER and thus it is no surprise that any error in CER strongly affects the relation between $N_d$ and LWP. Their experiments, i.e. using smaller scales (5° x 5°) to reduce spatial meteorological variability, or using snapshots to remove meteorological variability in time, did not lead to a conclusion whether the $N_d$ - LWP variability is due to spatial heterogeneity in the cloud fields or due to retrieval errors. The main message from this part of the study (using satellite data) by Arola et al. (2022) is "the spatial variability of CER introduces a bias which moreover becomes stronger in conditions where the CER values are lower on average". Experiments with simulated measurements show that "the main cause of the negative LWP vs $N_d$ slopes is the error in CER". Arola et al. emphasize that the spatial cloud variability and retrieval errors in CER and COT are similar sources for negative bias in LWP adjustment and that these sources could not be separately assessed in their simulations. The implication of the findings of Arola et al. (2022) on the adjustment of LWP for the results of the current study on the sensitivity of CER to aerosol (or CCN, using AOD as proxy) is that the assumption of constant LWP may be violated. This would affect the results presented in Section 4.3 where LWP was stratified and S was

found to vary with LWP. In view of the LWP adjustment to changes in aerosol, the variation of CER sensitivity with LWP may be somewhat different from that reported in section 4.3."

The results from the GDM interaction detector analysis in Section 4.6.2 clearly show the enhancement of the interaction q-values over the q-values for the individual factors. In other words, the explanatory power of the combined effects of a meteorological parameter and aerosol is larger than that of each parameter alone. Thus, the GDM provides an alternative way to obtain information on confounding effects of different parameters.

[Figure]

**Figure 1. The principle of the geographical detector method. See text for explanation.**

**Table 2. q values for factors which may influence cloud parameters over the ECS in areas with different sizes, evaluated for data collected in the period from 2008-2022.**

| Region | Cloud | AOD | RH | LTS | PVV |
|--------|-------|-----|-----|-----|-----|
| 4°x4° | CER | 0.67 | 0.43 | 0.14 | 0.59 |
| | COT | 0.27 | 0.51 | 0.28 | 0.44 |
| | LWP | 0.72 | 0.25 | 0.09 | 0.53 |
| | CF | 0.68 | 0.29 | 0.11 | 0.28 |
| | CTP | 0.31 | 0.50 | 0.29 | 0.57 |
| 5°x5° | CER | 0.79 | 0.42 | 0.40 | 0.72 |
| | COT | 0.54 | 0.41 | 0.25 | 0.48 |
| | LWP | 0.57 | 0.37 | 0.34 | 0.52 |
| | CF | 0.46 | 0.48 | 0.16 | 0.53 |
| | CTP | 0.49 | 0.45 | 0.25 | 0.31 |
| 6°x6° | CER | 0.85[**] | 0.47 | 0.49 | 0.54 |
| | COT | 0.62 | 0.62 | 0.37 | 0.56 |
| | LWP | 0.64 | 0.34 | 0.41 | 0.31 |
| | CF | 0.30 | 0.23 | 0.16 | 0.40 |
| | CTP | 0.46 | 0.47 | 0.22 | 0.45 |
| 7°x7° | CER | 0.73[***] | 0.36 | 0.39 | 0.63[***] |
| | COT | 0.60 | 0.37 | 0.37 | 0.53[**] |
| | LWP | 0.58 | 0.19 | 0.31 | 0.47 |
| | CF | 0.34 | 0.13 | 0.05 | 0.43[**] |

| Region | Cloud | AOD | RH | LTS | PVV |
|--------|-------|-----|-----|-----|-----|
| | CTP | 0.37 | 0.47 | 0.21 | 0.47 |
| 8°x8° | CER | 0.86*** | 0.48 | 0.57*** | 0.63** |
| | COT | 0.72*** | 0.50 | 0.54 | 0.65*** |
| | LWP | 0.71 | 0.41 | 0.48 | 0.51 |
| | CF | 0.39 | 0.27 | 0.14 | 0.42** |
| | CTP | 0.48 | 0.56 | 0.56 | 0.56 |
| 9°x9° | CER | 0.81*** | 0.33*** | 0.44*** | 0.70*** |
| | COT | 0.69*** | 0.40 | 0.38 | 0.67*** |
| | LWP | 0.68*** | 0.23 | 0.43*** | 0.49*** |
| | CF | 0.46*** | 0.20 | 0.09 | 0.47*** |
| | CTP | 0.47 | 0.53 | 0.18 | 0.58 |
| 10°x10° | CER | 0.86*** | 0.46*** | 0.54*** | 0.62*** |
| | COT | 0.71*** | 0.55** | 0.52*** | 0.67*** |
| | LWP | 0.72*** | 0.29 | 0.39*** | 0.47*** |
| | CF | 0.49*** | 0.19 | 0.09 | 0.44*** |
| | CTP | 0.53 | 0.58 | 0.29 | 0.66 |
| 11°x11° | CER | 0.87*** | 0.45*** | 0.39*** | 0.54*** |
| | COT | 0.71*** | 0.53*** | 0.45*** | 0.60*** |
| | LWP | 0.73*** | 0.73 | 0.26 | 0.44*** |
| | CF | 0.48*** | 0.13 | 0.04 | 0.29*** |
| | CTP | 0.62 | 0.52 | 0.26 | 0.57 |
| 12°x12° | CER | 0.84*** | 0.42*** | 0.31 | 0.55*** |
| | COT | 0.66*** | 0.46*** | 0.37 | 0.52*** |
| | LWP | 0.64*** | 0.30 | 0.05 | 0.41*** |
| | CF | 0.42*** | 0.13 | 0.11 | 0.24*** |
| | CTP | 0.53 | 0.49 | 0.27 | 0.54 |

Note: ***indicates that the q value is significant at the 0.01 level ($p < 0.01$), **indicates that the q value is significant at the 0.05 level ($p < 0.05$).

**Table 3. q values for factors which may influence cloud parameters over the YRD in areas with different sizes, evaluated for data collected in the period from 2008-2022.**

| Region | Cloud | AOD | RH | LTS | PVV |
|--------|-------|-----|-----|-----|-----|
| 4°x4° | CER | 0.47 | 0.64 | 0.04 | 0.27 |
| | COT | 0.75 | 0.62 | 0.37 | 0.55 |
| | LWP | 0.60 | 0.54 | 0.49 | 0.60 |
| | CF | 0.42 | 0.62 | 0.06 | 0.62 |
| | CTP | 0.85 | 0.59 | 0.53 | 0.77 |
| 5°x5° | CER | 0.28 | 0.53 | 0.24 | 0.17 |
| | COT | 0.79 | 0.43 | 0.43 | 0.53 |
| | LWP | 0.69 | 0.49 | 0.44 | 0.38 |
| | CF | 0.46 | 0.42 | 0.33 | 0.51 |
| | CTP | 0.86 | 0.69 | 0.57 | 0.60 |
| 6°x6° | CER | 0.17 | 0.14 | 0.18 | 0.11 |
| | COT | 0.75 | 0.30 | 0.27 | 0.41 |
| | LWP | 0.71 | 0.32 | 0.31 | 0.26 |
| | CF | 0.30 | 0.34 | 0.12 | 0.39 |
| | CTP | 0.81 | 0.53 | 0.40 | 0.54 |

| | | | | | |
|---|---|---|---|---|---|
| | CER | 0.18 | 0.21 | 0.06 | 0.19 |
| | COT | 0.75 | 0.47 | 0.21 | 0.53 |
| 7°x7° | LWP | 0.43 | 0.44 | 0.52 | 0.33 |
| | CF | 0.36 | 0.26 | 0.13 | 0.23 |
| | CTP | 0.73 | 0.75 | 0.46 | 0.65 |
| | CER | 0.31 | 0.24 | 0.34 | 0.17 |
| | COT | 0.66*** | 0.45 | 0.24 | 0.31 |
| 8°x8° | LWP | 0.21 | 0.43 | 0.60 | 0.38 |
| | CF | 0.28** | 0.07 | 0.68*** | 0.05 |
| | CTP | 0.60 | 0.75 | 0.45 | 0.56 |
| | CER | 0.31 | 0.25 | 0.13 | 0.18 |
| | COT | 0.61*** | 0.45*** | 0.12 | 0.29 |
| 9°x9° | LWP | 0.16 | 0.32 | 0.55*** | 0.18 |
| | CF | 0.30*** | 0.02 | 0.50*** | 0.07 |
| | CTP | 0.50 | 0.74*** | 0.32 | 0.56 |
| | CER | 0.41 | 0.28 | 0.20 | 0.27 |
| | COT | 0.63*** | 0.50** | 0.08 | 0.37 |
| 10°x10° | LWP | 0.21 | 0.36 | 0.51*** | 0.22 |
| | CF | 0.38*** | 0.06 | 0.48*** | 0.17 |
| | CTP | 0.50 | 0.78 | 0.31 | 0.60 |
| | CER | 0.35 | 0.28 | 0.17 | 0.22 |
| | COT | 0.69*** | 0.40*** | 0.06 | 0.46 |
| 11°x11° | LWP | 0.35** | 0.28 | 0.40 | 0.24 |
| | CF | 0.39*** | 0.06 | 0.47*** | 0.15 |
| | CTP | 0.48 | 0.72*** | 0.24 | 0.49 |
| | CER | 0.32 | 0.19 | 0.19 | 0.16 |
| | COT | 0.50*** | 0.28*** | 0.07 | 0.47 |
| 12°x12° | LWP | 0.18** | 0.25*** | 0.36*** | 0.26 |
| | CF | 0.37*** | 0.06 | 0.45*** | 0.12 |
| | CTP | 0.32 | 0.65*** | 0.25 | 0.35 |

Note: ***indicates that the q value is significant at the 0.01 level (p < 0.01), **indicates that the q value is significant at the 0.05 level (p < 0.05).

5. Another important factor is the calculation of LWP that is used for binning in the ACI calculations. As the LWP depends on the CER, does this not lead to an implicit filtering by CER, which would affect the calculation of ACI?

**Answer**: Stratification of the data for LWP was applied by, e.g., Saponaro et al. (2017) and Ma et al. (2018) in an attempt to satisfy the conditions for the Twomey effect. Indeed, Ma et al. (2018) show that the variation of the CER vs AI (both stratified according to LWP) relation changes with changes in LWP. The data in Section 4.3 also show the variation of S with the LWP interval and likely this is a more continuous variation if smaller LWP intervals (quasi-constant LWP, Ma et al., 2018) would be used. So indeed the assumption of constant LWP is not truly satisfied and this indeed affects the calculation of ACI as can be deduced from the data in Table 4 below (Table 3 in the revised manuscript). We further note that Arola et al. (2022) and others show a clear LWP - $N_d$ relationship, in agreement with other studies. And LWP and $N_d$ are both calculated from CER and COT, so a relationship is expected. We have addressed the findings by Arola et al. (2022) and this text was copied in our response to

comment 4. We have added the following text in Section 4.3 (lines 528-534): "The variation of S with changes in LWP indicates that the condition of constant LWP is not truly satisfied: if the data would be stratified according to smaller LWP intervals (quasi-constant LWP, Ma et al., 2018), S would likely vary more smoothly with LWP. As mentioned in the Introduction, LWP is not directly retrieved but calculated form CER and COT and thus also the calculation of S is to some extend affected by LWP. We further note the results by Ma et al. (2018), i.e. the slope of CER versus AI (comparable to S in this paper) varies little with LWP, with positive values over land and negative values over ocean and thus behaves similar to the data in Table 4 (Table 3 in the revised manuscript) for YRD and ECS.".

**Table 4. Estimates of S, computed using Eq. (1), and correlation coefficients R between CER and AOD, stratified by LWP, over the ECS for 0.1<AOD<0.3 and over the YRD for AOD>0.3. Statistically significant data points are indicated with * (p value < 0.01).**

| LWP (g m$^{-2}$) | ECS (0.1<AOD<0.3) | | YRD (AOD>0.3) | |
| | S | R | S | R |
| --- | --- | --- | --- | --- |
| 0-40 | 0.10 | 0.94* | 0.08 | 0.63* |
| 40-80 | -0.19 | -0.98* | 0.10 | 0.81* |
| 80-120 | -0.38 | -0.99* | 0.06 | 0.57* |
| 120-160 | -0.41 | -0.99* | -0.03 | -0.11 |
| 160-200 | -0.46 | -0.98* | -0.14 | -0.42* |

6. Is there a reason for using AOD, rather than a product such as the aerosol index (Nakajima et al, 2001), which has a stronger link to the CCN concentration?

**Answer**: AI=AOD*AE, but AE retrieval over land from MODIS is problematic (Ma et al., 2018 refers to Sayer et al., 2013) and therefore is no longer provided as a MODIS product in C6! We cite Ma et al.: "using AOD instead of AI does not influence the conclusions. (next to their Table 1)"

Another argument may come from Gryspeerdt et al. (2023): "The larger relative error in the aerosol retrieval under clean conditions reduces the correlation between the CCN and the retrieved aerosol due to regression dilution (Pitkänen et al., 2016). This reduces the magnitude of β under clean conditions, as observed in Fig. 1a and b. This issue is particularly severe for AI, which is calculated using the ratio of aerosol optical depths at two wavelengths, resulting in a relative error which tends to infinity under clean conditions" (β = d ln Nd / d Ln A, where A is the aerosol proxy AI or AOD). The problem occurs under clean aerosol conditions because the contribution of the surface to the TOA result in larger uncertainty in the retrieved AOD.

We have added the following text in the Introduction (lines 106-122): "In studies on S utilizing satellite data, which is the subject of the current study, the aerosol optical depth (AOD) is often used as a proxy for the aerosol concentration, which is justified by the correlation of AOD and CCN published by Andreae (2009). However, AOD is determined by all aerosol particles in the atmospheric column, including particles that do not act as CCN, depends on the relative humidity (RH) throughout the atmospheric column, does not provide information on chemical composition and may be influenced by aerosol in disconnected layers. The use of the Aerosol Index (AI), the product of AOD and the Ångström Exponent (AE; describing the spectral

variation of AOD), is suggested as a better indicator of CCN because AE includes information on aerosol size (e.g., Nakajima et al., 2001). However, the AE is determined from AOD retrieved at two or more wavelengths and the evaluation of the results versus ground-based reference data shows the large uncertainty in AE. Therefore, in recent MODIS product Collections, AE is not provided over land (e.g., Levy et al., 2013; Kourtidis et al., 2015). AE is also not well-defined for low AOD for which uncertainty is largest (Bellouin et al., 2020; Gryspeerdt et al., 2023). The issues associated with using AOD or AI as proxy for CCN were discussed by, among others, Rosenfeld et al. (2014) who do not recommend the use of AI while also concluding that no better proxy is available. Therefore, in this study, AOD is used as a proxy for CCN to study S. It is noted that in other studies, e.g., Jia et al., 2022, both AOD and AI have been used and the results show similar behaviour.".

**Specific points**

1. The abstract mostly list results, rather than providing an overview of the paper and the conclusions. Is there an overall picture or aim of the study that could help to structure this?

**Answer**: We have revised the abstract substantially and added to following sentence upfront, to provide the overall picture "The sensitivity (S) of cloud parameters to the influence of different aerosol and meteorological parameters has in most previous aerosol-cloud interaction (aci) studies been addressed using traditional statistical methods. In the current study, relationships between cloud droplet effective radius (CER) and aerosol optical depth (AOD, used as a proxy for cloud condensation nuclei, CCN), i.e. the sensitivity (S) of CER to AOD, is investigated with different constraints of AOD and cloud liquid water path (LWP). In addition to traditional statistical methods, the geographical detector method (GDM) has been applied to quantify the relative importance of the effects of aerosol and meteorological parameters, and their interaction, on S." Note that many other changes were made to the abstract in "track changes".

2. L39 - Is this opposite effect just because the sign of the pressure vertical velocity is defined differently? I am not sure what opposite means in this context.

**Answer**: We can see that the CER decreases with increasing AOD over the ECS, which is consistent with the Twomey effect. The meteorological parameters do no change the trend of CER variation to the AOD. However, the CER is larger for higher LTS and RH but lower for higher PVV. We also reorganized the text with "The study further shows that over the ECS the CER is larger for higher LTS and RH but lower for higher PVV." (see lines 32-33) in the revised manuscript.

3. L68 - The terms first and second indirect effect are less commonly used in more recent studies. I would suggest referring to adjustments instead (see IPCC AR5), as this more closely links in with the radiative forcing/effective radiative forcing distinction and aligns more closely with other recent work.

**Answer**: Thank you for this valuable comment. We have changed the terminology throughout the revised manuscript and used several key references to guide us, such as IPCC AR5, Gryspeerdt et al. (2023), Bellouin et al. (2020) and several others.

4. L81 - I would have said that satellites typically have a fairly poor temporal resolution (unless the authors are referring to geostationary satellites?)

**Answer**: In this paper we only use MODIS data. We have removed "and high spatial and temporal resolution"

5. L93 - Is there any reason for choosing these studies? They seem to be rather disjointed, with some looking at the Twomey effect directly and some considering adjustments. Some notable studies looking at the impact of meteorological parameters on potential adjustments (e.g. Koren et al, 2010) and the particularly Twomey effect (Jones et al, 2009; Jia et al. 2022) are left out.

**Answer**: We have reorganized the text and added the notable references looking at the impact of meteorological parameters on potential adjustments (e.g. Koren et al, 2010) and the particularly Twomey effect (Jones et al, 2009; Jia et al. 2022). See the text on pages 5-6 (lines 129-164) in the revised manuscript.

6. L96 - PVV is redefined here

**Answer**: Consistent notation has been used through the revised manuscript.

7. L116 - There needs to be some discussion of how the GDM is affected by the results of Grandey and Stier (2010), who suggest that spatial correlations are unreliable. It may be that the results of GS10 are not applicable here, as the GDM method is capable of accounting for the co-variations that drive the results in GS10. If so, it would be good to have some evidence of this, as it would provide more significance to the results presented in this work.

**Answer**: Spatially-varying aerosol and cloud properties may contribute towards observed relationships between aerosol and cloud properties. This may affect the results of many of the aforementioned studies which analyze data on a relatively large regional scale. Aerosol type, cloud regime and synoptic regime may vary over such large spatial scales. If data are analyzed for the region as a whole, false correlations may be introduced. Grandy and Stier (2010) suggested that for region sizes larger than 4°x4°, spurious spatial variations in retrieved cloud and aerosol properties can introduce widespread significant errors to calculation S. However, we can observe that at the regional scales of 8°x8° and 15°x15°, although significant errors are introduced, the spatial distribution patterns of S (the sensitivities of CER and $N_d$ to AOD) look very similar, as shown in Figure 2 of Grandy and Stier (2010).

GDM is a spatial statistical analysis method aimed at studying the degree of influence and spatial patterns of different factors on the changes in geographic phenomena. In the analysis, we can simultaneously consider the interactions and impacts among multiple factors, thus revealing the relationships of synergistic changes. Therefore, the geographic detector method can encompass the analysis of co-variations.

We have added the following text in the Sect 4.6.1 (lines 647 to 666): "Tables 5 and 6 list q values for individual factors, together with p showing the absence of statistical significance in many cases, especially over the YRD, and often the explanatory power is not high when the significance is low. These data show that cloud parameters are dominated by aerosol effects over the ECS but meteorological influences on cloud parameters predominate over the YRD, as was also concluded from the analysis from "traditional" statistical methods presented in Section 4.5 and these conclusions are consistent with the results published by Andersen and Cermak (2015). Among the meteorological parameters, we also find that PVV (with highest q in the three meteorological parameters) predominantely influences cloud parameters over the ECS. Jones et al. (2009) and Jia et al. (2022) reported that stronger aerosol cloud interactions

typically occur under higher updraft velocity conditions. In addition, we find that CTP is mainly affected by RH (q = 0.74***) and PVV (q = 0.56) over the YRD, as suggested by Koren et al. (2010). Koren et al. reported that observed cloud top height correlates best with model pressure updraft velocity and relative humidity. To some extent, LTS influences CER (q = 0.44***) and LWP (q = 0.43***) over the ECS, while, in contrast, over the YRD LTS predominately influences CF (q = 0.50***) and LWP (q = 0.55***). Matsui et al. (2004) and Tan et al. (2017) reported that aerosol impact on CER is stronger in more dynamic environments that feature a lower LTS and argue that very high LTS environments dynamically suppress cloud droplet growth and reduce aci intensity. While strong correlations between AOD and cloud parameters have been previously observed, they are likely due to the swelling of aerosol particles in humid airmasses (Quaas et al, 2010), rather than an aerosol influence, which is in agreement with findings by, e.g., Myhre et al. (2007), Twohy et al. (2009) and Quaas et al. (2010)."

This study provides a general description of the sensitivity (S) of cloud parameters to the influence of different aerosol and meteorological parameters over the YRD and the ECS. Correlations between AOD and cloud parameters are found over the target regions, which can be attributed in part to the influence of general circulation. In general, there are many relations between the various parameters, both related to cloud microphysics and meteorology. Thus, establishing cause and effect relationships between parameters is difficult and must be made with care. It is not possible to completely separate meteorological influences from aerosol influences on clouds. This work can therefore only provide further evidence of the aerosol and meteorological effect on clouds and quantify the relative contributions and combined effects on clouds, but cannot quantify the absolute contributions with confidence.

In the current study, based on a regional scale of 9°x9°, the GDM method is used to explore the relative importance of various factors on cloud parameters and identify possible correlations between different factors. In the future, aerosol cloud interactions can be studied on smaller regional scale (<4°x4°) using higher resolution source data.

8. L154 - Why only 2008 to 2022? The MODIS record runs back to 2002/3

**Answer**: There is no particular reason for the selection of 2008 as the starting year. Most other studies use a shorter period of time. Based on your comment and that of other referees, we have thought about shorter periods and realized that, in principle, periods were included when the AOD was at its maximum (2008-2014) and when the AOD was decreasing in response to implementation of emission reduction policy. We therefore split the data sets for these 2 periods and plotted CER vs AOD, see Figures 2 and 3 below. We noticed that over the ECS there was not a significant difference between the CER/AOD relations during these two periods. However, over the YRD, for the high AOD period, CER clearly decreased with increasing AOD for 0.1<AOD<0.3 and for larger AOD the CER increased with R=0.87. For the second period, however, there was no clear correlation between CER and AOD for both AOD intervals. The data also show that over the YRD the CER for AOD>0.3 increased to larger values during the first period than during the second period. We did not look for explanations of this difference, possibly the aerosol properties changed in response to emission reduction, or confounding meteorological factors played a role.

We also looked for shorter periods, considering each year between 2008 and 2022. The results show similar behavior for each year over both study areas with interannual variations between

the fits, and thus the value of S. However, the statistical significance is low (large p) due to the small number of samples.

These findings were briefly summarized in the Discussion (lines 733-746): "These results were obtained using data from a period of 15 years. During this period, the aerosol properties changed in response to expanding economy, resulting in the increase of the AOD until 2007, and the implementation of emission reduction policy resulting in the decrease of the AOD from 2014 which flattened from about 2018 (de Leeuw et al., 2021; 2022; 2023). To account for these changes, the sensitivity S was determined for the periods 2008-2014 and 2014-2022, without stratification for LWP (see Figures S1 and S2 in the Supplementary). The results for the ECS show no significant difference between the CER-AOD relations during these two periods. Over the YRD, however, the data for 2008-2014 show a clear decrease of CER with increasing AOD for 0.1<AOD<0.3 and for larger AOD the CER increased, with a statistical significant correlation (R=0.87) and S=0.10 as compared to S=0.08 for the whole period. In contrast, the data for 2014-2022 show no clear correlation between CER and AOD for both AOD intervals over the YRD. A similar exercise for shorter periods, i.e. for each year between 2008 and 2022, show similar behavior as for the whole period 2008-2022, over both study areas, with interannual variations of the value of S. However, the statistical significance is low (large p) due to the small number of data samples in each year."

[Figure]

**Figure 2. CER vs AOD over the YRD for the periods 2008-2014 (left) and 2015-2022 (right).**

[Figure]

**Figure 3. CER vs AOD over the ECS for the periods 2008-2014 (left) and 2015-2022 (right).**

9. L159 - The aerosol and cloud retrievals are necessarily conducted in different regions of the 1x1 degree gridbox (aerosol retrievals are only conducted in clear sky), meaning that they are not coincident. This may not affect the results if the regions are non-precipitating. Jia et al (2022) showed that wet scavenging can have a considerable impact on the susceptibility.

**Answer**: The use of non-collocated aerosol and cloud data is addressed in Section, 2.2. (lines 225-229): "Aerosol retrieval is only executed in clear sky conditions whereas cloud properties can only be retrieved in cloudy skies. Hence, it is not possible to obtain co-located aerosol and cloud data from satellite. For satellite-based aci studies it is assumed that, following, e.g., Jia et al. (2022), aerosol properties are homogeneous enough to be representative for those in adjacent cloud areas. Consequences of this assumption were discussed by McComiskey and Feingold (2012)."

We have filtered the data (exclude precipitating clouds) following the method used in Ma et al. (GRL2018) and Saponaro et al. (2017) (see lines 265-267 in the revised manuscript): "To ensure that the data used only included single layer liquid clouds and nonprecipitating cases, the filtering criteria described by Saponaro et al. (2017) were applied.". This issue is shown throughout the revised manuscript (all the figures were changed/modified in this respect).

10. L153 - I would suggest referencing Platnick et al (2016), given the authors are using MODIS collection 6.1.

**Answer**: We have added the following text to Section 2.2. (lines 236-245): "The MODIS Collection 6.1 AOD product over China has been validated by, e.g., Che et al. (2019) and globally over land and ocean by Wei et al. (2019). MODIS C6.1 cloud products were evaluated by Platnick et al. (2017). The validation of CER and LWP, the primary cloud products used in this paper, was described by Painemal and Zuidema (2011), who compared MODIS C5 with in situ data (aircraft), and likewise the MODIS C6.1 CER product was evaluated by Fu et al. (2022) by comparison with airborne measurements. Fu et al. (2022) concluded that their "validation, along with in situ validation of MODIS CER from other regions (e.g., Painemal and Zuidema, 2011; Ahn et al., 2018), provides additional confidence in the global distribution of bias-adjusted MODIS CER reported in Fu et al. (2019)." It is noted that COT and CER are retrieved whereas LWP is secondarily derived (e.g., Painemal and Zuidema, 2011)."

11. L169 - Brendan et al (2005) suggests that cloud contamination becomes an issue when the AOD is larger than 0.6. Why is a larger threshold used here?

**Answer**: The conclusion of Brendan et al. (2005) applies to the MOD06 Collection 04 cloud product and these authors conclude with "The cloud masking technique in the recently updated Collection 05 cloud retrieval algorithm has been improved, and the Collection 05 cloud products available in the near future will largely eliminate the aerosol contamination effect". Christenson et al. (ACP 2017) used MOD06 C6 data (1km x1km) and reported that "large aerosol optical depths remain in the MODIS-observed pixels near cloud edges, due primarily to 3-D effects (Varnái and Marshak, 2009) and the swelling of aerosols by higher relative humidity." And "Varnái and Marshak (2009) also noted that beyond 15 km contamination effects were minimized in MODIS data (1km x1km)." Therefore Christensen et al. only used data pairs beyond the 15 km length scale in their aci study.

In our study we use MODIS L3 collection 6.1 with a spatial resolution of 1°x1°. Comparisons with surface-based sun photometer data revealed that Collection 6 should improve upon

Collection 5, and overall, 69.4% of MODIS Collection 6 AOD fell within an expected uncertainty of ± (0.05 + 15%) (Levy et al., 2013; Tan et al., 2017). In this study, to eliminate 1° by 1° scenes in which the aerosol distribution is heterogeneous, retrievals with a standard deviation higher than the mean values are discarded (Saponaro et al., 2017; Jia et al., 2022). In addition, many previous researches do not set a threshold of AOD when using MODIS L3 C6 data (Grandey and Stier, 2010; Tang et al., 2014; Saponaro et al., 2017; Tan et al., 2017; Ma et al., 2018; Jia et al., 2022). Based on these findings, we used the larger threshold of 1.5.

These explanations have been summarized in the text added to section 2.2 (lines 252-260): "The choice of this threshold, rather than 0.6 used by Brendan et al. (2006), who used MOD06 Collection 04 products, is based on reports by Christenson et al. (2017) and (Varnái and Marshak, 2009). Christenson et al. (2017) used MOD06 C6 data (1km x1km) and reported that "large aerosol optical depths remain in the MODIS-observed pixels near cloud edges, due primarily to 3-D effects (Varnái and Marshak, 2009) and the swelling of aerosols by higher relative humidity." Varnái and Marshak (2009) noted that beyond 15 km contamination effects were minimized in MODIS data (1km x1km). Furthermore, we discarded scenes (1° by 1°) in which the aerosol distribution is heterogeneous, i.e. with a standard deviation higher than the mean value (Saponaro et al., 2017; Jia et al., 2022)."

12. L172 - Why is 200gm^-2 used as a threshold for the LWP?

**Answer**: In the text we added (line 261): "LWP larger than 200 g m$^{-2}$ is excluded to avoid deep convective clouds (Wang et al., 2014)".

13. L184 - I would suggest putting the URL links in the references or acknowledgements

**Answer**: Although it is nowadays quite common to provide url + last accessed data as a reference in the text, we have followed this suggestion.

14. L187 - ERA5 and ERA Interim both seem to be mentioned at different points in this work. I suggest using only one (preferably ERA5)

**Answer**: Indeed we used ERA-5 and have corrected this throughout the text.

15. L190 - I am not sure this is the definition of the first indirect effect as all of these properties also vary with cloud adjustments.

**Answer**: In response to your comments and those from other reviewers, we have changed the terminology to the terminology recommended in IPCC AR5 (see also our response to your specific point 3). As a result, we have change the title of Section 3.1 to "Sensitivity of cloud parameters to changes in aerosol concentrations" and the first sentence now reads "Changes in aerosol loading lead to an adjustment of cloud optical or microphysical parameters (COT, CER, etc.).", together with many other changes throughout the revised manuscript.

16. L192 - Ice nuclei are usually referred to as "ice nucleating particles" (INP) - Vali et al (2015)

**Answer**: Thank you for this suggestion: this has been changed here and elsewhere in the revised manuscript.

17. L204 - Do the authors mean CCN here (as in Andreae, 2009)

**Answer**: In the original formulation by Feingold et al (2001), α is the AOD. This relation was derived assuming that the cloud droplet number concentration $N_d$ varies with the aerosol number as $N_a{}^{a1}$(their eq. 5), with a1=0.7. Following Andreae (2009) there is a power law relation between AOD and CCN. We changed the sentence below eq. 1 (see lines 300-301 in the revised manuscript) to "Where $r_e$ represents the CER and $\alpha$ represents the AOD. Following Andreae (2009), AOD and CCN are correlated and AOD varies with CCN following a power law relationship.", while also changes were made to the rest of this paragraph.

18. L216 - An explicit list of these parameters, perhaps in the diagram, could be useful for others trying to replicate this study.

**Answer**: Thank you for this comment. We have added this information in Table 5 (Table 1 in the revised manuscript).

**Table 5. Parameters used in the present study, together with the sources, time periods and spatial resolutions.**

| Source | Time period | Resolution | Parameters |
|---|---|---|---|
| MYD08 | Jan 2008-Dec 2022 | Daily, 1°x1° | AOD at 550 nm |
| | | | COT at 2.1 um |
| | | | CER at 3.7 um and 2.1 um |
| | | | Cloud-top temperature |
| | | | Cloud-top pressure |
| | | | LWP at 2.1 um |
| | | | Cloud Fraction |
| | | | Solar zenith angle |
| | | | Sensor zenith angle |
| | | | Cloud multi-layer flag |
| | | | Cloud phase flag |
| ERA5 | Jan 2008-Dec 2022 | hourly, 0.25°x0.25° | Temperatures at 700 and 1000 hPa |
| | | | Relative humidity at 750 hPa |
| | | | Vertical velocity at 750 hPa |

19. L221 - I have not used the Jenks method before, but from what I understand, you have to specify the number of regions/regimes? How is this done and does the number of regions chosen affect the results?

**Answer**: The geographic detector model requires the input independent variable to be a type variable. The Jenks natural breaks classification method is one of many discretization methods and is commonly used in literature. The Jenks natural breaks classification method (Brewer and Pickle, 2002), aiming to minimize the variance within the group and maximize the variance between groups, was applied to categorize the whole region into n subregions. For example,

AOD needs to be classified into 5 levels using the Jenks natural breaks classification method, and the AOD source data needs to be reclassified into 1-5 natural numbers from small to large, and then counted into the grid. Therefore, the input of the independent variable AOD is a type variable. However, it should be noted that the GDM also has unstable characteristics. On the one hand, it is due to the MAUP (Modified Area Unit Problem) variable area unit problem, which can be understood as the influence of "scale effect". Due to the limitation of data resolution used in this study, the spatial statistical unit is 1°x1°. On the other hand, the methods used for data discretization can also have an impact. This study attempts to determine the optimal number of classifications by examining the impact of different classification numbers (3-8) on the GDM output results (as shown in Tables 6 and 7 below). The data shows that the classification number of regions does not affect the relative importance of cloud factors on the cloud. Here we classify the values of each cloud factor into 5 levels during the period of 2008-2022.

**Table 6. q values for factors which may influence cloud parameters over the ECS (9°x9°) in different number of classification levels (3~8) (see text) using Jenks natural breaks classification method, evaluated for data collected in the period from 2008-2022.**

| cloud parameters | number of classification levels | AOD | RH | LTS | PVV |
|---|---|---|---|---|---|
| CER | 3 | $0.80^{***}$ | $0.33^{**}$ | $0.42^{***}$ | $0.69^{***}$ |
| | 4 | $0.81^{***}$ | $0.40^{***}$ | $0.43^{***}$ | $0.67^{***}$ |
| | 5 | $0.81^{***}$ | $0.33^{**}$ | $0.44^{***}$ | $0.70^{***}$ |
| | 6 | $0.85^{***}$ | $0.41$ | $0.52^{***}$ | $0.73^{***}$ |
| | 7 | $0.83^{***}$ | $0.37$ | $0.44$ | $0.74^{***}$ |
| | 8 | $0.84^{***}$ | $0.40$ | $0.48^{**}$ | $0.70^{***}$ |
| COT | 3 | $0.66^{***}$ | $0.43$ | $0.42^{**}$ | $0.64^{***}$ |
| | 4 | $0.69^{***}$ | $0.45$ | $0.43$ | $0.66^{***}$ |
| | 5 | $0.69^{***}$ | $0.40$ | $0.38$ | $0.67^{***}$ |
| | 6 | $0.72^{***}$ | $0.47$ | $0.50$ | $0.72^{***}$ |
| | 7 | $0.75^{***}$ | $0.49$ | $0.43$ | $0.71^{***}$ |
| | 8 | $0.75^{***}$ | $0.48$ | $0.46$ | $0.68^{***}$ |
| LWP | 3 | $0.68^{***}$ | $0.18$ | $0.34^{***}$ | $0.57^{***}$ |
| | 4 | $0.67^{***}$ | $0.25$ | $0.37^{**}$ | $0.48^{***}$ |
| | 5 | $0.68^{***}$ | $0.23$ | $0.43^{***}$ | $0.49^{***}$ |
| | 6 | $0.72^{***}$ | $0.27$ | $0.44$ | $0.55^{***}$ |
| | 7 | $0.71^{***}$ | $0.30$ | $0.36$ | $0.59^{***}$ |
| | 8 | $0.75^{***}$ | $0.26$ | $0.45$ | $0.58^{***}$ |
| CF | 3 | $0.42^{***}$ | $0.19$ | $0.05$ | $0.46^{***}$ |
| | 4 | $0.46^{***}$ | $0.18$ | $0.07$ | $0.44^{***}$ |
| | 5 | $0.46^{***}$ | $0.20$ | $0.09$ | $0.47^{***}$ |
| | 6 | $0.47^{***}$ | $0.22$ | $0.07$ | $0.50^{***}$ |
| | 7 | $0.49^{***}$ | $0.19$ | $0.08$ | $0.56^{***}$ |
| | 8 | $0.49^{***}$ | $0.22$ | $0.09$ | $0.50^{***}$ |
| CTP | 3 | $0.47$ | $0.48$ | $0.24$ | $0.60$ |
| | 4 | $0.44$ | $0.56$ | $0.21$ | $0.58$ |
| | 5 | $0.47$ | $0.53$ | $0.18$ | $0.58$ |
| | 6 | $0.51$ | $0.56$ | $0.36$ | $0.69$ |
| | 7 | $0.50$ | $0.57$ | $0.27$ | $0.66$ |

| | | | | | |
|---|---|---|---|---|---|
| 8 | 0.51 | 0.58 | 0.26 | 0.65 |

Note: ***indicates that the q value is significant at the 0.01 level ($p < 0.01$), **indicates that the q value is significant at the 0.05 level ($p < 0.05$).

**Table 7. q values for factors which may influence cloud parameters over the YRD (9°x9°) in different number of classification levels (3~8) (see text) using Jenks natural breaks classification method, evaluated for data collected in the period from 2008-2022.**

| cloud parameters | number of classification levels | AOD | RH | LTS | PVV |
|---|---|---|---|---|---|
| CER | 3 | 0.22 | 0.14 | 0.01 | 0.12 |
| | 4 | 0.32 | 0.19 | 0.05 | 0.14 |
| | 5 | 0.31 | 0.25 | 0.13 | 0.18 |
| | 6 | 0.33 | 0.17 | 0.17 | 0.23 |
| | 7 | 0.34 | 0.25 | 0.17 | 0.15 |
| | 8 | 0.38 | 0.27 | 0.19 | 0.23 |
| COT | 3 | 0.52*** | 0.47** | 0.08 | 0.19 |
| | 4 | 0.53*** | 0.52*** | 0.10 | 0.31 |
| | 5 | 0.61*** | 0.45 | 0.12 | 0.29 |
| | 6 | 0.56** | 0.45 | 0.11 | 0.28 |
| | 7 | 0.60*** | 0.49 | 0.12 | 0.28 |
| | 8 | 0.59 | 0.54 | 0.15 | 0.32 |
| LWP | 3 | 0.17 | 0.35 | 0.52*** | 0.16 |
| | 4 | 0.17 | 0.34 | 0.54*** | 0.00 |
| | 5 | 0.16 | 0.32 | 0.55*** | 0.18 |
| | 6 | 0.18 | 0.34 | 0.55 | 0.21 |
| | 7 | 0.18 | 0.38 | 0.54** | 0.18 |
| | 8 | 0.23 | 0.37 | 0.55 | 0.20 |
| CF | 3 | 0.30*** | 0.01 | 0.34*** | 0.04 |
| | 4 | 0.37*** | 0.02 | 0.45*** | 0.03 |
| | 5 | 0.30*** | 0.02 | 0.50*** | 0.07 |
| | 6 | 0.39*** | 0.03 | 0.50*** | 0.09 |
| | 7 | 0.36*** | 0.05 | 0.58*** | 0.06 |
| | 8 | 0.38*** | 0.04 | 0.56*** | 0.10 |
| CTP | 3 | 0.49 | 0.72** | 0.26 | 0.48 |
| | 4 | 0.46 | 0.74*** | 0.35 | 0.52 |
| | 5 | 0.50 | 0.74*** | 0.32 | 0.56 |
| | 6 | 0.52 | 0.75 | 0.32 | 0.56 |
| | 7 | 0.55 | 0.79 | 0.38 | 0.57 |
| | 8 | 0.50 | 0.79 | 0.36 | 0.56 |

Note: ***indicates that the q value is significant at the 0.01 level ($p < 0.01$), **indicates that the q value is significant at the 0.05 level ($p < 0.05$).

20. Eq2 - I am not familiar with this method, so might need a bit more explanation. Is sigma here the variance of y within the specified region/regime?

**Answer**: Sigma $\sigma$ here is the standard deviation of y within the specified region/regime and $\sigma^2$ is the variance of y within the specified region/regime. This is specified in the text on page 13 of the revised manuscript (see lines 334-335): "and $\sigma_i^2$ and $\sigma^2$ denotes variance of samples in the subregion i and the total variance in the entire study area, respectively.".

21. Eq2 - How does this method compare to a more common correlation measure for non-linear relationships, such as Spearman's Rank?

**Answer**: Spearman's Rank analysis and GDM are two different statistical methods used to study the correlation and degree of influence between variables.

Spearman's Rank analysis is a non-parametric statistical method used to measure the correlation between two variables. It assesses the monotonic relationship between variables by only calculating the rank order of the variables.

GDM is a spatial statistical analysis method mainly used to study the spatial correlation and influencing factors between geographical phenomena. It can identify the dominant role, interaction, and non-linear effects of different factors on the spatial distribution of geographical phenomena. It not only accounts for the rank order of the variables but also spatial information.

The results of Spearman's Rank analysis are shown in Table 8 and Table 9 below. Over the ECS, the correlation coefficient $\rho$ between dependent a y variable (CER, COT, LWP) and an independent x variable (AOD, RH, LTS and PVV) are highest for AOD and following by PVV, LTS and RH. For CF and CTP, the correlation coefficient $\rho$ is highest for PVV, followed by AOD, RH and LTS. The orders of correlation coefficient $\rho$ are consistent with that of GDM q values (Table 5 in the revised manuscript). Over the YRD, for the CF the orders of correlation coefficient $\rho$ are different from that of GDM q values (Table 6 in the revised manuscript). It shows that the correlation coefficient $\rho$ is lowest for LTS but the GDM q value is highest for LTS. It may be attribute to that GDM not only accounts for the rank order of the variables as determined by the Spearman's Rank method but also spatial information.

**Table 8. Statistics of Spearman's Rank analysis between x (AOD and meteorological conditions) and y (cloud parameters) over the ECS during 2008-2022. Statistically significant data points are indicated with \*\*\* (p value < 0.01)**

| Cloud parameter | AOD | RH | LTS | PVV |
|---|---|---|---|---|
| CER | -0.92\*\*\* | 0.61\*\*\* | 0.65\*\*\* | -0.83\*\*\* |
| COT | 0.85\*\*\* | -0.63\*\*\* | -0.63\*\*\* | 0.83\*\*\* |
| LWP | -0.85\*\*\* | 0.48\*\*\* | 0.59\*\*\* | -0.71\*\*\* |
| CF | 0.65\*\*\* | -0.46\*\*\* | -0.23\*\* | 0.71\*\*\* |
| CTP | -0.70\*\*\* | 0.73\*\*\* | 0.37\*\*\* | -0.81\*\*\* |

**Table 9. Statistics of Spearman's Rank analysis between x (AOD and meteorological conditions) and y (cloud parameters) over the YRD during 2008-2022. Statistically significant data points are indicated with * (p value < 0.01)**

| Cloud parameter | AOD | RH | LTS | PVV |
|---|---|---|---|---|
| CER | 0.40*** | -0.36*** | -0.09 | 0.25** |
| COT | -0.76*** | 0.63*** | -0.19 | -0.42*** |
| LWP | -0.35*** | 0.59*** | -0.63*** | -0.44*** |
| CF | -0.49*** | 0.30*** | 0.26** | -0.32*** |
| CTP | 0.72*** | -0.85*** | 0.48*** | 0.71*** |

22. L379 - The p-value for testing here is quoted as 0.01, but elsewhere it appears that 0.1 (a fairly lax criteria) is used.

**Answer**: Done. We have made unified standards that the p-value for testing here is quoted as 0.01 through the revised manuscript.

23. L422 - The high explanatory power of AOD for CF variations suggests that this method is not actually identifying causal relationships. While strong correlations between AOD and CF have been previously observed, they are likely due to aerosol humidification (Quaas et al, 2010), rather than an aerosol influence. It seems likely the same effect is being observed here, so care should be taken in the presentation of the results not to mis-attribute causality (unless applicable).

**Answer**: This study provides a general description of the sensitivity (S) of cloud parameters to the influence of different aerosol and meteorological parameters over YRD and ECS. Correlations between AOD and cloud parameters are found over the target regions, which can be attributed in part to the influence of general circulation. In general, there are many relations between the various parameters, both related to cloud microphysics and meteorology. It is not possible to completely separate meteorological influences from aerosol influences on clouds. This work can therefore only provide further evidence of the aerosol and meteorological effects on clouds and quantify the relative contributions and combined effects on clouds, but cannot quantify the absolute contributions with confidence. Thus, establishing cause and effect relationships between parameters is difficult and must be made with care.

We have added the following text in the Sect 4.6.1 (lines 647 to 666): "Tables 5 and 6 list q values for individual factors, together with p showing the absence of statistical significance in many cases, especially over the YRD, and often the explanatory power is not high when the significance is low. These data show that cloud parameters are dominated by aerosol effects over the ECS but meteorological influences on cloud parameters predominate over the YRD, as was also concluded from the analysis from "traditional" statistical methods presented in Section 4.5 and these conclusions are consistent with the results published by Andersen and Cermak (2015). Among the meteorological parameters, we also find that PVV (with highest q in the three meteorological parameters) predominantely influences cloud parameters over the ECS. Jones et al. (2009) and Jia et al. (2022) reported that stronger aerosol cloud interactions typically occur under higher updraft velocity conditions. In addition, we find that CTP is mainly affected by RH (q = 0.74***) and PVV (q = 0.56) over the YRD, as suggested by Koren et al. (2010). Koren et al. reported that observed cloud top height correlates best with model pressure updraft velocity and relative humidity. To some extent, LTS influences CER (q = 0.44***) and LWP (q = 0.43***) over the ECS, while, in contrast, over the YRD LTS

predominately influences CF (q = 0.50***) and LWP (q = 0.55***). Matsui et al. (2004) and Tan et al. (2017) reported that aerosol impact on CER is stronger in more dynamic environments that feature a lower LTS and argue that very high LTS environments dynamically suppress cloud droplet growth and reduce aci intensity. While strong correlations between AOD and cloud parameters have been previously observed, they are likely due to the swelling of aerosol particles in humid airmasses (Quaas et al, 2010), rather than an aerosol influence, which is in agreement with findings by, e.g., Myhre et al. (2007), Twohy et al. (2009) and Quaas et al. (2010).".

We have also added the following text in the Sect 4.6.2 (lines 652 to 655 and lines 695-699): "Among the meteorological parameters, we find that the combined effect of AOD and PVV predominately influences on cloud parameters over the ECS. The result is in accord with the finding of Jones et al. (2009) and Jia et al. (2022) that stronger aerosol cloud interactions typically occur under higher updraft velocity conditions." and "The results from the GDM interaction detector analysis clearly show the enhancement of the interaction q-values over the q-values for the individual factors. In other words, the explanatory power of the combined effects of aerosol and a meteorological parameter is larger than that of each parameter alone. Thus, the GDM provides an alternative way to obtain information on confounding effects of different parameters.".

24. L469 - After the introduction of the GDM, sections 4.5 and 4.6 appear to go back to more 'traditional' methods as used by previous paper. I am not sure I really see how these section support the paper in determining the cause of the different ACI values in these regions. It would be good to have a clearer link to the other work performed and how it supports the overall aim and conclusions of the paper.

**Answer**: In the revised version, we have moved Sections 4.5 (now 4.4.) and 4.6 (now 4.5) before Section 4.4 (now 4.6). Thus, we first discuss findings from "traditional" methods, followed by findings using the GDM. We have also added Section 5 (Discussion) and Section 6 (Conclusions) where we discuss the different findings using "traditional" methods and GDM, with more emphasis on the added value of GDM.

25. L601 - Could the authors be more specific on how this study will help improve model parametrisations?

**Answer**: Aerosol particles, acting as cloud condensation nuclei, affect the number and size of cloud droplets. The link between aerosol and the formation and properties of clouds could better simulate changes in cloud parameters. By comparing with observational data of aerosols and clouds, the model's ability to simulate changes in cloud parameters can be evaluated. Meteorological factors are key influencing parameters for the formation and evolution of clouds, and a more accurate description of the relative contribution of meteorological factors can improve the parameterization scheme of the model. Therefore, by more accurately simulating and predicting the impact of aerosols and meteorological parameters on clouds, parameterization schemes will be adjusted and improved, which further improve the simulation ability and accuracy of climate models for cloud parameter changes.

The text in the Conclusion has been reorganized as "By comparison with aerosol and cloud observations, the regional climate model's ability to simulate changes in cloud parameters can be evaluated. A more accurate description of the relative contribution of meteorological factors

can improve the parameterization scheme of the model over eastern China." in the revision manuscript (lines 821-824).

**References**

We thanks referee#1 for providing these excellent references. They have been used in the manuscript and most of them have been quoted or summarized when appropriate. We have added some more references in the reference list of the revised manuscript.

Andersen, H., & Cermak, J. (2015). How thermodynamic environments control stratocumulus microphysics and interactions with aerosols. Environmental Research Letters, 10, 024004. https://doi.org/10.1088/1748-9326/10/2/024004

Andreae, M. O. (2009). Correlation between cloud condensation nuclei concentration and aerosol optical thickness in remote and polluted regions. Atmospheric Chemistry and Physics, 9(2), 543–556. https://doi.org/10.5194/acp-9-543-2009

Boucher, O., & Quaas, J. (2012). Water vapour affects both rain and aerosol optical depth. Nature Geoscience, 6(1), 4–5. https://doi.org/10.1038/ngeo1692

Grandey, B. S., & Stier, P. (2010). A critical look at spatial scale choices in satellite-based aerosol indirect effect studies. Atmospheric Chemistry and Physics, 10(23), 11459–11470. https://doi.org/10.5194/acp-10-11459-2010

Gryspeerdt, E., Stier, P., & Grandey, B. S. (2014). Cloud fraction mediates the aerosol optical depth-cloud top height relationship. Geophysical Research Letters, 41, 3622–3627. https://doi.org/10.1002/2014GL059524

Gryspeerdt, Edward, Quaas, J., Ferrachat, S., Gettelman, A., Ghan, S., Lohmann, U., et al. (2017). Constraining the instantaneous aerosol influence on cloud albedo. Proceedings of the National Academy of Sciences of the United States of America, 114(19), 4899–4904. https://doi.org/10.1073/pnas.1617765114

Hasekamp, O. P., Gryspeerdt, E., & Quaas, J. (2019). Analysis of polarimetric satellite measurements suggests stronger cooling due to aerosol-cloud interactions. Nature Communications, 10(1). https://doi.org/10.1038/s41467-019-13372-2

Jia, H., Quaas, J., Gryspeerdt, E., Böhm, C., & Sourdeval, O. (2022). Addressing the difficulties in quantifying droplet number response to aerosol from satellite observations. Atmospheric Chemistry and Physics, 22(11), 7353–7372. https://doi.org/10.5194/acp-22-7353-2022

Jones, T. A., Christopher, S. A., & Quaas, J. (2009). A six year satellite-based assessment of the regional variations in aerosol indirect effects. Atmospheric Chemistry and Physics, 9, 4091.

Koren, I., Feingold, G., & Remer, L. A. (2010). The invigoration of deep convective clouds over the Atlantic: aerosol effect, meteorology or retrieval artifact? Atmospheric Chemistry and Physics, 10(18), 8855–8872. https://doi.org/10.5194/acp-10-8855-2010

McComiskey, A., & Feingold, G. (2012). The scale problem in quantifying aerosol indirect effects. Atmospheric Chemistry and Physics, 12, 1031. https://doi.org/10.5194/acp-12-1031-2012

Myhre, G., Stordal, F., Johnsrud, M., Kaufman, Y. J., Rosenfeld, D., Storelvmo, T., et al. (2007). Aerosol-cloud interaction inferred from MODIS satellite data and global aerosol models. Atmospheric Chemistry and Physics, 7(12), 3081–3101. https://doi.org/10.5194/acp-7-3081-2007

Quaas, J., Stevens, B., Stier, P., & Lohmann, U. (2010). Interpreting the cloud cover – aerosol optical depth relationship found in satellite data using a general circulation model. Atmospheric Chemistry and Physics, 10(13), 6129–6135. https://doi.org/10.5194/acp-10-6129-2010

Tan, S., Han, Z., Wang, B., & Shi, G. (2017). Variability in the correlation between satellite-derived liquid cloud droplet effective radius and aerosol index over the northern Pacific Ocean. Tellus B: Chemical and Physical Meteorology, 69(1), 1391656. https://doi.org/10.1080/16000889.2017.1391656

Tang, J., Wang, P., Mickley, L. J., Xia, X., Liao, H., Yue, X., et al. (2014). Positive relationship between liquid cloud droplet effective radius and aerosol optical depth over Eastern China from satellite data. Atmospheric Environment, 84, 244–253. https://doi.org/10.1016/j.atmosenv.2013.08.024

Vali, G., DeMott, P. J., Möhler, O., & Whale, T. F. (2015). Technical Note: A proposal for ice nucleation terminology. Atmospheric Chemistry and Physics, 15(18), 10263–10270. https://doi.org/10.5194/acp-15-10263-2015

Yuan, T., Li, Z., Zhang, R., & Fan, J. (2008). Increase of cloud droplet size with aerosol optical depth: An observation and modeling study. Journal of Geophysical Research, 113(D4). https://doi.org/10.1029/2007JD008632

---

## Author Comment (AC2)

**Response to Referee #3**

This study investigates the aerosol and meteorological parameters on warm clouds using satellite measurements. The authors focus on the period of 2008-2022 over the two contrasting regions over eastern China, i.e. Yangtze River Delta, a heavily polluted region in eastern China, and the East China Sea with a relatively clean atmosphere. The interaction between AOD and CER has been investigated by considering different AOD and LWP regimes in the both two different aerosol regimes. A new method (geographical detector method) was applied to explore the relative importance of AOD and meteorological parameters on cloud properties. The content of this manuscript is highly relevant to ACP readers. In general, the manuscript is well organized, and the analysis conducted is quite comprehensive. Based on the overall quality, it is recommended that the manuscript be considered for publication if the specific comments provided are addressed.

The authors thank Referee #3 for the valuable time spent on thorough reading our manuscript and providing expert views to guide us for improving the manuscript with the specific comments and a reference. We have taken notice of all comments, listed below in black, and made many changes to the manuscript to address these, together with the comments from the other referees. We address each of your comments below and refer to our responses in the revised manuscript and provide line numbers and copy text in "quotes".

To ensure that the data used only included single layer liquid clouds and nonprecipitating cases, the filtering criteria described by Saponaro et al. (2017) were applied. It is noted that all the figures have been updated throughout the revised manuscript.

**Specific comments**

1. Abstract: it would be beneficial if the authors emphasized the overall significance or implications of their study at the end of the abstract.

**Answer**: We have substantially revised the abstract and added to following sentence upfront, to provide the overall picture "The sensitivity (S) of cloud parameters to the influence of different aerosol and meteorological parameters has in most previous aerosol-cloud interaction (aci) studies been addressed using traditional statistical methods. In the current study, relationships between cloud droplet effective radius (CER) and aerosol optical depth (AOD, used as a proxy for cloud condensation nuclei, CCN), i.e. the sensitivity (S) of CER to AOD, is investigated with different constraints of AOD and cloud liquid water path (LWP). In addition to traditional statistical methods, the geographical detector method (GDM) has been applied to quantify the relative importance of the effects of aerosol and meteorological parameters, and their interaction, on S.". In addition, many other changes were made to the abstract in "track changes".

2. In order to provide a more comprehensive analysis, it would be beneficial for the authors to compare the results obtained in this study with findings from other regions around the world. By doing so, they can examine the unique aerosol effects on clouds in the specific target region.

**Answer**: In the revised text, the results are compared with many other findings. We have added the following text in the Sect 4.3 (lines 528 to 534): "The variation of S with changes in LWP indicates that the condition of constant LWP is not truly satisfied: if the data would be stratified according to smaller LWP intervals (quasi-constant LWP, Ma et al., 2018), S would likely vary

more smoothly with LWP. As mentioned in the Introduction, LWP is not directly retrieved but calculated form CER and COT and thus also the calculation of S is to some extend affected by LWP. We further note the results by Ma et al. (2018), i.e. the slope of CER versus AI (comparable to S in this paper) varies little with LWP, with positive values over land and negative values over ocean and thus behaves similar to the data in Table 3 for YRD and ECS."

We have also added the following text in the Sect 4.3 (lines 647 to 666): "Tables 5 and 6 list q values for individual factors, together with p showing the absence of statistical significance in many cases, especially over the YRD, and often the explanatory power is not high when the significance is low. These data show that cloud parameters are dominated by aerosol effects over the ECS but meteorological influences on cloud parameters predominate over the YRD, as was also concluded from the analysis from "traditional" statistical methods presented in Section 4.5 and these conclusions are consistent with the results published by Andersen and Cermak (2015). Among the meteorological parameters, we also find that PVV (with highest q in the three meteorological parameters) predominately influences cloud parameters over the ECS. Jones et al. (2009) and Jia et al. (2022) reported that stronger aerosol cloud interactions typically occur under higher updraft velocity conditions. In addition, we find that CTP is mainly affected by RH (q = $0.74^{***}$) and PVV (q = 0.56) over the YRD, as suggested by Koren et al. (2010). Koren et al. reported that observed cloud top height correlates best with model pressure updraft velocity and relative humidity. To some extent, LTS influences CER (q = $0.44^{***}$) and LWP (q = $0.43^{***}$) over the ECS, while, in contrast, over the YRD LTS predominately influences CF (q = $0.50^{***}$) and LWP (q = $0.55^{***}$). Matsui et al. (2004) and Tan et al. (2017) reported that aerosol impact on CER is stronger in more dynamic environments that feature a lower LTS and argue that very high LTS environments dynamically suppress cloud droplet growth and reduce aci intensity. While strong correlations between AOD and cloud parameters have been previously observed, they are likely due to the swelling of aerosol particles in humid airmasses (Quaas et al, 2010), rather than an aerosol influence, which is in agreement with findings by, e.g., Myhre et al. (2007), Twohy et al. (2009) and Quaas et al. (2010)."

We have also added the following text in the Discussion (lines 747 to 756): "It is noticed that in recent papers (e.g., Gryspeerdt et al., 2023; Arola et al., 2022) the usefulness of correlating aerosol and cloud parameters has been seriously challenged because cloud variability and retrieval errors are such that correlations between AOD and cloud properties ($N_d$, CER, LWP) can be spurious. Gryspeerdt et al. (2023) discussed aci in terms of the susceptibility β of Nd to aerosol rather than the sensitivity S of CER to aerosol (see the discussion in the Introduction on the use of $N_d$ vs CER), and the problem arises with low aerosol conditions due to larger aerosol retrieval uncertainty due to surface correction (larger surface effect on the radiance at the top of the atmosphere), which applies equally to β and S. In the current study we did not consider the lowest aerosol conditions by limiting the data to situations with AOD ≥ 0.1, as discussed in Section 4.2. Furthermore, we stratified the analysis for moderate (0.1 ≤ AOD < 0.3) and high (0.3 ≥ AOD) aerosol regimes, based on the data."

This text is followed by the discussion of the implications of the findings of Arola et al (2022) for our results (lines 757-777): "Arola et al. (2022) addressed the susceptibility of $N_d$ to changes in aerosol and the adjustment of LWP (using satellite observations), and confounding factors, in particular co-variability of $N_d$ and LWP induced by meteorological effects. They show how errors in the retrieved CER and COT or spatial heterogeneity in cloud fields influence the $N_d$ - LWP relation. However, both $N_d$ and LWP are not retrieved but derived from CER and COT. Using Eq. 1 and Eq. 2 in Arola et al. (2022), the $N_d$ -LWP relationship can be shown to have a

highly non-linear dependence on CER and thus it is no surprise that any error in CER strongly affects the relation between $N_d$ and LWP. Their experiments, i.e. using smaller scales (5° x 5°) to reduce spatial meteorological variability, or using snapshots to remove meteorological variability in time, did not lead to a conclusion whether the $N_d$ - LWP variability is due to spatial heterogeneity in the cloud fields or due to retrieval errors. The main message from this part of the study (using satellite data) by Arola et al. (2022) is "the spatial variability of CER introduces a bias which moreover becomes stronger in conditions where the CER values are lower on average". Experiments with simulated measurements show that "the main cause of the negative LWP vs $N_d$ slopes is the error in CER". Arola et al. emphasize that the spatial cloud variability and retrieval errors in CER and COT are similar sources for negative bias in LWP adjustment and that these sources could not be separately assessed in their simulations. The implication of the findings of Arola et al. (2022) on the adjustment of LWP for the results of the current study on the sensitivity of CER to aerosol (or CCN, using AOD as proxy) is that the assumption of constant LWP may be violated. This would affect the results presented in Section 4.3 where LWP was stratified and S was found to vary with LWP. In view of the LWP adjustment to changes in aerosol, the variation of CER sensitivity with LWP may be somewhat different from that reported in section 4.3."

3. Page 5,line 146:add "." in the end.

Thank you: done

4. Page 6,line 150:change "Eastern China Sea (ECS) area (20°N-28°N, 126°E-134°E)" to "Eastern China Sea area (ECS, 20°N-28°N, 126°E-134°E)"

Thank you: done

5. Page 7,line 191:change "(cloud optical thickness, cloud droplet effective radius, etc.)" to "(COT, CER, etc.)".

Thank you: done

6. Page 8, line 202: change "Where re represents the cloud droplet effective radius (CER)" to "Where re represents the CER".

Thank you: done

7. Page 11, line 276: it suggests to define the acronyms about "NW" and "SW".

Thank you. We feel that the use of wind directions is common in geographical descriptions of spatial variation and since we refer here to the maps in Fig. 4, it is not necessary to define the abbreviations for the wind directions. However, after reading the relevant text again, we noticed that we have used the full names for other wind directions (like south) and therefore decided to write them in full throughout the manuscript, i.e. we replace NW with northwest etc.

8. Page 14, line 344: remove right parenthesis.

Thank you: done

9. Page 15, line 365: replace "for the YRD" with "over the YRD".

Thank you: done

10. Page 21, line 501: add right parenthesis after "by Liu et al., (2017".

Thank you: done

11. Page 22, line 516: change "for three different LWP intervals" to "for five different LWP intervals".

Thank you: done

12. Page 16, line 377-380︰ The statistically significance is used through the manuscript, so it suggests to describe at the first place in the manuscript.

**Answer**: We have added the following text in the Sect 3.1 (lines 307 to 310): "The significance of these relations is determined by using the student's t test, i.e. the results are statistically significant when the p value is smaller than 0.01, where p is defined as the probability of obtaining a result equal to or "more extreme" than what was actually observed.".

**References**

Ma, X., Jia, H., Yu, F., & Quaas, J. (2018). Opposite aerosol index-cloud droplet effective radius correlations over major industrial regions and their adjacent oceans. Geophysical Research Letters, 45, 5771–5778. https://doi.org/10.1029/2018GL077562

---

## Author Comment (AC3)

**Response to Referee #2**

This study uses correlation and the geographical detector method (GDM) to study relationships between aerosol optical depth (AOD), meteorological indicators, and cloud properties, contrasting a heavily polluted region in mainland China to a cleaner region of the Pacific Ocean influenced by transported pollution. The authors find different signs of the AOD-cloud effective radius relationships and find that AOD explains a very large fraction of variability in cloud properties, especially in the cleaner region.

The manuscript is well written, and the Figures and Tables illustrate the results and discussion well. But I have a mixed opinion about the study. On the one hand, there are innovative aspects, such as the application of the GDM to the aerosol-cloud problem. On the other hand, the study uses large-scale, time-averaged correlations of aerosols and clouds that have been shown to say little about aerosol-cloud interactions. And the impact of AOD on cloud variability that results from the GDM is so large that it would require a strong case to bring confidence in the method and results. On balance, I suggest major revisions to give the authors the chance to justify their results.

The authors thank Referee #2 for the valuable time spent on thorough reading our manuscript and providing expert views to guide us for improving the manuscript with the main and other comments. We have taken notice of all comments, listed below in black, and made many changes to the manuscript to address these, together with the comments from the other referees. We address each of your comments below and refer to our responses in the revised manuscript and provide line numbers and copy text in "quotes".

To ensure that the data used only included single layer liquid clouds and nonprecipitating cases, the filtering criteria described by Saponaro et al. (2017) were applied. It is noted that all the figures have been updated throughout the revised manuscript.

Main comments:

1. Recent papers, and especially Gryspeerdt et al. (2023, 10.5194/acp-23-4115-2023) and Arola et al. (2022, 10.1038/s41467-022-34948-5), have seriously challenged the usefulness of correlating aerosol and cloud parameters as done in the present study. Cloud variability and retrieval errors are such that correlations between aerosol optical depth and cloud properties (CNDC, CER, LWP) can in fact be spurious. That means that a large fraction of past literature on aerosol-cloud interactions (including the studies cited lines 86-107) needs to be look at again critically. Attempts to minimise retrievals uncertainties (lines 172-175) will not address parts of the issues. I think the present study remains interesting (especially the GDM analysis), but the authors need to acknowledge the possibility that the correlations they find do not say much about aerosol-cloud interactions.

**Answer**: Gryspeerdt et al. (2023) discuss aci in terms of $N_d$ rather than CER and emphasize the importance of the susceptibility $\beta$ of $N_d$ to aerosol. The main interest of their study is to determine $RF_{aci}$ and the importance of accurate determination of $\beta$. The variation in $\beta$ is responsible for much of the uncertainty in $ERF_{aci}$ in climate models and $\beta$ is central to the strength of cloud adjustments. Gryspeerdt et al. point out that uncertainties and differences occur with low aerosol conditions where satellite derived values of $\beta$ are uncertain due to retrieval assumptions and separation of the (weak) aerosol signal from the surface reflectance. In high aerosol conditions, the (stronger) aerosol signal is relatively larger than the surface

reflectance rendering more accurate aerosol retrieval in polluted conditions than in clean conditions. The larger uncertainty in clean condition reduces the correlation between CCN and the retrieved AOD due to regression dilution and thus reduces the magnitude of β in clean conditions. In the discussion, Gryspeerdt et al. argue that, for observational studies, the aerosol-$N_d$ relationship is non-linear and the value of β determined for high aerosol conditions is not necessarily "a good guide" for β in low aerosol conditions. Furthermore, they argue that β in high conditions is more likely an underestimate.

In our study we used CER rather than $N_d$, for reasons discussed in the Introduction (see lines 177-183 in the revised manuscript): "It is noted that $RF_{aci}$ is formulated in terms of $N_d$, whereas studies on the Twomey effects often use CER instead of $N_d$. CER is readily available as a satellite retrieval product, although in particular over land the reliability is questioned (Grandey and Stier, 2010), whereas $N_d$ is derived from CER and the cloud optical thickness (COT) (e.g., Grandey and Stier, 2010; Arola et al., 2022). This implies that $N_d$ is subject to the same retrieval errors as CER, including a possible relation between CER and LWP. The comparison of global maps of the sensitivities of CER and $N_d$ to AOD by Grandey and Stier (2010) exhibits very similar patterns.", and we stratified by aerosol regime. We acknowledge the findings of Gryspeerdt et al. (2023) and possible consequences to our results with the following text added in Section 5, Discussion (lines 747-756):

"It is noticed that in recent papers (e.g., Gryspeerdt et al., 2023; Arola et al., 2022) the usefulness of correlating aerosol and cloud parameters has been seriously challenged because cloud variability and retrieval errors are such that correlations between AOD and cloud properties ($N_d$, CER, LWP) can be spurious. Gryspeerdt et al. (2023) discussed aci in terms of the susceptibility β of Nd to aerosol rather than the sensitivity S of CER to aerosol (see the discussion in the Introduction on the use of $N_d$ vs CER), and the problem arises with low aerosol conditions due to larger aerosol retrieval uncertainty due to surface correction (larger surface effect on the radiance at the top of the atmosphere), which applies equally to β and S. In the current study we did not consider the lowest aerosol conditions by limiting the data to situations with AOD ≥ 0.1, as discussed in Section 4.2. Furthermore, we stratified the analysis for moderate (0.1 ≤ AOD < 0.3) and high (0.3 ≥ AOD) aerosol regimes, based on the data."

This text is followed by the discussion of the implications of the findings of Arola et al (2022) for our results (lines 757-777): "Arola et al. (2022) addressed the susceptibility of $N_d$ to changes in aerosol and the adjustment of LWP (using satellite observations), and confounding factors, in particular co-variability of $N_d$ and LWP induced by meteorological effects. They show how errors in the retrieved CER and COT or spatial heterogeneity in cloud fields influence the $N_d$ - LWP relation. However, both $N_d$ and LWP are not retrieved but derived from CER and COT. Using Eq. 1 and Eq. 2 in Arola et al. (2022), the $N_d$ -LWP relationship can be shown to have a highly non-linear dependence on CER and thus it is no surprise that any error in CER strongly affects the relation between $N_d$ and LWP. Their experiments, i.e. using smaller scales (5° x 5°) to reduce spatial meteorological variability, or using snapshots to remove meteorological variability in time, did not lead to a conclusion whether the $N_d$ - LWP variability is due to spatial heterogeneity in the cloud fields or due to retrieval errors. The main message from this part of the study (using satellite data) by Arola et al. (2022) is "the spatial variability of CER introduces a bias which moreover becomes stronger in conditions where the CER values are lower on average". Experiments with simulated measurements show that "the main cause of the negative LWP vs $N_d$ slopes is the error in CER". Arola et al. emphasize that the spatial cloud variability and retrieval errors in CER and COT are similar sources for negative bias in LWP adjustment and that these sources could not be separately assessed in their simulations.

The implication of the findings of Arola et al. (2022) on the adjustment of LWP for the results of the current study on the sensitivity of CER to aerosol (or CCN, using AOD as proxy) is that the assumption of constant LWP may be violated. This would affect the results presented in Section 4.3 where LWP was stratified and S was found to vary with LWP. In view of the LWP adjustment to changes in aerosol, the variation of CER sensitivity with LWP may be somewhat different from that reported in section 4.3."

2. The idea of using the GDM is interesting, but it is difficult to make physical sense of the results (Table 4 and 5). First, the q factors do not sum up to 1. What does that mean that AOD explains 87% of CER variability, while RH explains a further 36%? Then, the size of the q factors for AOD in Table 4, and to a lesser extent Table 5, stretches belief. If aerosols were that important in determining cloud properties, then estimating aerosol-cloud radiative forcing would have been very easy. Clearly, something goes wrong here. Is it perhaps that the four variables studied (AOD, RH, LTS, and PVV) are not independent? That the large q-values are simply a symptom of correlations caused by atmospheric circulation in the ECS and YRD? This is essentially what Figure 10 suggests. A strong discussion is needed to support the results.

**Answer**: A similar comment was made by Referee#1, i.e. that the q-values sum up to over 100% and about the interpretation of the q-values when the variables are not independent. Indeed, due to the interaction, the influence of a parameter may be strengthened or weakened and therefore the influence of different parameters need to be considered together. Therefore, our response is the same as that to Referee#1 Main point 1, while we also refer to the second part of our answer to your comment 3.

In statistics, the q-value is a measure used to evaluate the explanatory power of variables on the dependent variable. When multiple independent variables are considered separately, it is indeed possible for the sum of the q-values of multiple X variables to exceed 100%. When they are considered together, this is referred to as 'interaction q-value'. This situation is quite common and similar to the issue in multiple linear regression. The main reason for this is the presence of correlation among the X variables, indicating that these variables are not independent. Consequently, multiple independent variables may contribute to the dependent variable in a similar manner, leading to a sum of q-values over 100%.

To better explain this and clarify "interaction detector" and "interaction q-values", we have replaced the text below figure 2 (lines 354-374) with "The interaction detector can be used to test for the influence of interaction between different influencing factors, e.g., x1 and x2, on the dependent factor (y) and whether this interaction weakens or enhances the influence of each of x1 or x2 on the dependent variable, y, or whether they are independent in influencing y. For example, Figure 3(a) shows the spatial distribution of the dependent variable, y. The factors x1 and x2 both vary across the study region, but in different ways, and for each factor different sub-regions can be distinguished by application of the Jenks classification method described above to each factor separately. This is illustrated in Figures 3(b) and 3(c) where, as an example, three different sub-regions are considered for each factor. Usually, the dependent variable y is influenced by several different factors xi (Figure 3) and the combined effect of two or more factors may have a weaker or stronger influence on y than each of the individual factors. The q values for the influences of factors x1 and x2 on y, obtained from the application of the factor detector method (Eq. 2), may be represented as q (x1) and q (x2). Hence, a new spatial unit and subregions may be generated by overlaying the factor strata x1 and x2, written as x1∩x2, where ∩ denotes the interaction between factor strata x1 and x2 as illustrated in Figure 3(d).

Thus, the q value of the interaction of x1∩x2 may be obtained, represented as q (x1∩x2). Comparing the q value of the interaction of the pair of factors and the q value of each of the two individual factors, five categories of the interaction factor relationship can be considered which are summarized in Table 2. If $q(x1∩x2) > q(x1) + q(x2)$, this is referred to as a nonlinear enhancement of two variables. And if $q(x1∩x2) > Max[q(x1), q(x2)]$, this is referred to as a bilinear enhancement of two variables. The occurrence of nonlinear enhancement and bilinear enhancement are indicated with the q values in Table 2 and in the caption of Figure 7.".

GDM is a spatial statistical analysis method aimed at studying the degree of influence and spatial patterns of different factors on the changes in geographic phenomena. In the analysis, we can simultaneously consider the interactions and impacts among multiple factors, thus revealing the relationships of synergistic changes. Therefore, geographic detector can encompass the analysis of synergistic changes.

We have added the following text in the Sect 4.6.1 (lines 647 to 666): "Tables 5 and 6 list q values for individual factors, together with p showing the absence of statistical significance in many cases, especially over the YRD, and often the explanatory power is not high when the significance is low. These data show that cloud parameters are dominated by aerosol effects over the ECS but meteorological influences on cloud parameters predominate over the YRD, as was also concluded from the analysis from "traditional" statistical methods presented in Section 4.5 and these conclusions are consistent with the results published by Andersen and Cermak (2015). Among the meteorological parameters, we also find that PVV (with highest q in the three meteorological parameters) predominantely influences cloud parameters over the ECS. Jones et al. (2009) and Jia et al. (2022) reported that stronger aerosol cloud interactions typically occur under higher updraft velocity conditions. In addition, we find that CTP is mainly affected by RH ($q = 0.74^{***}$) and PVV ($q = 0.56$) over the YRD, as suggested by Koren et al. (2010). Koren et al. reported that observed cloud top height correlates best with model pressure updraft velocity and relative humidity. To some extent, LTS influences CER ($q = 0.44^{***}$) and LWP ($q = 0.43^{***}$) over the ECS, while, in contrast, over the YRD LTS predominately influences CF ($q = 0.50^{***}$) and LWP ($q = 0.55^{***}$). Matsui et al. (2004) and Tan et al. (2017) reported that aerosol impact on CER is stronger in more dynamic environments that feature a lower LTS and argue that very high LTS environments dynamically suppress cloud droplet growth and reduce aci intensity. While strong correlations between AOD and cloud parameters have been previously observed, they are likely due to the swelling of aerosol particles in humid airmasses (Quaas et al, 2010), rather than an aerosol influence, which is in agreement with findings by, e.g., Myhre et al. (2007), Twohy et al. (2009) and Quaas et al. (2010)."

This study provides a general description of the sensitivity (S) of cloud parameters to the influence of different aerosol and meteorological parameters over the YRD and the ECS. Correlations between AOD and cloud parameters are found over the target regions, which can be attributed in part to the influence of general circulation. In general, there are many relations between the various parameters, both related to cloud microphysics and meteorology. Thus, establishing cause and effect relationships between parameters is difficult and must be made with care. It is not possible to completely separate meteorological influences from aerosol influences on clouds. This work can therefore only provide further evidence of the aerosol and meteorological effect on clouds and quantify the relative contributions and combined effects on clouds, but cannot quantify the absolute contributions with confidence.

In the current study, based on a regional scale of 9°x9°, the GDM method is used to explore the relative importance of various factors on cloud parameters and identify possible correlations between different factors. In the future, aerosol cloud interactions can be studied on smaller regional scale (<4°x4°) using higher resolution source data.

3. The GDM assumes that the spatial distributions of independent and dependent variables "should have evident similarities" [line 211]. But at what scale is that assumption true for aerosol-cloud interactions? One could expect the assumption to break down when going down to the scale of a cloud field because clouds evolve after their aerosol-influenced formation phase. Is that a problem?

**Answer**: Spatially-varying aerosol and cloud properties may contribute towards observed relationships between aerosol and cloud properties. This may affect the results of many of the aforementioned studies which analyze data on a relatively large regional scale. Aerosol type, cloud regime and synoptic regime may vary over large spatial scales. If data are analyzed for the region as a whole, false correlations may be introduced. Grandy and Stier (2010) suggested that for region sizes larger than 4°x4°, spurious spatial variations in retrieved cloud and aerosol properties can introduce widespread significant errors to calculation S. However, we can observe that at the regional scales of 8°x8° and 15°x15°, although significant errors are introduced, the spatial distribution patterns of S (the sensitivities of CER and $N_d$ to AOD) look very similar, as shown in Figure 2 of Grandy and Stier (2010).

GDM is a spatial statistical analysis method aimed at studying the degree of influence and spatial patterns of different factors on the changes in geographic phenomena. In the analysis, we can simultaneously consider the interactions and impacts among multiple factors, thus revealing the relationships of synergistic changes. Therefore, geographic detector can encompass the analysis of synergistic changes.

We have added the following text in the Sect 4.6.1 (lines 647 to 666): "Tables 5 and 6 list q values for individual factors, together with p showing the absence of statistical significance in many cases, especially over the YRD, and often the explanatory power is not high when the significance is low. These data show that cloud parameters are dominated by aerosol effects over the ECS but meteorological influences on cloud parameters predominate over the YRD, as was also concluded from the analysis from "traditional" statistical methods presented in Section 4.5 and these conclusions are consistent with the results published by Andersen and Cermak (2015). Among the meteorological parameters, we also find that PVV (with highest q in the three meteorological parameters) predominantely influences cloud parameters over the ECS. Jones et al. (2009) and Jia et al. (2022) reported that stronger aerosol cloud interactions typically occur under higher updraft velocity conditions. In addition, we find that CTP is mainly affected by RH (q = 0.74***) and PVV (q = 0.56) over the YRD, as suggested by Koren et al. (2010). Koren et al. reported that observed cloud top height correlates best with model pressure updraft velocity and relative humidity. To some extent, LTS influences CER (q = 0.44***) and LWP (q = 0.43***) over the ECS, while, in contrast, over the YRD LTS predominately influences CF (q = 0.50***) and LWP (q = 0.55***). Matsui et al. (2004) and Tan et al. (2017) reported that aerosol impact on CER is stronger in more dynamic environments that feature a lower LTS and argue that very high LTS environments dynamically suppress cloud droplet growth and reduce aci intensity. While strong correlations between AOD and cloud parameters have been previously observed, they are likely due to the swelling of aerosol particles in humid airmasses (Quaas et al, 2010), rather than an aerosol influence, which is in

agreement with findings by, e.g., Myhre et al. (2007), Twohy et al. (2009) and Quaas et al. (2010)."

This study provides a general description of the sensitivity (S) of cloud parameters to the influence of different aerosol and meteorological parameters over the YRD and the ECS. Correlations between AOD and cloud parameters are found over the target regions, which can be attributed in part to the influence of general circulation. In general, there are many relations between the various parameters, both related to cloud microphysics and meteorology. Thus, establishing cause and effect relationships between parameters is difficult and must be made with care. It is not possible to completely separate meteorological influences from aerosol influences on clouds. This work can therefore only provide further evidence of the aerosol and meteorological effect on clouds and quantify the relative contributions and combined effects on clouds, but cannot quantify the absolute contributions with confidence.

In the current study, based on a regional scale of 9°x9°, the GDM method is used to explore the relative importance of various factors on cloud parameters and identify possible correlations between different factors. In the future, aerosol cloud interactions can be studied on smaller regional scale (<4°x4°) using higher resolution source data.

Other comments:

1. The name "ACI index" is vague. That quantity is really a sensitivity of cloud effective radius to changes in aerosol optical depth, in a similar way to beta_ln(N)-ln(tau) in Bellouin et al. (2020, doi: 10.1029/2019RG000660)

**Answer**: Following your suggestions below (in particular comment 5 referring to lines 55-78) we have changed the nomenclature and use sensitivity S (rather than β used in Bellouin et al. (2020)) and changed the title of Section 3.1 to "Sensitivity of cloud parameters to changes in aerosol concentrations" and used sensitivity throughout the manuscript. We also explained that we don't use cloud droplet number concentration ($N_d$) but cloud droplet effective radius (CER) and why we made this choice (see lines 175-186).

2. Lines 26 and 27: "significant" – is that in the statistical sense?

**Answer**: Yes, it is in the statistical sense and "significant" has been removed in order to make clear presentation.

3. Line 47: "in practice" in the atmospheric sciences. Other disciplines use the term more properly.

**Answer**: Thank you, we have changed "in practice" to "usually".

4. Lines 44-54: Those generalities on aerosols are not necessary, so that section could be shortened. In fact, the introduction could start directly from line 54: "Aerosol particles are important for climate…"

**Answer**: The manuscript discusses aerosol cloud interaction in contrasting regions as regards aerosol properties (high/low concentrations, composition). Therefore we feel that a short introduction on aerosols, their origin and their variability in space and time is appropriate. This short text also provides context for the description of the choice of study area (Section 2.1).

5. Lines 55-78: Note that many papers since Chapter 7 of the IPCC AR5 (Boucher et al. 2013) use the concept of aerosol-radiation and aerosol-cloud interactions and their respective adjustments (e.g., Bellouin et al. 2020, Quaas et al. 2022 10.5194/acp-22-12221-2022). The terms direct/1st indirect/2nd indirect remain in use in parts of the community, but it would be good to connect to the new terminology.

**Answer**: Thank you for this comment. Because the term "(in)direct" is still used quite often, also in recent publications, and in particular in the older literature, we had followed this terminology which is more common to us. However, we have of course also noticed the change and, although it was not easy to follow up on your comment, we have made an attempt and hope we have done it correctly, made no large mistakes, and done it everywhere where appropriate throughout the revised manuscript.

6. Lines 85-86: It should be said that using AOD as a proxy for aerosol concentrations when looking at aerosol-cloud interactions raises issues. See Section 6 of Bellouin et al. (2020).

**Answer**: Thank you for this comment. Indeed there are issues with the use of AOD and often AI (the product of AOD and the Ångström parameter, AE) is used. However there are also issues with AE from satellites and AE has even been withdrawn as a MODIS product from the more recent collections. The use of AOD in aci studies is discussed in the Introduction (lines 106-122): "In studies on S utilizing satellite data, which is the subject of the current study, the aerosol optical depth (AOD) is often used as a proxy for the aerosol concentration, which is justified by the correlation of AOD and CCN published by Andreae (2009). However, AOD is determined by all aerosol particles in the atmospheric column, including particles that do not act as CCN, depends on the relative humidity (RH) throughout the atmospheric column, does not provide information on chemical composition and may be influenced by aerosol in disconnected layers. The use of the Aerosol Index (AI), the product of AOD and the Ångström Exponent (AE; describing the spectral variation of AOD), is suggested as a better indicator of CCN because AE includes information on aerosol size (e.g., Nakajima et al., 2001). However, the AE is determined from AOD retrieved at two or more wavelengths and the evaluation of the results versus ground-based reference data shows the large uncertainty in AE. Therefore, in recent MODIS product Collections, AE is not provided over land (e.g., Levy et al., 2013; Kourtidis et al., 2015). AE is also not well-defined for low AOD for which uncertainty is largest (Bellouin et al., 2020; Gryspeerdt et al., 2023). The issues associated with using AOD or AI as proxy for CCN were discussed by, among others, Rosenfeld et al. (2014) who do not recommend the use of AI while also concluding that no better proxy is available. Therefore, in this study, AOD is used as a proxy for CCN to study S. It is noted that in other studies, e.g., Jia et al., 2022, both AOD and AI have been used and the results show similar behaviour.".

7. Line 168: Andreae (2009) is often cited as justification for using AOD for looking at aerosol-cloud interactions, but ironically its Figure 1 shows that the correlation only exists across aerosol regimes. For a given regime (as done in the present study) there is essentially no correlation. I could not see why Kourtidis et al. (2015) justifies the use of AOD, but I may have missed it.

**Answer**: See our response to your comment 6 as regards using AOD as aerosol proxy. Specifically to this comment 7: Andreae (2009) plots AOD vs CCN for 4 aerosol types, which happen to be separated in two groups for low and high CCN and indeed the correlation was evaluated across these 4 aerosol regimes. However, it is noted that the number of data pairs is scarce but within each aerosol type the AOD increases with increasing CCN. This AOD-CCN

relation may however vary between aerosol types (as would be expected for aerosol types with different composition and thus also hygroscopic properties), but the number of data points is too small to derive different relationships for different aerosol types. Furthermore, the aerosol types over each of the two study regions varies with season (seasonal emissions like desert dust in spring, biomass burning from different sources, domestic heating in winter) and large scale meteorological condition resulting in different transport pathways in different seasons.

We referenced Kourtidis et al. (2015) because these authors justify the use of AOD instead of AE based on personal communication with Lorraine Remer, as mentioned in our response to your comment 6.

8. Lines 169-170: That assumes that cloud contamination has a lesser impact on smaller AODs. Is that true?

In aerosol retrieval, cloud screening is a major source for over-estimation of the AOD, in particular in the vicinity of clouds which is important for aci. The impact of undetected clouds depends on the COD, and may be small (in an absolute sense) for thin Cirrus clouds, but also discrimination between high AOD and clouds is often a problem. The removal of cloud-contaminated pixels is not straightforward. A post-processing method shows the effect of removal of residual clouds on the AOD in both relatively clean and polluted areas (doi:10.5194/amt-10-491-2017). Alternative methods have been proposed such as setting a threshold for AOD < 0.6 proposed by Brendan et al. (2006) who used the MOD06 Collection 04 cloud product. Brendan et al. (2006) conclude with "The cloud masking technique in the recently updated Collection 05 cloud retrieval algorithm has been improved, and the Collection 05 cloud products available in the near future will largely eliminate the aerosol contamination effect". Christenson et al. (ACP 2017) used MOD06 C6 data (1km x1km) and reported that "large aerosol optical depths remain in the MODIS-observed pixels near cloud edges, due primarily to 3-D effects (Varnái and Marshak, 2009) and the swelling of aerosols by higher relative humidity." And "Varnái and Marshak (2009) also noted that beyond 15 km contamination effects were minimized in MODIS data (1km x1km)." Therefore Christensen et al. only used data pairs beyond the 15 km length scale in their aci study.

In our study we use MODIS L3 collection 6.1 with a spatial resolution of 1°x1°. Comparisons with surface-based sun photometer data revealed that Collection 6 should improve upon Collection 5, and overall, 69.4% of MODIS Collection 6 AOD fell within an expected uncertainty of ± (0.05 + 15%) (Levy et al., 2013; Tan et al., 2017). In this study, to eliminate 1° by 1° scenes in which the aerosol distribution is heterogeneous, retrievals with a standard deviation higher than the mean values are discarded (Saponaro et al., 2017; Jia et al., 2022). In addition, many previous researches do not set a threshold of AOD when using MODIS L3 C6 data (Grandey and Stier, 2010; Tang et al., 2014; Saponaro et al., 2017; Tan et al., 2017; Ma et al., 2018; Jia et al., 2022). Based on these findings, we used the larger threshold of 1.5.

We have added the following text to Section 2.2 (lines 252-260): "The choice of this threshold, rather than 0.6 used by Brendan et al. (2006), who used MOD06 Collection 04 products, is based on reports by Christenson et al. (2017) and (Varnái and Marshak, 2009). Christenson et al. (2017) used MOD06 C6 data (1km x1km) and reported that "large aerosol optical depths remain in the MODIS-observed pixels near cloud edges, due primarily to 3-D effects (Varnái and Marshak, 2009) and the swelling of aerosols by higher relative humidity." Varnái and Marshak (2009) noted that beyond 15 km contamination effects were minimized in MODIS data (1km x1km). Furthermore, we discarded scenes (1° by 1°) in which the aerosol distribution

is heterogeneous, i.e. with a standard deviation higher than the mean value (Saponaro et al., 2017; Jia et al., 2022)."

9. Line 207: "intermittently" Probably not the correct word. Interchangeably?

**Answer**: Thank you for this suggestion, we have changed "intermittently" to "Interchangeably"

10. Line 231: Does q sum up to 1 for all factors considered? Is it also able to quantify an unexplained fraction that could suggest the need for more factors?

**Answer:** This comment is similar to a comment made by Referee#1 and was addressed in our response to your comment 2. Because the contribution of each independent variable (each factor) is calculated separately according to Eq. (2), the contributions of some factors (if needed) that are not considered can also be calculated separately according to Eq. (2).

11. Figure 3: What does that Figure tell the reader? It's impossible to say from its caption or from lines 241-245. The discussion needs to cover each of the panels in turn.

**Answer:** To better explain the Figure and clarify "interaction detector" and "interaction q-values", we have replaced the text below figure 2 (see lines 354-374 in the revised manuscript) with "The interaction detector can be used to test for the influence of interaction between different influencing factors, e.g., x1 and x2, on the dependent factor (y) and whether this interaction weakens or enhances the influence of each of x1 or x2 on the dependent variable, y, or whether they are independent in influencing y. For example, Figure 3(a) shows the spatial distribution of the dependent variable, y. The factors x1 and x2 both vary across the study region, but in different ways, and for each factor different sub-regions can be distinguished by application of the Jenks classification method described above to each factor separately. This is illustrated in Figures 3(b) and 3(c) where, as an example, three different sub-regions are considered for each factor. Usually, the dependent variable y is influenced by several different factors xi (Figure 3) and the combined effect of two or more factors may have a weaker or stronger influence on y than each of the individual factors. The q values for the influences of factors x1 and x2 on y, obtained from the application of the factor detector method (Eq. 2), may be represented as q (x1) and q (x2). Hence, a new spatial unit and subregions may be generated by overlaying the factor strata x1 and x2, written as x1∩x2, where ∩ denotes the interaction between factor strata x1 and x2 as illustrated in Figure 3(d). Thus, the q value of the interaction of x1∩x2 may be obtained, represented as q (x1∩x2). Comparing the q value of the interaction of the pair of factors and the q value of each of the two individual factors, five categories of the interaction factor relationship can be considered which are summarized in Table 2. If q(x1∩x2) > q(x1) + q(x2), this is referred to as a nonlinear enhancement of two variables. And if q(x1∩x2) > Max[q(x1), q(x2)], this is referred to as a bilinear enhancement of two variables. The occurrence of nonlinear enhancement and bilinear enhancement are indicated with the q values in Table 2 and in the caption of Figure 7.".

12. Lines 251-252: "for several different purposes". Give examples based on the papers cited.

**Answer:** Examples based on the GDM have been added in the revised manuscript below Table 2 (lines 383-387): "The geographical detector method has been used to detect influencing factors for several different purposes (e.g., Wang et al., 2018; Zhang and Zhao, 2018; Zhou et al., 2018). For example, the GDM was used to detect the influence of annual and seasonal

factors on the spatial-temporal characteristics of surface water quality (Wang et al., 2018). Other examples are the application of the GDM to examine factors influencing regional energy-related carbon emissions (Zhang and Zhao, 2018) and to examine effects of socioeconomic development on fine particulate matter (PM2.5) in China (Zhou et al., 2018)."

13. Line 265: "averaged over the years 2008-2022". Does the study use 14-year averaged distributions, or a less dramatic averaging (e.g., multi-annual monthly means)? How can correlation of distributions averaged over such a long period still inform about the physical correlation between clouds and aerosols?

**Answer:** In Figure 4 we provide and overview of the data as averages over the whole study period (2008-2022). However, in the text we indicate that individual data pairs are used in the research, e.g., in Section 3.2. (lines 338-339) "In this study, multi-years of mean values of influencing factors (x) and dependent factors (y) were calculated for each raster grid" and in Section 4.2. (lines 429-431) "To investigate S, we used correlated data pairs for 15 years and the data was binned in AOD intervals with a bin width of 0.02, and the CER data in each AOD bin were averaged."

The use of multi-year averages is not uncommon in aci studies, e.g. Ma et al. (GRL 2018) use 2003-2016. Such large samples allow for large numbers of data pairs. However, with hindsight we agree that 15 years is very long. We have thought about shorter periods and realized that, in principle, periods were included when the AOD was at its maximum (2008-2014) and when the AOD was decreasing in response to implementation of emission reduction policy. We therefore split the data sets for these 2 periods and plotted CER vs AOD, see the Figures 1 and 2 below. We noticed that over the ECS there was not a significant difference between the CER/AOD relations during these two periods. However over the YRD, for the high AOD period, CER clearly decreased with increasing AOD for $0.1<AOD<0.3$ and for larger AOD the CER increased with R=0.87. For the second period, however, there was no clear correlation between CER and AOD for both AOD intervals. The data also show that over the YRD the CER for AOD>0.3 increased to larger values during the first period than during the second period. We did not look for explanations of this difference, possibly the aerosol properties changed in response to emission reduction, or confounding meteorological factors played a role.

We also looked for shorter periods, considering each year between 2008 and 2022. The results show similar behavior for each year over both study areas with interannual variations between the fits, and thus the value of S. However, the statistical significance is low (large p) due to the small number of samples.

These findings were briefly summarized in the discussion (lines 733-746): "These results were obtained using data from a period of 15 years. During this period, the aerosol properties changed in response to expanding economy, resulting in the increase of the AOD until 2007, and the implementation of emission reduction policy resulting in the decrease of the AOD from 2014 which flattened from about 2018 (de Leeuw et al., 2021; 2022; 2023). To account for these changes, the sensitivity S was determined for the periods 2008-2014 and 2014-2022, without stratification for LWP (see Figures S1 and S2 in the Supplementary). The results for the ECS show no significant difference between the CER-AOD relations during these two periods. Over the YRD, however, the data for 2008-2014 show a clear decrease of CER with increasing AOD for $0.1<AOD<0.3$ and for larger AOD the CER increased, with a statistical significant correlation (R=0.87) and S=0.10 as compared to S=0.08 for the whole period. In

contrast, the data for 2014-2022 show no clear correlation between CER and AOD for both AOD intervals over the YRD. A similar exercise for shorter periods, i.e. for each year between 2008 and 2022, show similar behavior as for the whole period 2008-2022, over both study areas, with interannual variations of the value of S. However, the statistical significance is low (large p) due to the small number of data samples in each year."

In the GDM, the y data are recorded in a raster grid, over a total study area of 9°x9°, as illustrated in Figure 3 (Figure 2 in the revised manuscript). The data in the raster grid is transformed into dot files, each dot containing a value for y and for one of the influencing parameters x. The dependent (y) and influencing (x) parameters are separated into 2 layers with the same grid. As the resolution of MYD08 data used in this study is 1°x1°, the data transformed into dot files is based on raster grid 1°x1°. Thus, 15-year averaged distributions of clouds (y, 5 layers) and aerosols/meteorological conditions (x, 4 layers) are used as input in the GDM. This is specified in the text on page 13 of the revised manuscript (see lines 338-341): "In this study, multi-years of mean values of influencing factors (x) and dependent factors (y) were calculated for each raster grid. Then, we classified the influencing factors (e.g. AOD and meteorological parameters) into 5 sub-regions by the Jenks natural breaks classification method (Brewer and Pickle, 2002).".

[Figure]

**Figure 1. CER vs AOD over the YRD for the periods 2008-2014 (left) and 2015-2022 (right).**

[Figure]

**Figure 2. CER vs AOD over the ECS for the periods 2008-2014 (left) and 2015-2022 (right).**

[Figure]

**Figure 3. The principle of the geographical detector method. See text for explanation.**

14. Lines 285-286: Or it could be due to meteorological factors.

**Answer:** In this section we describe the observations based on Figure 4 in the revised manuscript. We note that over ocean the aerosol properties are different than over land and certainly sea spray aerosol is abundant, while over land other aerosol types dominate. In the next Section we notice that the sensitivity of CER to AOD is larger over ocean than over land, which confirms the observations in lines 285-286 (now lines 421-422 in the revised manuscript). Certainly, meteorological factors influence the aci too and the interactive q-factors presented and discussed in Section 4.6.2 show that the combined effect has a larger influence than the effect of one factor alone. However, we prefer to structure the paper and go step by step through the different aspects. To explain this, we have added the following text above Figure 4 (lines 422-424): "The influence of different factors on the sensitivity of cloud parameters to aerosol and the adjustments are discussed in the following sections, based on both statistical methods and the application of the GDM."

15. Line 373: Missing word: "in the range of"

**Answer:** Thank you for this comment: corrected.

---

## Author Response (AR2)

**Response to Referee #1**

I thank the authors for their excellent response to the majority of my points and the extra analysis they have done. My previous points have been largely addressed, although a few small points remain that I think would be good to clarify before publication.

The authors are grateful to Referee #1 for the valuable time spent on thorough reading of our revised manuscript and responses. The constructive comments guided us to further improve the manuscript. We have taken notice of all comments to the revised version, listed below in black, with our responses in red. Associated changes in the revised manuscript (using "track changes") are also indicated, copied in the responses in "quotes" and line numbers are provided.

**Main points**

1. Many thanks for this additional explanation of the q-factors. I have one further question about the interpretation. With the new text 'Explanatory power of combined effects is larger...' (L44), this seems to imply that the impact of aerosol and meteorological factors are combining to generate a larger effect. Could it not be some other factor that causes the meteorology and aerosol to co-vary along with the clouds, such that it appears there is a combined effect, when it is actually a confounder?

**Answer**: Yes, this could indeed be the case. The larger correlation and high explanatory power of combined pairs of aerosol and a meteorological parameter, it is not clear whether there is a combined or a confounding effect on cloud properties.

The text in lines 46-50: "The results from the GDM analysis show that the explanatory power of the combined effects of aerosol and a meteorological parameter is larger than that of each parameter alone." has been changed to "The results from the GDM analysis show that cloud parameters are more sensitive to the combination of aerosol and a meteorological parameter than to each parameter alone but confounding effects due to co-variation of both parameters cannot be excluded." The sentence "Thus, the GDM provides an alternative way to obtain information on confounding effects of different parameters." has been removed.

2. It is good to have a justification for using the CER instead of the $N_d$, but some of the points should be referenced or removed. The studies that used CER alone were not really looking at the Twomey effect in isolation, such that they were not really studying the $RF_{aci}$ either (McComiskey and Feingold, 2012). While $N_d$ is affected by biases in the CER retrieval, these are different to the CER biases alone (and in some cases may offset each other; Painemal and Zuidema, JGR, 2011). For marine stratocumulus clouds, the $N_d$ retrieval appears to be surprisingly accurate (Gryspeerdt et al, ACP, 2022). I would also note that while it is clear there is a relationship between CER and LWP, given that LWP is calculated from CER and cloud optical depth (as is $N_d$), neither of these is a "retrieval error" as such. All the properties could be retrieved perfectly and you could still find a relationship between CER and LWP (particularly for adiabatic clouds).

**Answer**: Thank you for this comment. The text in the Introduction (lines 181-190) has been reorganized as "It is noted that $RF_{aci}$ is formulated in terms of $N_d$, whereas studies on the Twomey effects often use CER alone instead of $N_d$, such that they were not really looking at the Twomey effect in isolation and not really studying the $RF_{aci}$ either (McComiskey and Feingold, 2012). CER is readily available as a satellite retrieval product, although in particular

over land the reliability is questioned (Grandey and Stier, 2010), whereas $N_d$ is derived from CER and the cloud optical thickness (COT) (e.g., Grandey and Stier, 2010; Arola et al., 2022). While $N_d$ is affected by biases in the CER retrieval, these are different to the CER biases alone (and in some cases may offset each other; Painemal and Zuidema, JGR, 2011). For marine stratocumulus clouds, the $N_d$ retrieval appears to be surprisingly accurate (Gryspeerdt et al, 2022)."

3. I think it would be fine just to state that you are stratifying by LWP and focussing on CER sensitivity (without a focus on the RFaci). I would still suggest you consider $N_d$ for future studies. There are easily available $N_d$ products that might be useful (such doi:10.5285/864a46cc65054008857ee5bb772a2a2b and https://catalogue.ceda.ac.uk/uuid/cf97ccc802d348ec8a3b6f2995dfbbff).

**Answer**: Thank you for this valuable comment. We will consider $N_d$ for future studies. In this paper we keep the sentence "the current study focuses on understanding effects of different parameters on CER sensitivity to aerosol rather than the application to determine $RF_{aci}$" (Line 193-194) as it is.

4. Also, it seems like if the GDM doesn't detect significant relationships for regions smaller than 9x9. When compared to the 4x4 region recommended by Grandey and Stier (2010), might this suggest that the results in this work are due to a misleading spatial covariation? Naturally, the GDM depends on spatial variability, so cannot easily operate on small regions (as I understand it) and there are other benefits to using it. I am not suggesting that this invalidates your results or that you have to do a lot of extra work, but I think this is an important aspect that should be discussed in the paper so it is clear to further potential users of this method.

**Answer**: Thank you for this comment. We have added the following text at the end of Discussion (lines 811-816): "As regards large regions: Grandey and Stier (2010) recommend 4° x 4° as the largest size and "if data exist at higher gridded resolution the possibility of analyzing data at this higher resolution should be seriously considered." In this study the resolution of MYD08 data used is 1° x 1°, the GDM doesn't detect significant relationships for regions smaller than 9° x 9° due to insufficient samples. In the future, higher resolution data can be used for GDM by controlling the size of the study area to be less than 4° x 4°."

**Minor points**

1. L68 - $RF_{ari}$ is defined before RF

**Answer**: We have removed the definition of RFari before RF.

The text in the Introduction (lines 69-71): "Aerosol particles affect climate by their interaction with radiation (aerosol radiation interaction, ari) which exerts a radiative forcing (RFari) on the Earth'energy budget which results in rapid adjustments of global mean atmospheric quantities such as temperature." has been changed to "Aerosol particles affect climate by their interaction with radiation (aerosol radiation interaction, ari) which exerts a radiative forcing on the Earth energy budget, which results in rapid adjustments of global mean atmospheric quantities such as temperature."

2. L89 - The 'cloud lifetime effect' is less used today, as it is not clear a cloud lifetime is really involved (see the IPCC AR5 chapter on clouds and aerosols). I think you could just say 'are sometimes referred to', if you want to keep the terms in.

**Answer:** done. The text in the Introduction (lines 91-93): "These two effects of aci are also referred to as the cloud albedo and cloud lifetime effects (Quaas et al., 2008)." has been changed to "These two effects of aci are sometimes referred to as the cloud albedo and cloud lifetime effects (Quaas et al., 2008)."

3. L254 - Christensen - this paper also doesn't seem to show up in the references.

**Answer**: We apologize for the misspelling the name. The paper by Christensen et al. is included in the references and the name Christensen has been corrected in the text (lines 260-263): "… is based on reports by Christensen et al. (2017) and Varnái and Marshak (2009), rather than 0.6 used by Brendan et al. (20062005), who used MOD06 Collection 04 products. Christensen et al. (2017) used MOD06 C6 data (1km x1km) and reported that…".

Christensen, M. W., Neubauer, D., Poulsen, C. A., Thomas, G. E., McGarragh, G. R., Povey, A. C., Proud, S. R., Grainger, R. G.: Unveiling aerosol-cloud interactions - Part 1: Cloud contamination in satellite products enhances the aerosol indirect forcing estimate, Atmos. Chem. Phys., 17, 13151-13164, 2017.

4. L274 - I don't think Liu, 2002b is the usual ERA5 reference

**Answer**: Done. We have deleted the reference.

5. L323 - I am not clear what 'dot files' are. I assume it is some kind of 2D(?)array representation?

**Answer**: The sentence has been corrected. The text (lines 325-327): "The data in the raster grid is transformed into dot files, each dot containing a value for the CER and for one of the influencing parameters x." has been changed to "The data in the raster grid is transformed into 2D point vector files, each point containing a value for the CER and for one of the influencing parameters x.".

6. L420 - Aerosol is likely a secondary control on CF, rather than the primary cause. CER is similar, with cloud depth (and hence LWP) being a more important factor in determining CER, rather than aerosol.

**Answer**: Yes, a large part of the correlation between aerosol and CF is thought to be due to effects other than aerosol-cloud interaction such as aerosol humidification. The text in lines 424-429: "The high values of the CER and CF over the ECS could be due to the dominance sea spray aerosol, the high hygroscopicity of which makes these particles very efficient CCN." has been changed to "The high values of the CER over the ECS could be due to the dominance of sea spray aerosol, the high hygroscopicity of which makes these particles very efficient CCN, which in this environment over ocean with high water vapor concentrations, results in larger CER."

7. L552 - While there is a strong correlation between AOD, CF and CTP, that is not good evidence of an aerosol effect (Quaas et al., ACP, 2010; Gryspeerdt et al., ACP, 2014c).

**Answer**: Yes, we agree. We have added the following text in the Sect 4.4 (lines 562-565): "Although there is a strong correlation between AOD, CF and CTP, this does not imply evidence of an aerosol effect (Quaas et al., ACP, 2010; Gryspeerdt et al., ACP, 2014)."

8. Results section - I would be a little careful in attributing causality to these results, as they show a correlation, but not evidence that the aerosol variation caused some change in cloud properties. This doesn't require much change, but I would watch out for cases talking about the 'effect' of AOD on cloud properties (e.g. L689) or 'influence' (e.g. L787), as it is not clear there is actually an impact of aerosol on cloud properties from these results.

**Answer**: Following these comments, we have reorganized the following two sentences and added some words in the Discussion.

The text in lines 698-702: "The data in Fig. 10 also show that the explanatory power is largest for the combined influence of AOD together with other factors, and is somewhat larger than the influence of AOD alone (Table 6) for all 5 cloud parameters." has been changed to "The data in Fig. 10 also show that cloud parameters are more sensitive to the combination of AOD and a meteorological parameter than to AOD alone (Table 6)."

The text in lines 796-798: "The factor detector analysis (Section 4.6.1) shows that over the ECS, AOD has the largest influence on cloud parameters, as indicated by the large and statistically significant q values." has been changed to "The factor detector analysis (Section 4.6.1) shows that over the ECS, cloud parameters are most sensitive to AOD, as indicated by the large and statistically significant q values."

The text in lines 809-811: "Moreover, it should be noted that although the results show correlations, they do not provide evidence that the aerosol variation indeed causes some change in cloud properties." has been added in the Discussion.